# What Matters in Hierarchical Search for Solving Combinatorial Problems?

## Abstract

Combinatorial problems, particularly the notorious NP-hard tasks, remain a significant challenge for AI research. A common, successful approach to addressing them combines search with heuristics learned from demonstrations with Imitation Learning (IL). Recently, hierarchical planning has emerged as a powerful framework in this context, enabling agents to decompose complex problems into manageable subgoals. However, the foundations of this approach, particularly the behavior and limitations of learned heuristics, remain underexplored. Our goal is to advance research in this area and establish a solid conceptual and empirical foundation. Specifically, we identify the following key characteristics, whose presence favors the choice of hierarchical search methods: *hard-to-learn value functions, complex action spaces, presence of dead ends in the environment,* or *training data collected from diverse sources*. Through in-depth empirical analysis, we establish that hierarchical search methods consistently outperform standard search methods across these dimensions, and we formulate insights for future research. On the practical side, we also propose a set of evaluation guidelines to enable meaningful comparisons between methods and reassess the state-of-the-art algorithms.

## 1 Introduction

The ability to solve discrete tasks that require sophisticated reasoning, particularly those involving NP-hard problems, is essential for advancing AI (Bengio et al., 2021). These include complex problems like theorem proving (Wu et al., 2021; Trinh et al., 2024), constraint satisfaction problem (Achiam et al., 2017), molecule alignment (Needleman & Wunsch, 1970; Smith & Waterman, 1981), social network analysis (Kipf & Welling, 2017), or navigation (LaValle, 2006; Choset et al., 2005). Even driving a car, which typically involves continuous control of steering and speed, requires high-level discrete decision-making, e.g., when to overtake, when to change lanes, or how to navigate through traffic (Kiran et al., 2022).

Addressing such tasks, known as combinatorial problems, requires efficient planning strategies due to the vast and complex search spaces involved (Bruck & Goodman, 1987). A widely adopted solution is to learn heuristics from demonstrations, even suboptimal, via imitation learning – a flexible approach that combined with effective planning methods has become a standard paradigm.

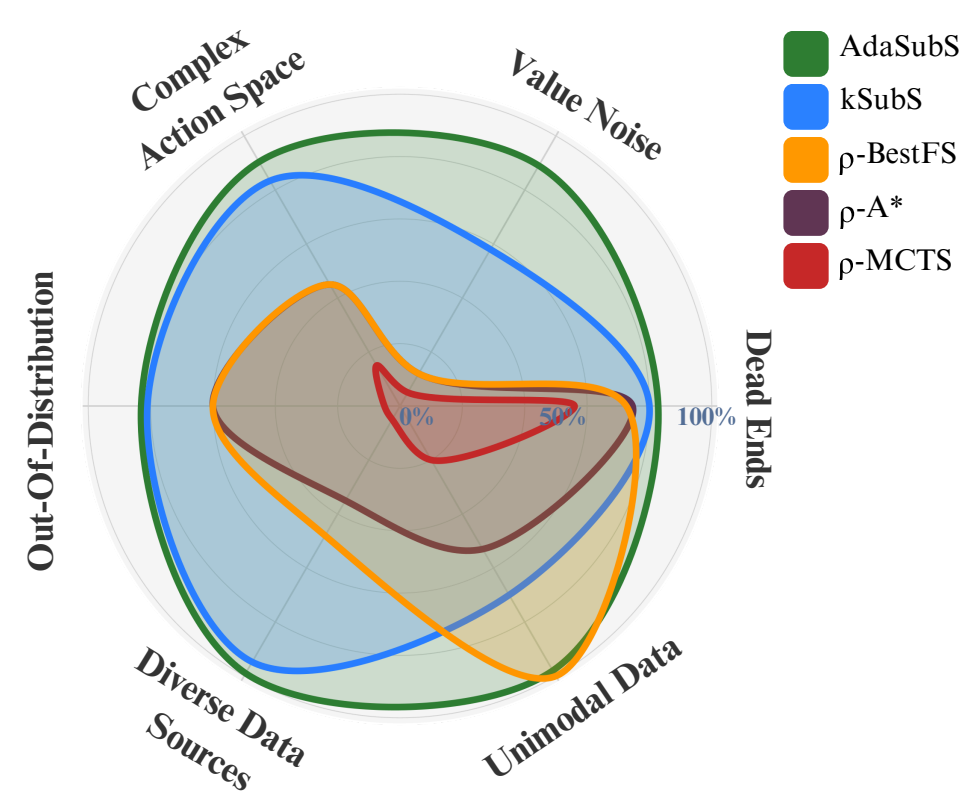

Figure 1: Schematic performance comparison of hierarchical methods (AdaSubS, kSubS) and low-level methods ($\rho$-BestFS, $\rho$-A*, $\rho$-MCTS) across six dimensions studied in our analysis: *handling data collected from diverse sources*, *learning from clean unimodal demonstrations*, *avoiding dead ends*, *performance under high-value approximation errors*, *handling complex action space*, and *generalizing to out-of-distribution instances.*

Recently, hierarchical search emerged as a powerful framework in this context, inspired by how humans plan their actions (Hull, 1932; Fishbach & Dhar, 2005; Kool & Botvinick, 2014). This general-purpose method breaks down a problem into manageable subproblems, or subgoals, making the overall task more tractable. Hierarchical search has been successfully applied to a variety of combinatorial tasks, as evidenced by methods like Subgoal Search (kSubS) (Czechowski et al., 2021), and further advanced by Adaptive Subgoal Search (AdaSubS) (Zawalski et al., 2023), Hierarchical Imitation Planning with Search (HIPS) (Kujanpää et al., 2023a), or HIPS-$\varepsilon$ (Kujanpää et al., 2023b).

Our goal is to advance research in hierarchical planning and establish a solid conceptual and empirical foundation. We choose *kSubS* and *AdaSubS* as representative examples of simple yet effective hierarchical methods. These approaches introduce a single layer of subgoals over low-level search algorithms, enabling us to isolate and measure the benefits of hierarchical decomposition in a controlled manner. Through extensive empirical analysis, we identify four key properties of environments and training data whose presence favors the use of hierarchical search methods: *hard-to-learn value functions, complex action spaces, presence of dead ends in the environment*, or *data collected from diverse sources*. We further analyze these findings from a general perspective and prove theorems that explain the most unexpected results. Our findings offer a clearer understanding of when hierarchical approaches should be preferred over low-level methods.

In summary, our contributions are as follows:

- We conduct a detailed empirical comparison of hierarchical and low-level search methods, both guided by learned heuristics, across a range of combinatorial tasks, revealing consistent trends in performance and robustness.
- We provide a theoretical analysis that explains key empirical findings and extends to a broader class of hierarchical methods, offering insights that generalize beyond the specific algorithms evaluated.
- We identify key tasks and data characteristics that favor hierarchical approaches and propose evaluation guidelines to support more consistent and transparent benchmarking in future research.

## 2 Related Work

**Solving Decision-Making Problems** Decision-making problems are often framed as Markov Decision Processes (MDPs) (Sutton et al., 1999), which can be solved using Reinforcement Learning (RL) algorithms like PPO (Schulman et al., 2017) or DQN (Mnih et al., 2015). These methods learn policies through interaction with the environment. An alternative to learning from trial and error is Imitation Learning (IL), training models directly from offline demonstrations. The availability of large-scale datasets (Walke et al., 2023; Collaboration et al., 2023; Grauman et al., 2022; Dosovitskiy et al., 2017), make it applicable to the most complex domains like robotics (Mandlekar et al., 2018; Edmonds et al., 2017; Kim et al., 2024), autonomous driving (Kelly et al., 2019; Li et al., 2022; Zhang & Cho, 2017), and physics-based control (Kim et al., 2020; Fickinger et al., 2022). Key foundational methods such as Behavioral Cloning (BC) (Sutton & Barto, 1998), Inverse Reinforcement Learning (IRL) (Baker et al., 2009), or DAgger (Ross et al., 2011) have been instrumental in advancing IL for complex environments where direct exploration is less practical. In this work, we use IL to train components for the search methods, such as the policy and value function, which is a widely adopted approach (Czechowski et al., 2021; Zawalski et al., 2023; Takano, 2023).

**Subgoal Methods** Hierarchical Reinforcement Learning methods tackle complex decision-making tasks by breaking them into subgoals. HIRO (Nachum et al., 2018) reuses past data by goal relabeling. HAC (Levy et al., 2019) builds a multi-layer hierarchy of policies trained with hindsight. Hierarchical Diffuser (Chen et al., 2024) learns to predict future states with diffusion models. Graph-based methods, such as SoRB (Eysenbach et al., 2019) or DHRL (Lee et al., 2022) build a high-level graph of states, which then allow for efficient shortest path finding. GCP (Pertsch et al., 2020) learns to predict middle states between two given observations. Algorithms such as HPG (Ghavamzadeh & Mahadevan, 2003) or H-DDPG (Yang et al., 2018) extend the classical RL algorithms to the hierarchical setting.

In the area of combinatorial problems, there has been growing interest in applying HRL techniques. kSubS (Czechowski et al., 2021) introduces a hierarchical search algorithm that iteratively generates subgoals to construct a search tree. Building on this, AdaSubS (Zawalski et al., 2023) incorporates multiple subgoal generators, each trained to predict subgoals at different distances from the target, allowing for dynamic adaptation of the planning horizon based on problem complexity. HIPS (Kujanpää et al., 2023a) and HIPS-$\varepsilon$ (Kujanpää et al., 2023b) perform search using subgoals generated by VQ-VAE models (van den Oord et al., 2017).

**Low-level Search Algorithms** Traditional search algorithms like Best-First Search (BestFS), A* (Cormen et al., 2009; Russell & Norvig, 2009), and Monte Carlo Tree Search (MCTS) (Veness et al., 2009; James et al., 2017) have long been the foundation for solving complex decision-making problems. Recent advancements have improved these methods by integrating neural network-based heuristics, improving their efficiency in large search spaces (Silver et al., 2018; Yonetani et al., 2021). A variant of $\rho$-BestFS used in (Czechowski et al., 2021; Zawalski et al., 2023), leverage heuristics learned through behavioral cloning to guide search. More recent algorithms, like PHS (Orseau & Lelis, 2021) or LevinTS (Orseau et al., 2023), combine policy-driven and value-based approaches, offering both theoretical guarantees and strong empirical performance. Additionally, PDDL planners (Haslum et al., 2019) solve decision-making problems by using predefined action models and goals, with domain-independent planners offering broad applicability, while domain-specific ones achieve higher performance in specialized tasks.

**Empirical Studies on Algorithmic Performance** Our work aligns with recent empirical studies that investigate the conditions under which various algorithmic approaches excel. For instance, Andrychowicz et al. (2020) investigate how specific design choices influence the performance of PPO, while other research compares offline reinforcement learning with behavioral cloning (Kumar et al., 2022) or explores design choices for language-conditioned robotic imitation learning (Mees et al., 2022). In this paper, we focus on hierarchical search in combinatorial problems, specifically studying the conditions where hierarchical methods outperform low-level planners. To the best of our knowledge, this is the first systematic study of the relationship between hierarchical and low-level search in this context.

## 3 Combinatorial Environments

Our study targets solving combinatorial environments – domains with discrete, compact state representations corresponding to exponentially large configuration spaces, which makes them highly challenging to solve. This class includes several NP-hard problems, such as the Traveling Salesman Problem (Applegate et al., 2006), the Rubik's Cube (Singmaster, 1981), Sokoban (Culberson, 1997), or solving non-linear inequalities (Sahni, 1974). In our study, we specifically focus on goal-reaching tasks. To efficiently solve combinatorial problems an algorithm should have the following desirable properties:

In combinatorial environments, each problem instance is typically entirely distinct from others, making it unrealistic to assume that offline data provides comprehensive state space coverage. This is especially critical in problems like the Rubik's Cube, where even with a vast training dataset, any new state will be entirely different from those previously encountered. Some approaches rely on sufficient state space coverage, but in many combinatorial problems, this assumption is impractical.

1. **Learning from offline data.** Since combinatorial environments are characterized by a large space of possible configurations, learning without priors or handcrafted dense rewards is infeasible due to the challenge of exploration[1]. To address this, a canonical solution is to leverage offline data, even suboptimal. Other possible approaches, such as clever reward shaping, usually require significant domain knowledge.

2. **Combinatorial space abstraction.** In combinatorial environments, each problem instance is typically entirely distinct from others. Hence, it is unrealistic to assume a comprehensive state space

[1] For instance, we tested PPO (Schulman et al., 2017) on the Rubik's Cube, but, unsurprisingly, it failed to make any progress due to never reaching the goal in the haystack of $4.3 \times 10^{19}$ states, hence never observing a positive reward.

coverage by training data or repeated visits to nearby states, an assumption that some approaches implicitly rely on.

3. **Search.** Methods that don't use search and follow a single action trajectory are inherently limited by computational complexity, since they can perform only a constant number of operations before choosing an action. Solving NP-hard problems within a fixed computation budget is computationally infeasible (Bruck & Goodman, 1987).

### 3.1 Problem Formulation

We model each combinatorial environment as a deterministic fully observable planning problem $\Pi = (S, A, f, c, s_0, G)$: $S$ is a state set, $A$ primitive actions, $f : S \times A \to S$ a deterministic transition function, $c : A \to \mathbb{R}_{\geq 0}$ a step cost (we take $c = 1$), $s_0$ the start state, and $G \subseteq S$ the goal set. A *plan* is an action sequence that drives $s_0$ into $G$ with minimum cost $C^{\star}(\Pi)$. The same data induce a deterministic, episodic MDP with binary reward $R(s, a) = \mathbb{1}\{f(s, a) \in G\}$. During training, the agent receives a fixed offline dataset $\mathcal{D} = \{\tau^{(i)}\}_{i=1}^{N}$ of (possibly sub-optimal) trajectories collected on different instances. At test time, given $(\Pi, \mathcal{D})$ and a budget $B$ nodes to visit, the algorithm must return a successful plan or declare failure.

## 4 Subgoal Methods

Subgoal methods, or hierarchical methods, are a family of algorithms designed to solve complex decision-making tasks by breaking down the overall objective into smaller, more manageable subgoals (Sutton et al., 1999). Instead of searching for a sequence of low-level actions that directly lead from the initial state to the goal, the agent first identifies high-level intermediate targets – subgoals – that guide the trajectory toward the final goal. The use of subgoals is widely considered as a method that scales better to longer horizons (Chen et al., 2024; Lee et al., 2022), mitigates errors in value approximations (Czechowski et al., 2021), and reduces overall complexity by decomposing the problem into smaller subproblems (Sutton et al., 1999; Zawalski et al., 2023). The process of searching involves the following components:

- **Subgoal generator** that, given a state within the search tree, outputs a set of subgoals. For instance, a subgoal may be a future state (Czechowski et al., 2021; Zawalski et al., 2023) or a class of desired outcomes (Jiang et al., 2019; Panov & Skrynnik, 2018). See Figure 16 for example subgoals. The subgoal generator can be implemented using models such as transformers with beam search (Czechowski et al., 2021; Zawalski et al., 2023), VQ-VAE (Kujanpää et al., 2023a), or other generative architectures. The generator is used by the planner to construct a search tree of subgoals.
- **Low-level policy** that determines a path of low-level actions between subgoals. For instance, it may be a trained goal-reaching policy (Czechowski et al., 2021; Zawalski et al., 2023), a local search (Czechowski et al., 2021; Kujanpää et al., 2023a), or a stored path from previous episodes (Eysenbach et al., 2019; Lee et al., 2022).
- **Planner** that determines the order in which subgoals are generated. Standard planning algorithms like BestFS (Czechowski et al., 2021), PHS (Kujanpää et al., 2023a), or their modified forms (Zawalski et al., 2023), are typically used.
- **Value function** that estimates the distance between the given state and the goal state. The planner uses this information to select the next node to expand with the subgoal generator. In some works it is also called *heuristic value*. In our study, we focus on value functions learned from demonstrations, but in general, values learned through RL or even scripted heuristics can be used in search.

In our experiments, we use kSubS (Czechowski et al., 2021) and AdaSubS (Zawalski et al., 2023) as subgoal methods well-suited for combinatorial problems, as they satisfy the conditions formulated in Section 3. We

also experimented with HIPS and HIPS-$\varepsilon$ (Kujanpää et al., 2023a;b), but these methods generally fail to solve the problems within a reasonable computational budget. Therefore, their results are omitted from the main text and discussed in Appendix I.

We compare the performance of the selected subgoal approaches against three popular low-level methods: BestFS, A*, and MCTS. To ensure a fair comparison and improve efficiency, we augment these algorithms by using a trained policy to select the top actions before each node expansion. We refer to them as $\rho$-BestFS, $\rho$-A*, and $\rho$-MCTS. A detailed description, analysis, and pseudocode for each of these algorithms can be found in Appendix F. See also Appendix H for diagrams explaining different search methods.

### 4.1 Training Components

In our experiments, the models for both subgoal methods and low-level searches were trained using imitation learning, following standard practice (Nair et al., 2018; Czechowski et al., 2021). Specifically, we collected a dataset of approximately 500 000 trajectories for each environment. Trajectories are sequences of consecutive states and actions leading to the goal state. We used various methods of dataset collection, like hand-crafted algorithms, trained policies, reversed random shuffles, and others, which let us to study the influence of training data characteristics on the performance of search methods. Each component was trained and evaluated in over 30 independent runs with different random seeds, all in a behavioral-cloning setup, a training paradigm known to be very stable.

All components are implemented using HuggingFace Transformers. The subgoal generators and INT policies (including both CLLP and baseline approaches) are built on the BART architecture (Lewis et al., 2020), while the remaining policies and value functions utilize BERT (Devlin et al., 2019).

For the INT task, the learning rates were set to $1 \times 10^{-4}$ for the generator, $1 \times 10^{-4}$ for the CLLP, $3 \times 10^{-4}$ for the policy, and $1 \times 10^{-4}$ for the value network. For the Rubik's Cube task, the respective learning rates were $1 \times 10^{-4}$ for the generator, $5 \times 10^{-4}$ for the CLLP, $3 \times 10^{-7}$ for the policy, and $1 \times 10^{-4}$ for the value network. For both the Sokoban and n-Puzzle tasks, the generator's learning rate was $1 \times 10^{-5}$, while the learning rates for the CLLP, the policy, and the value network were all set to $1 \times 10^{-4}$.

To ensure a fair comparison, all methods shared common components whenever applicable (e.g., each method uses the same value function). This allows us to focus on the differences between the search algorithms, rather than heuristic biases. No additional heuristics were used, ensuring that performance differences arise solely from the algorithmic approaches. Training runs were conducted on a single NVIDIA A100 (40 GB) GPU for up to 48 hours per component, with data loading utilizing one CPU core. Evaluation was performed on a 24-core Xeon Platinum 8268 node with 192 GB of RAM.

More details on training the components, including specific objectives, are provided in Appendices C and D.

### 4.2 Performance Metrics

Our primary performance metric is the *success rate*, defined as the percentage of problem instances solved within a given *complete search budget*. The complete search budget is the total number of visited states in the search tree. In particular, for subgoal methods, the budget includes both the generated subgoals and the states visited by the low-level policy used to connect these subgoals.

By accounting for the total number of visited states, this metric provides a unified and fair comparison of search efficiency across different methods. We argue that reporting only the number of visited subgoal nodes would unfairly favor subgoal methods (see Appendix I for details).

While *wall-clock time* could serve as an alternative budget metric, it suffers from high variance and is sensitive to hardware, making it difficult to reproduce results. Nevertheless, we report wall-clock measurements to validate their correlation with the complete search budget. Detailed discussion of that metric is provided in Appendix G.

## 5 Analysis

We investigate how environmental properties and training data influence the performance of hierarchical methods compared to low-level search approaches in combinatorial tasks. While previous works (Czechowski et al., 2021; Zawalski et al., 2023; Kujanpää et al., 2023a;b) show a considerable advantage of hierarchical methods, our experiments reveal that this advantage is not consistent across all scenarios (see Figures 4 or 5 for specific examples). Specifically, we answer the following research questions:

- Q1. Is hierarchical search more effective than low-level search for solving combinatorial problems?
- Q2. What environmental properties and characteristics of the training data amplify performance differences? When hierarchical search should be preferred over low-level search?
- Q3. What pitfalls should be avoided when interpreting experimental results?

To address these questions, we conducted a wide range of experiments comparing subgoal and low-level search algorithms across a variety of combinatorial tasks. Below, in each subsection we summarize the key findings that reveal the most significant factors affecting performance, followed by a brief discussion. For each finding, we link it to the relevant research questions. The extended analysis of these factors can be found in Appendix B.

We present our findings using the *Rubik's Cube, Sokoban, N-Puzzle*, and *Inequality Theorem Proving* (INT) (Wu et al., 2021) environments[2]. Each of these four combinatorial problems exhibits distinct properties that help us test complementary aspects of hierarchical search: (i) *dead-end frequency*—Sokoban contains many irreversible pushes that can render the goal unreachable, letting us observe whether subgoal discovery avoids such traps; (ii) *action-space complexity*—every INT move selects an axiom and its arguments from millions of possibilities, introducing an extremely large branching factor that stresses reasoning over complex actions; and (iii) *state-space size and horizon length*—Rubik's Cube and N-Puzzle both provide vast state graphs with uniform, low-level moves, making them suitable for studying long-horizon planning and learning from diverse, sub-optimal traces. These benchmarks are widely used in planning research (McAleer et al., 2019; Czechowski et al., 2021) and are known to be NP-hard (Demaine et al., 2018; Culberson, 1997; Ratner & Warmuth, 1986). Since different algorithms exhibit significant performance variations depending on the problem structure, we evaluate them in a range of environments to ensure the robustness of our findings. Detailed descriptions of these environments can be found in Appendix A.

All methods in our study were trained using imitation learning. In particular, all algorithms share the same value function, as stated in Section 4.1. To ensure fair comparisons, we measured complete search budgets, in contrast to counting only high-level search nodes, to avoid giving any unfair advantage to subgoal methods, as discussed in Section 4.2 (which contributes to the research question Q3). We tuned hyperparameters of each method separately for each experiment to ensure optimal performance.

### 5.1 Subgoal Methods are Robust to Diverse Sources of Data

Achieving superhuman performance in complex tasks often involves large-scale datasets of demonstrations obtained from agents with varying skill levels and strategies (Silver et al., 2016). By training models on data collected from a variety of solvers and testing them in the Rubik's Cube and N-Puzzle environments, we show that the variability in training data has a significant impact on the performance of search algorithms. Our training data included algorithmic solvers, computational solvers, and random shuffles, as detailed in Appendix B.1.

As shown in Figures 2-3, subgoal methods consistently outperform low-level methods by a wide margin (Q1). However, when the training dataset is limited to a single source of demonstrations – whether the demonstrations are long and structured or short and direct – this performance gap disappears (see Figures

[2]We note that classical environments have domain-specific solvers that achieve high performance by relying on expert knowledge. However, our goal is to compare general-purpose search methods that require no domain knowledge.

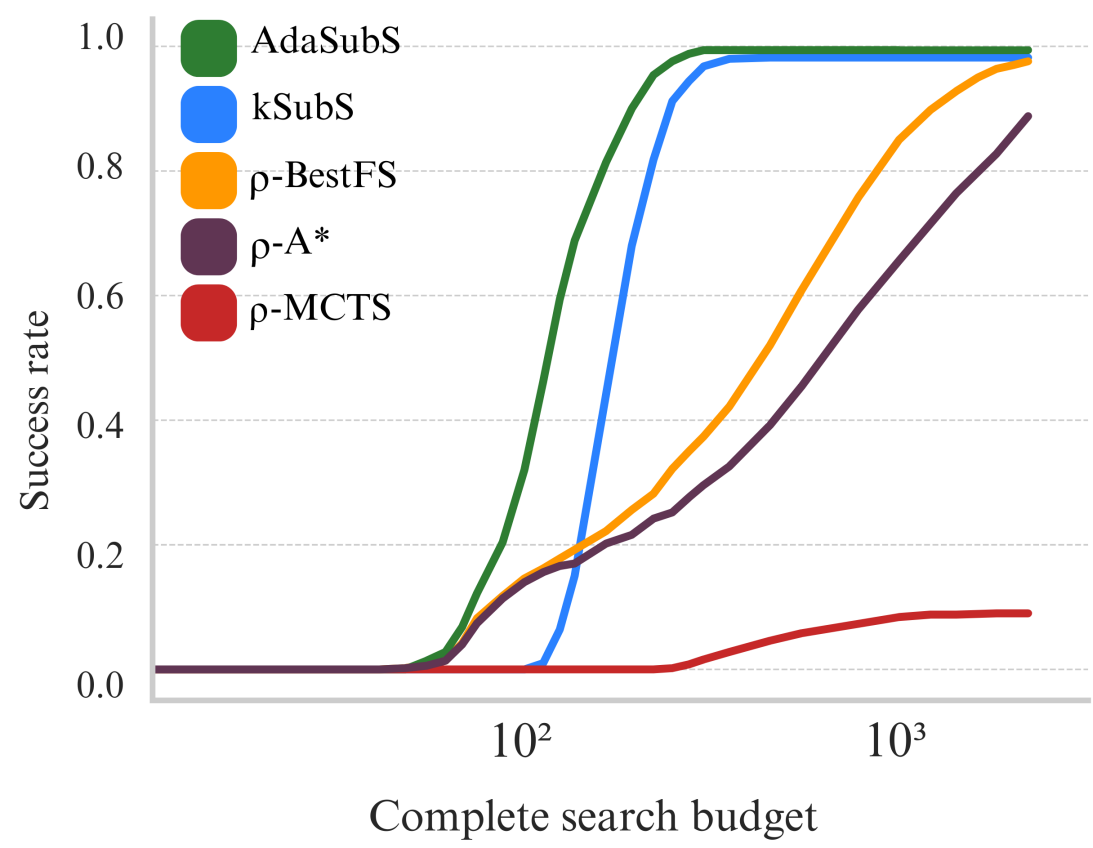

Figure 2: Solving the Rubik's Cube. Components are trained on data from 4 different solvers.

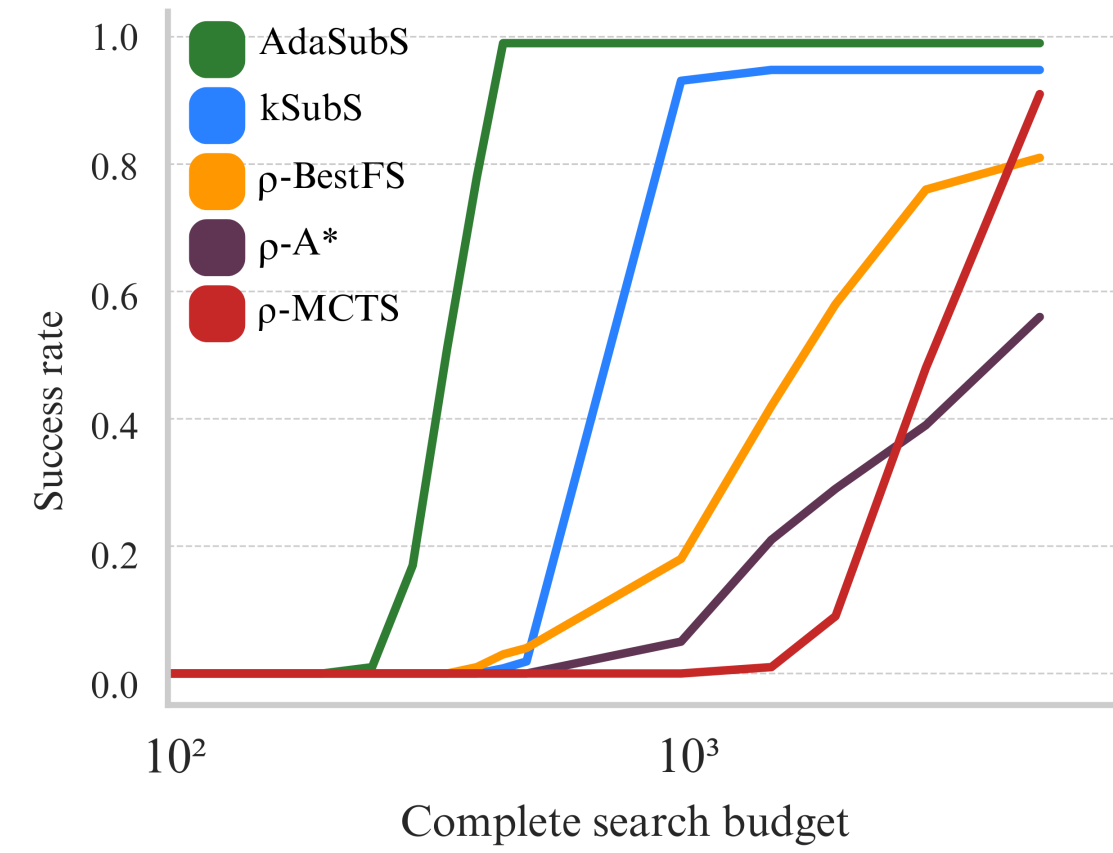

Figure 3: Solving the N-Puzzle. Components are trained on data from 2 different solvers.

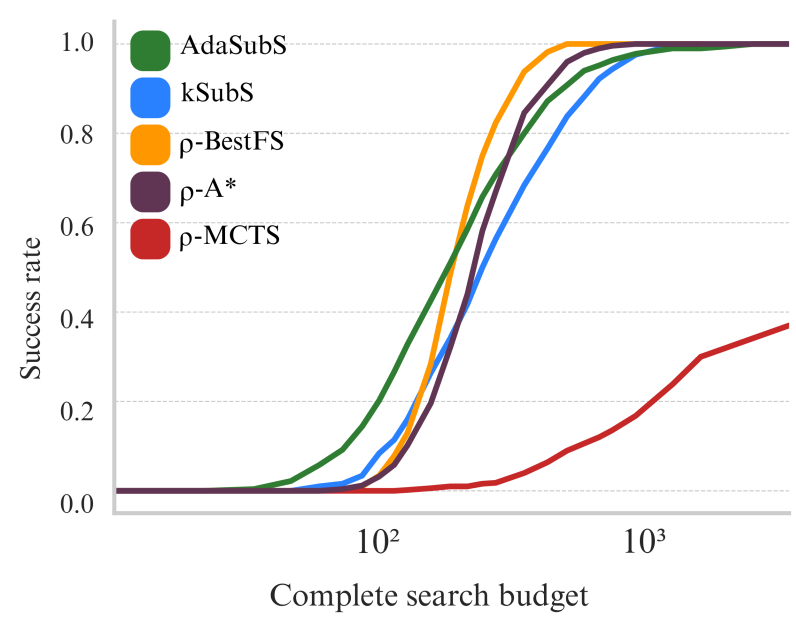

Figure 4: Solving the Rubik's Cube. Components are trained on reversed random shuffles.

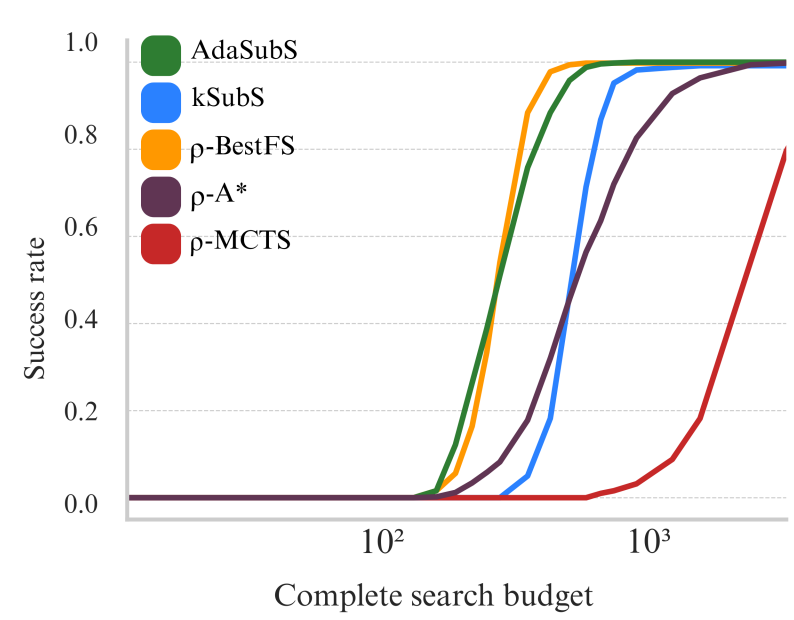

Figure 5: Solving the Rubik's Cube. Components are trained on the *Beginner* algorithmic solver.

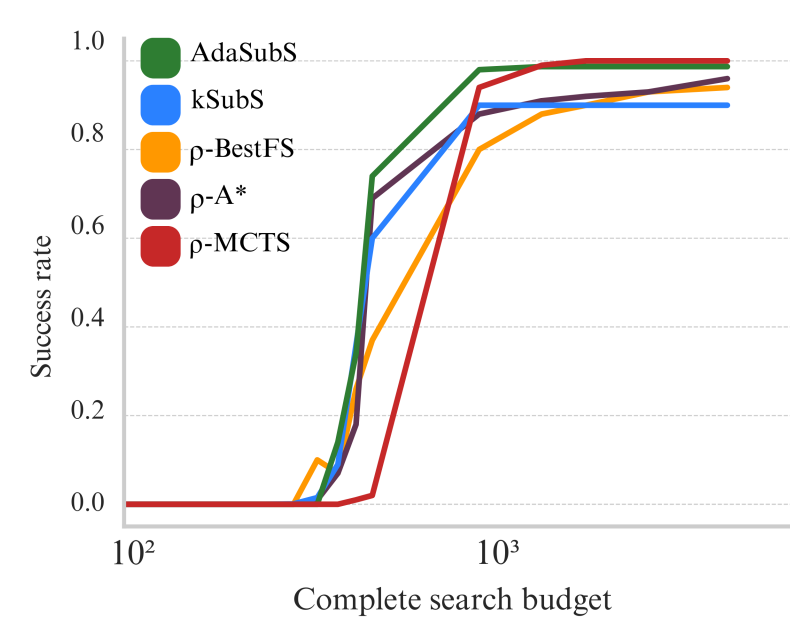

Figure 6: Solving N-Puzzle. Components are trained on an algorithmic solver.

4-6). Notably, subgoal methods, particularly AdaSubS, maintain stable performance across all training setups, while low-level methods are highly sensitive to the characteristics of the training data.

To explain those results, we found that value functions trained on diverse data often fail to assign consistently low values to the initial states of tasks. When demonstrations differ significantly in their length or execution style, the value function learns this variation, leading to inconsistent value predictions. Hierarchical methods can overcome this issue by relying on subgoals. Subgoals enable the agent to make long steps toward the solution, effectively bypassing regions of the state space where the value function is inconsistent or noisy, as it does not need to assess every small step along the way (this property is further studied in Section 5.2). In contrast, low-level methods operate on a finer, step-by-step level, executing small, atomic actions. This makes them more sensitive to the variability in the value function because they must evaluate each intermediate state on the way.

More detailed analysis of the experiments involving diverse data sources is provided in Appendix B.1.

**Takeaway** *Subgoal methods successfully leverage diverse demonstrations (Q2), while low-level search performs better when trained on homogeneous trajectories (Q2).*

### 5.2 Subgoal Methods are Value Noise Filters

We found that the classical search algorithms are highly sensitive to the quality of the value function. To show this in a controlled setting, we added Gaussian noise to the value estimates and observed how different noise levels impacted the success rate of solving tasks.

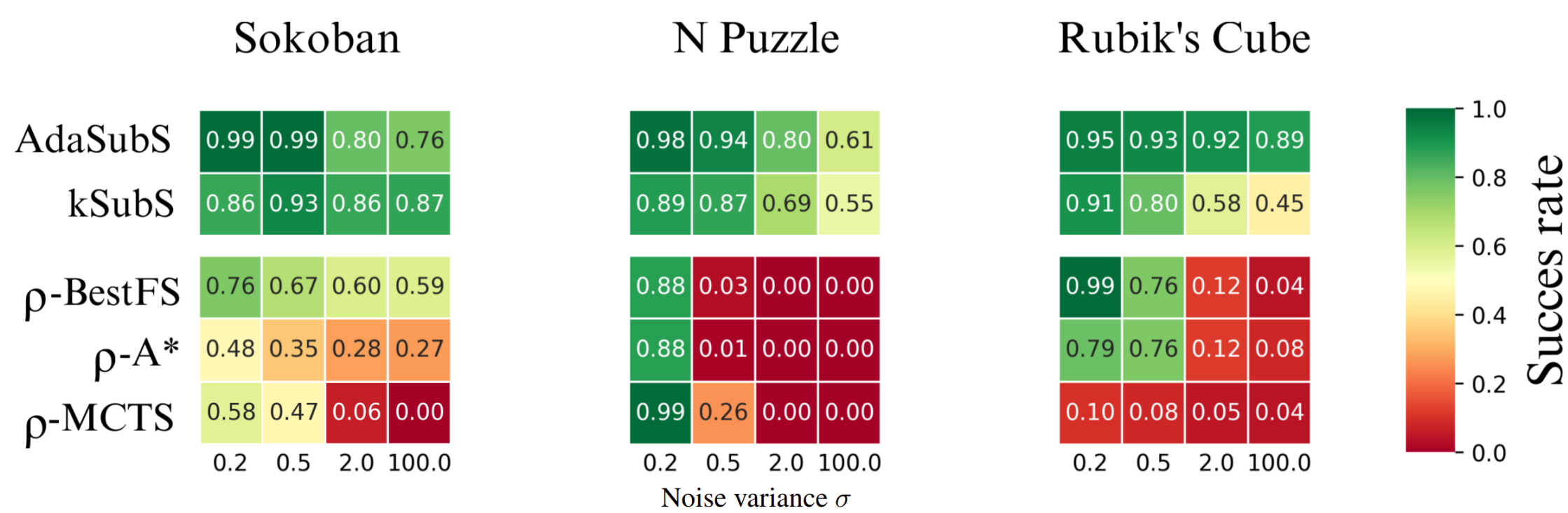

Figure 7: Success rate of low-level and subgoal methods as the approximation errors of the value function increase. $\sigma = 100$ results in completely random value estimates.

While $\rho$-BestFS is able to solve nearly all instances under ideal conditions, its performance significantly declines as value function errors increase, even to 0% (see Figure 7). $\rho$-A* and $\rho$-MCTS behave similarly. In contrast, the subgoal methods show remarkable resilience. Particularly AdaSubS, which maintains nearly unchanged success rate, despite high value errors (Q2).

These results align with our findings in Section 5.1, where using diverse training data naturally introduced value estimation errors. As observed by Zawalski et al. (2023), the search process of subgoal methods is guided by subgoal generators, which reduces reliance on the value function. Subgoal generators and the conditional policies connecting subgoals are not directly influenced by the value approximation errors. The value function is used only in high-level nodes, which represent only a fraction of the search tree.

In hierarchical methods, the distance between high-level nodes spans multiple steps, increasing the likelihood that value estimates for subsequent high-level nodes along the solution path will be monotonic (see Figure 8 for an illustrative example), which makes planning more efficient. This supports the claim by Czechowski et al. (2021) that subgoals effectively mitigate the impact of value noise. To further ground that result, we prove the following theorem:

**Theorem 1** (Search advancement formula)**.** *Let $g_k : S \to \mathcal{P}(S)$ be a stochastic $k$-subgoal generator that, given a state $s \in S$ samples a set of $b$ subgoals $\{s_i\}$ such that the distances $d(s_i, s)$ are independent, uniformly distributed in the interval $[-k; k]$. Let $V : S \to \mathbb{R}$ be a value function with approximation error uniformly distributed in the interval $[-\sigma; \sigma]$.*

*Then, after $n$ iterations of search, the expected total progress toward the goal is:*

$$\mathbb{E}_{Adv} = \frac{nb}{4\sigma k} \int_{-k}^{k} x \left( \int_{-\sigma}^{\sigma} \tilde{u}(x+h)^{b-1} \mathrm{d}h \right) \mathrm{d}x, \tag{1}$$

*where $\tilde{u}(x)$ is CDF of the sum of two uniform variables $U(-k,k) + U(-\sigma,\sigma)$. Additionally, if we approximate that sum as $U(-k-\sigma, k+\sigma)$, we get*

$$\mathbb{E}_{Adv} \approx \frac{n\left((k+\sigma)^b(bk^2 + bk\sigma - 2k\sigma - 2\sigma^2) + \sigma^b(2k\sigma + bk\sigma + 2\sigma^2) - k^b(bk^2)\right)}{(b+1)(b+2)k\sigma(k+\sigma)^{b-1}} \tag{2}$$

*Proof.* See Appendix K for the proof. □

Theorem 1 quantifies the expected progress of the search at each step, with Equation 1 giving an exact formula and Equation 2 providing a useful approximation. To compare subgoal methods with low-level

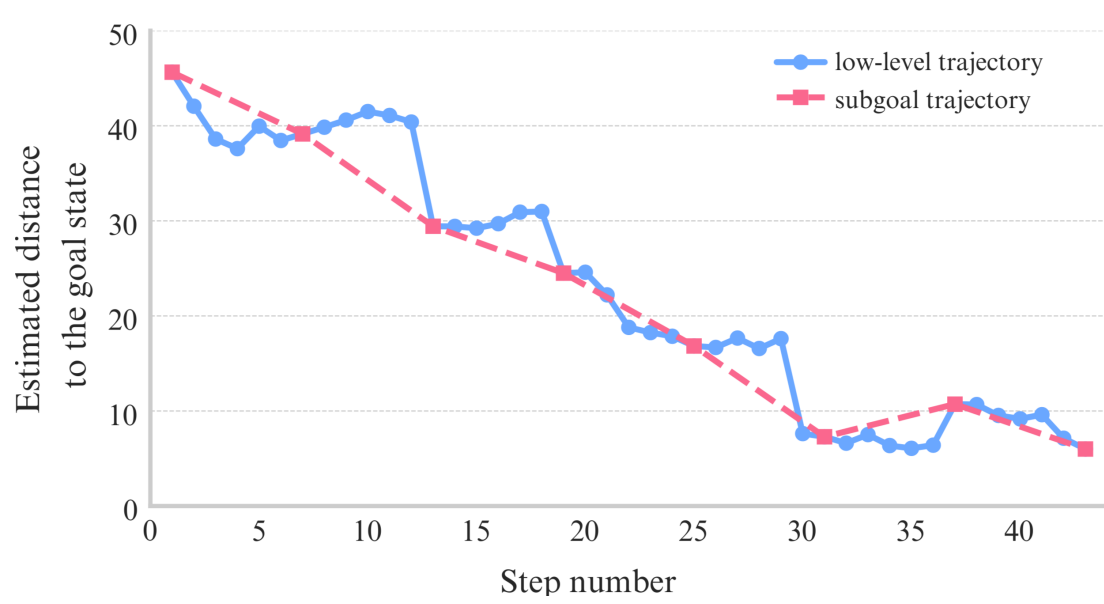

Figure 8: Value estimates along a solving trajectory generated by $\rho$-BestFS. Even small approximation errors cause non-decreasing values, slowing down the search. In contrast, the subgoal path mitigates these errors, leading to mostly monotonic values along the trajectory.

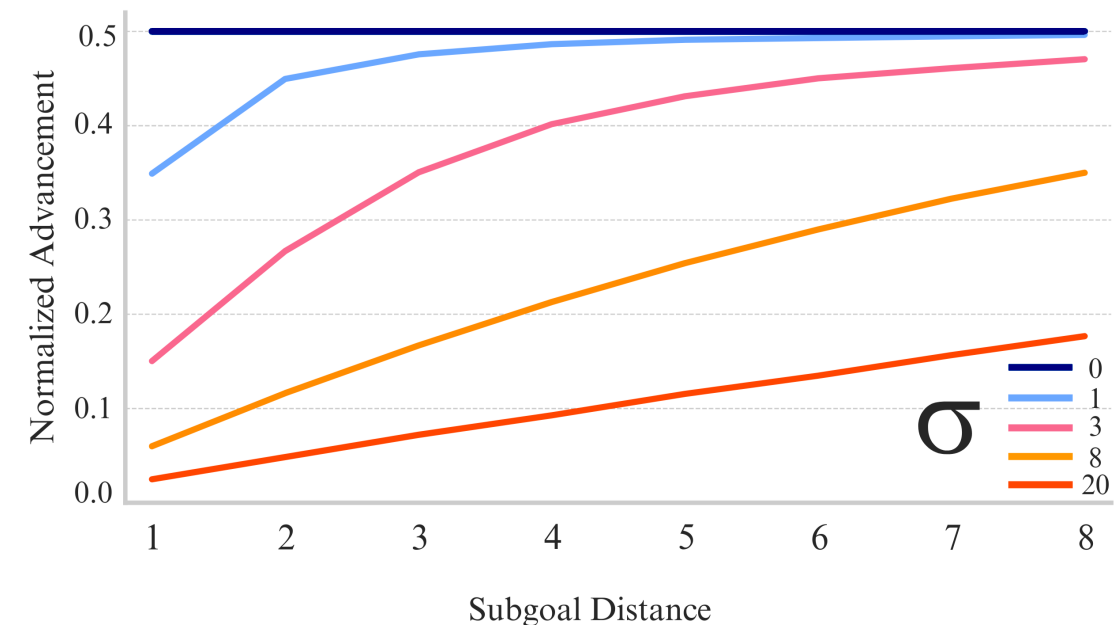

Figure 9: Normalized advancement $\mathbb{E}_{Adv}/k$ for a single search iteration, according to Theorem 1. The value for each subgoal is divided by its length to represent the advancement per atomic action for easier comparison.

methods in theory, under different levels of value approximation error, we model low-level search by setting $k = 1$, which represents a single action. Figure 9 shows the expected search progress with a branching factor of $b = 3$, normalized by the number of actions leading to a subgoal.

When value estimates are perfect (i.e., $\sigma = 0$), both subgoal and low-level searches perform similarly. However, as value approximation errors increase, subgoal methods become significantly more resilient. At high noise levels ($\sigma = 20$), single-step searches make very little progress, advancing only 0.025 per action. In contrast, subgoals of length 8 achieve much greater progress – 1.4 for the entire subgoal, which is 0.175 per action. This 7-fold increase in theoretical efficiency explains why subgoal methods outperform low-level methods in our experiments.

Further analysis of these experiments can be found in Appendix B.2.

**Takeaway** *Subgoal methods successfully handle value approximation errors. Thus, they should be used when estimating the value is hard, for instance, when learning from diverse and suboptimal demonstrations (Q2).*

### 5.3 Subgoal Methods Handle Complex Action Spaces

In environments with large action spaces, search methods often struggle due to the exponential increase in the number of choices (Sutton & Barto, 1998). As shown in Figure 10, subgoal methods demonstrate a clear advantage over low-level search methods in the INT environment (Wu et al., 2021), a benchmark on proving mathematical inequalities (Q1). The INT environment is particularly challenging because of its highly complex observation and action spaces, making it the most difficult benchmark among those used in (Czechowski et al., 2021; Zawalski et al., 2023; Kujanpää et al., 2023a;b).

Given a complex action space, each node expansion in low-level methods involves executing many similar actions, limiting their ability to efficiently search through the space. In contrast, subgoal methods compute actions only to connect subgoals, which is a much simpler task. This targeted approach reduces the negative impact of a large action space, allowing subgoal methods to maintain their efficiency even as the action space grows (Q2).

To confirm this explanation, we conducted experiments on a modified version of the Rubik's Cube, where the action space was artificially inflated by giving the agent access to 100 copies of each action. As shown in Figure 11, this simple modification drastically reduces the success rates of all low-level methods, even below 35%. In contrast, subgoal methods remain largely unaffected, performing similarly to the standard setup. We can explain that result with the following theorem:

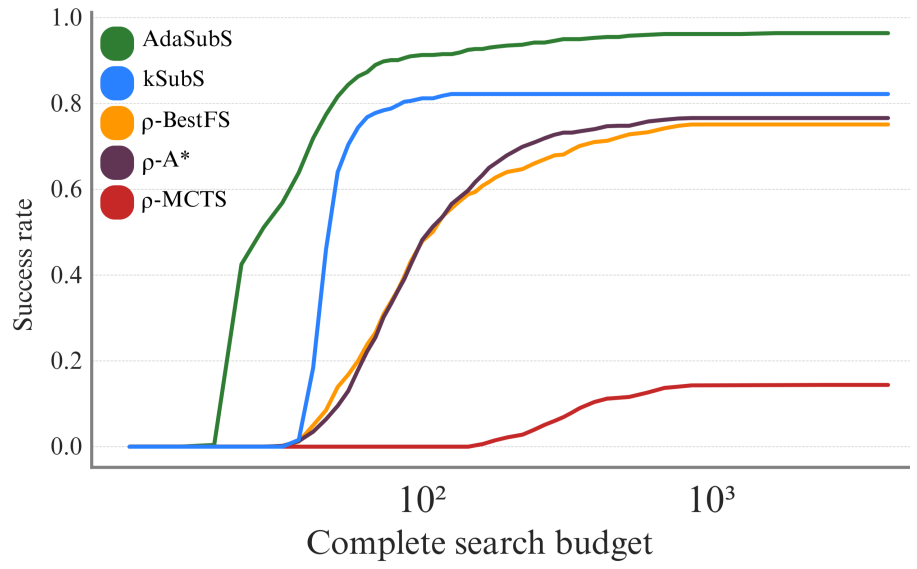

Figure 10: Solving INT. Components are trained on randomly generated proofs.

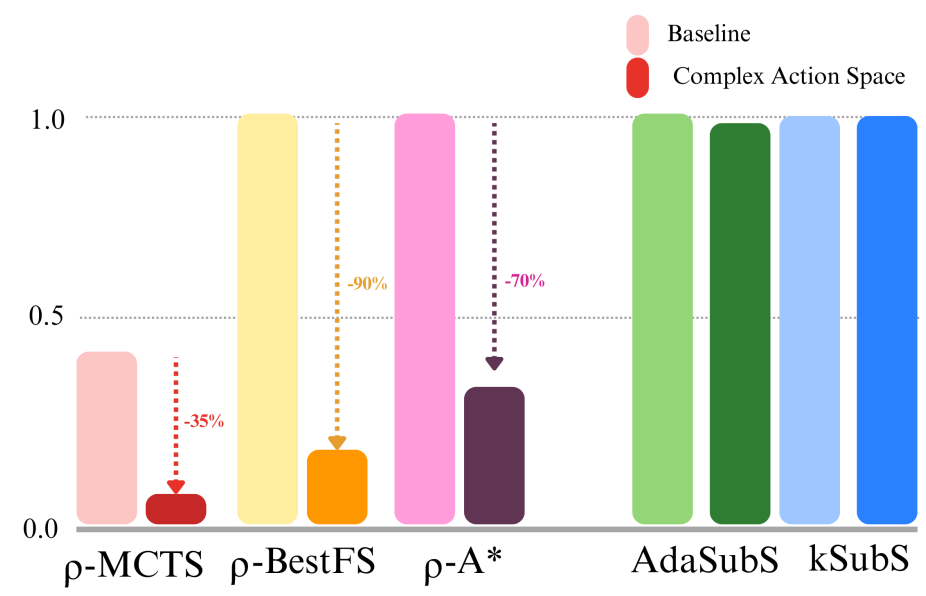

Figure 11: Solving the Rubik's cube with expanded action space, compared with the standard setup. Components are trained on reverse random shuffles.

**Theorem 2** (Densification of the action space)**.** *Fix any state $s$ from the state space $S$. Let $f : A \to [0, 1]$ be the action distribution induced by the data-collecting policy for the state $s$. Assume that $f$ is continuous and has a unique maximum.*

*For clarity, assume $A = [0, 1]$. Consider a sequence of increasingly dense discrete action spaces $A_n := \{i/n\}_{i=0}^{n} \subset A$. Let $\rho_n : S \times A_n \to [0, 1]$ be a family of policies that learn the distribution $f|_{A_n}$ over actions, with uniform approximation error $U(-E, E)$, where $E \in \mathbb{R}_+$. Let $r_n$ be the range of the top $K$ actions according to the probabilities estimated by $\rho_n$. Then*

$$\lim_{n\to\infty} \mathbb{E}[r_n] = 0.$$

*Proof.* See Appendix L □

Intuitively, this theorem states that as the action space become more dense and complex, the actions sampled for search become increasingly less diverse, which strongly impedes successful planning. Note that this analysis is strictly more general than the last experiment, where we simply copied the available actions. Further analysis of the experiments involving large action spaces is provided in Appendix B.3.

**Takeaway** *When facing a problem with a complex action space, subgoal methods should outperform low-level search (Q2).*

### 5.4 Subgoal Methods Avoid Dead Ends

Once an agent encounters a dead end, reaching the goal becomes impossible, leading to wasted computational effort. Our results, presented in Figure 5.4, show that subgoal methods tend to enter dead ends less often than low-level methods. Using longer subgoals improves the ability to bypass those areas.

Among low-level methods, $\rho$-A* performs the best at minimizing dead ends rate, as its node selection regularizes values by depth in the search tree, preventing it from over-committing to dead ends. However, even $\rho$-A* is outperformed by subgoal methods, which rely on greedy value estimates and subgoals.

| Search algorithm | Dead ends rate |
|---|---|
| $\rho$-MCTS | 22.0% |
| $\rho$-BestFS | 18.5% |
| $\rho$-A* | 13.7% |
| kSubS (4 steps) | 12.7% |
| kSubS (8 steps) | 10.0% |
| AdaSubS | 8.86% |

Figure 12: Fraction of dead ends encountered during search between hierarchical and low-level methods in Sokoban.

Deciding whether a state is a dead end can be NP-hard. Hence, it is much harder for the value function to penalize dead ends compared to the policy, which only ranks the available actions and does not have to identify dead ends (Feng et al., 2022). Furthermore, demonstrations used for imitation learning lead to the goal state, hence they contain no dead ends. Therefore the value function trained this way is never directly instructed to penalize dead ends. At the same time, during training of the policy the actions leading to dead ends are never reinforced. Our experiments show that hierarchical search relies much less on the value guidance compared to low-level search (Section 5.2), which further supports our conclusions. For a more detailed analysis, see Appendix B.4.

**Takeaway** *Subgoal Methods Are More Effective at Avoiding Dead Ends Compared to Low-Level Search (Q2).*

### 5.5 Subgoal Methods Generalize Out-Of-Distribution

Planners that can generalize to out-of-distribution (OOD) instances are essential for robust decision-making (Kirk et al., 2023; Shen et al., 2021). We tested two types of generalization in the Sokoban environment: by significantly changing the layout of the board and by using extremely difficult boards from the DeepMind dataset (Guez et al., 2018) (see Figure 13 for examples).

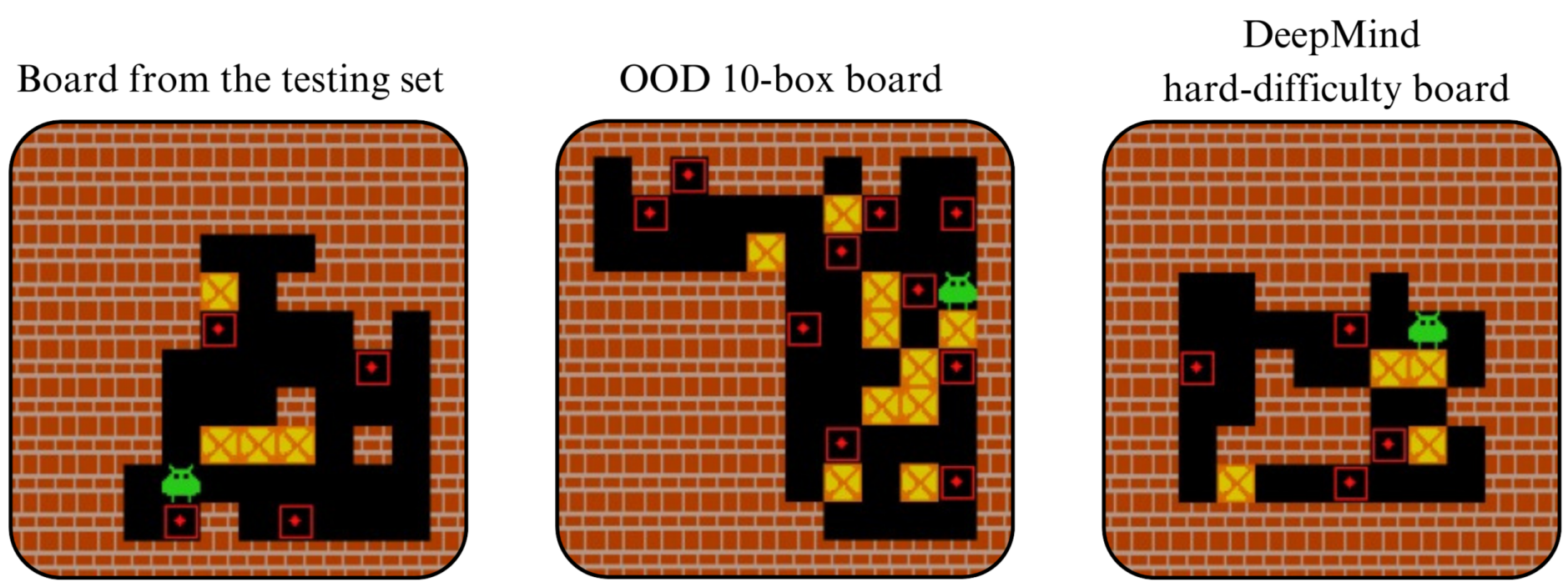

Figure 13: Examples of Sokoban boards used in OOD experiments

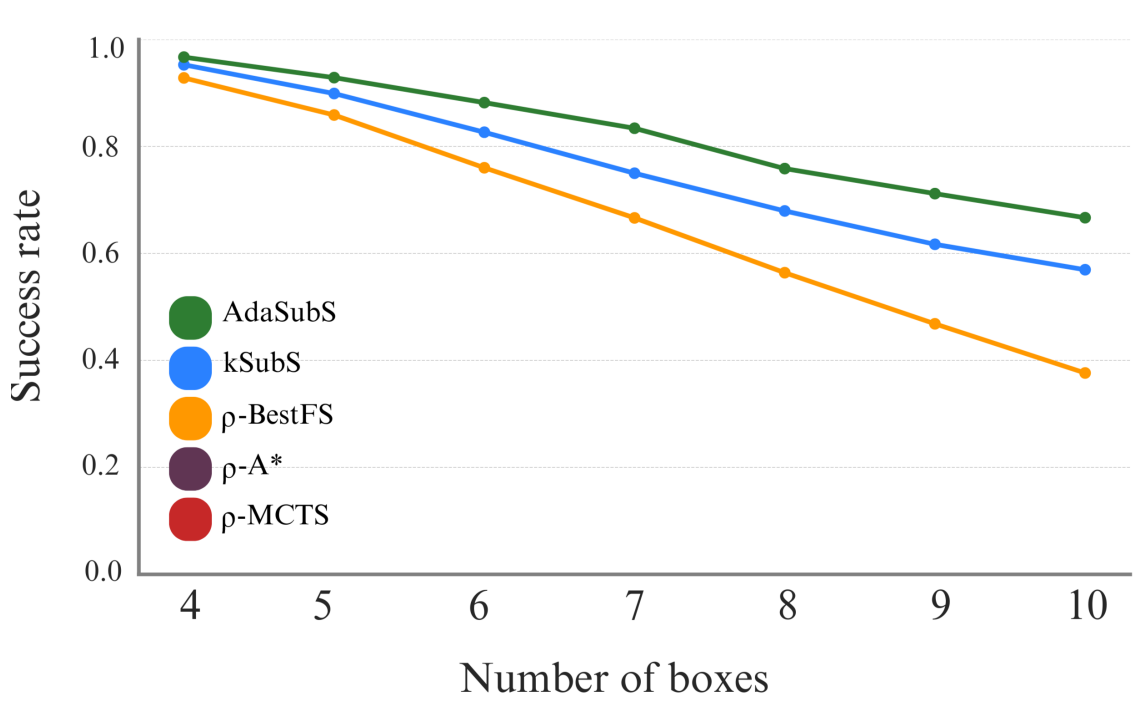

Figure 14: Averaged OOD results on Sokoban boards with OOD layouts. These instances were generated by systematically varying all parameters of the instance generator.

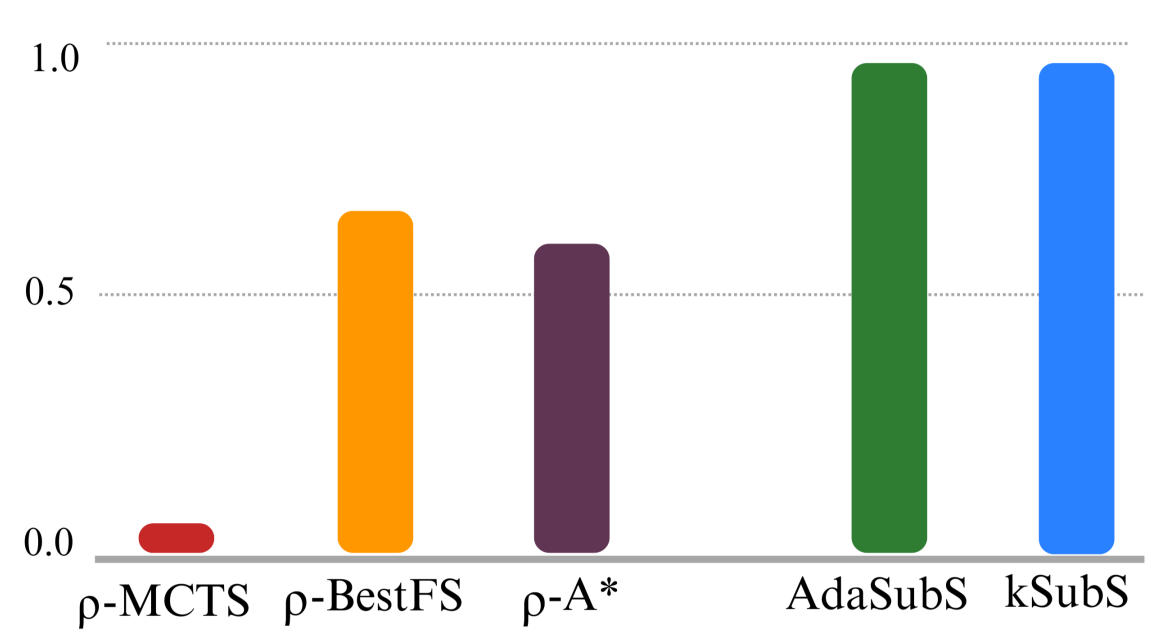

Figure 15: Performance on DeepMind extra hard boards.

In both cases, subgoal methods show better performance than low-level methods, with the gap increasing as the distribution shift become more visible (see Figures 14-15). However, we found that kSubS, when using twice longer subgoals, collapses in OOD evaluations, despite outperforming $\rho$-BestFS and other low-level

methods on in-distribution tasks. As the subgoal distance increases, predicting the distant future becomes more challenging, making it less likely for the generated subgoals to be valid and reachable, especially in OOD tasks. In contrast, low-level methods avoid this issue, as selecting an action from a limited set always results in a valid move. Thus, while subgoal methods can be effective in OOD scenarios, excessively long subgoals can degrade performance (Q2).

When evaluated on extremely challenging instances (see Figure 1) introduced by (Guez et al., 2018), all methods required a significantly higher search budget but maintained the same performance order as in the previous experiment (Q1). Solving these instances requires more advanced strategies than those learned during training. Subgoal methods are better equipped to handle this increased complexity because selecting subgoals is closely related to choosing a broader strategy because of their longer horizon. In contrast, low-level methods must assess each individual action, which limits their ability to foresee the long-term consequences of their choices.

**Takeaway** *Subgoal methods can scale better than low-level methods on OOD instances, provided the subgoals are not too long (Q2).*

## 6 Discussion of the Results and Future Directions

We identified several features that facilitate the performance of subgoal methods; however, this list is not exhaustive. Since our study is primarily empirical, it is difficult to make truly universal claims. This highlights the need for further research, including the analysis of additional subgoal-based and low-level algorithms, as well as a broader range of environments, such as the Traveling Salesman Problem and Maximum Independent Set. Although most of our conclusions were confirmed across multiple settings, extending the evaluation to more domains would further strengthen their validity. Furthermore, extending the analysis to include classical planning methods without learned components is also an interesting direction to explore.

Empirical results suggest general trends, but Theorems 1-2 offer theoretical support for key findings. These theorems, which apply to the general class of hierarchical methods described in Section 4, reinforce the broader relevance of our results. Additional findings from our study may also inspire further theoretical investigation.

We advocate for the use of a complete search budget as a more meaningful metric than alternatives such as the number of high-level nodes or wall-clock time. Nevertheless, for completeness, we report runtime comparisons in Appendix G. Developing a flexible, low-variance, and reproducible evaluation framework based on wall-clock time remains an important direction for future work.

Our study has broader implications for other complex domains. For example, advancements in robotics often face significant challenges due to limited data, leading many methods to rely on collective datasets like Open X-Embodiment (Collaboration et al., 2023). As shown in our experiments, hierarchical search methods benefit substantially from training on diverse expert data (Section 5.1). Furthermore, the data bottleneck increases the need for the models to generalize to out-of-distribution scenes and tasks, which is also an advantage of hierarchical methods (Section 5.5). Finally, an essential aspect of robotics involves preventing the robot from becoming stuck or losing the manipulated object, events that can be seen as dead-end scenarios (Section 5.4). Successful applications of hierarchical methods in robotics include models such as SuSIE (Black et al., 2024) and HIQL (Park et al., 2023).

Additionally, our experiments indicate that hierarchical methods scale well in long-horizon tasks, as evidenced by their performance in the N-Puzzle and the Rubik's Cube (using Beginner demonstrations), where the average sequence of steps often exceeds 200. Interestingly, while low-level methods can still perform well in these scenarios, we observed that they tend to be much more sensitive to hyperparameter tuning.

It is important to note that we do not claim hierarchical methods are universally superior to low-level approaches in all complex domains. Instead, the properties highlighted in our analysis suggest cases where they should be considered.

## 7 Conclusions

We conducted a thorough comparison of hierarchical and low-level search methods for combinatorial tasks. Our experiments provides empirical and some theoretical evidence that hierarchical approaches should be preferred in environments where value estimation is challenging and learned estimates face significant uncertainty, particularly when learning from diverse suboptimal data. Furthermore, subgoal methods demonstrate better scalability in complex action spaces and are more effective at avoiding dead ends than low-level methods. Thus, in environments characterized by those properties, it is advisable to consider subgoal methods as an alternative to low-level search.

While we use Subgoal Search as a representative hierarchical method in our experiments, our analysis is framed from the broader perspective of hierarchical methods, as introduced in Section 4. Notably, Theorems 1 and 2, as well as the general properties illustrated in Figures 8 and 11, apply to a wide class of subgoal methods, not just to the specific implementations used for our experiments.

Based on our results, we propose guidelines for future research in this area. According to our experiments, the best-performing low-level search was usually $\rho$-BestFS with a confidence threshold (see Appendix F). Although it is rather sensitive to the threshold value, which has to be optimized for each domain separately, we advocate using this simple method as a standard baseline for further research in hierarchical search. Our guidelines are comprehensively discussed in Appendix J.

Additionally, we identified easy-to-overlook mistakes in reporting the results that may lead to misleading conclusions. Most importantly, the reported *complete search budget* of hierarchical methods must include all the visited states and not only the high-level nodes as used in some prior works.

## 8 Reproducibility Statement

The code used to run all our experiments is available at `https://github.com/subgoalsearchmatters/what-matters-in-hierarchical-search`. We also link there datasets used for training our models. Hence, all our results are fully reproducible.

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

# Appendix

## Table of Contents

## A Environments

**Sokoban** Sokoban is a classic puzzle game where the objective is to push boxes onto target locations within a confined space. It is a popular testing ground for classical planning methods and deep-learning approaches due to its combinatorial complexity and difficulty in finding solutions. Recognized as a PSPACE-hard problem, Sokoban is used to evaluate different computational strategies. Our experiments use 12 × 12 Sokoban boards with four boxes to assess the performance of our proposed models. An illustrative example of a simple Sokoban search tree with a solving path is shown in Figure 16.

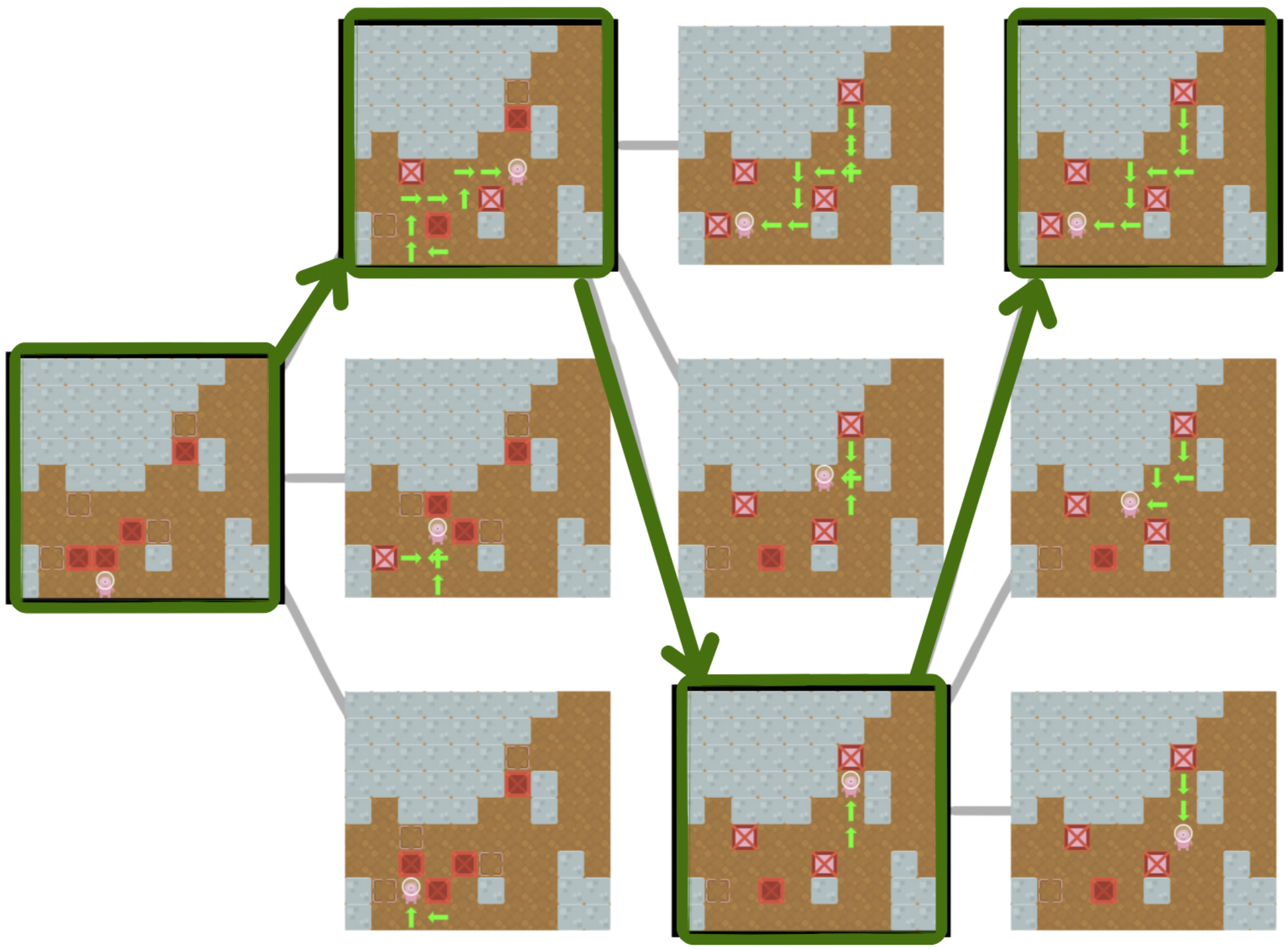

Figure 16: Hierarchical Search applied to solving Sokoban. This tree, depicted in figures, employs bolded green arrows to highlight selected subgoals within a hierarchical search framework earmarked for subsequent exploration. The illustration demonstrates that these intermediate goals exhibit variability in terms of both their spatial distance and the methodology by which a planning algorithm may leverage them.

**Rubik's Cube** The Rubik's Cube, a renowned 3D puzzle, has over $4.3 \times 10^{19}$ possible configurations, highlighting the huge search space and the computational challenge it poses. Recent advancements in solving the Rubik's Cube with neural networks underscore the potential of deep learning methods in navigating complex, high-dimensional puzzles. For the exact representation of the Rubik's Cube state, see Figure 17.

**N-Puzzle** The N-Puzzle, a classic sliding puzzle game, comes in various sizes, including the 3x3 (8-puzzle), 4x4 (15-puzzle), and 5x5 (24-puzzle). The goal is to rearrange a frame of numbered square tiles into a specific pattern, a task that tests the algorithm's ability to plan and execute a sequence of moves efficiently. Figure 18 shows a visualization of a trajectory in 24-puzzle.

**INT** INT (INequality Theorem proving) is an automated theorem-proving benchmark for high school algebraic inequality proofs. (Wu et al., 2021) provides a generator of mathematical inequalities and a proof verification tool. Each action in INT maps to a proof step, which specifies a chosen axiom and its input

| | | |
|---|---|---|
| wbrwyggwwoboybygbryrorroboygrbggbggbwybrooogrywrowywwy | $s_0$ | Initial State |
| wbrwyggggobwybwgbgooyrroyrbrrbwgbygbwybrooogroyrowywwy | $s_1$ | One Action (= single rotation) |
| wbywyoggbobwybwgbgoorrryyryywrggrbbbgybgoorgroyoowrwww | $s_2$ | |
| gyowyoggbwbwwbwwbgoorrryyryywwggbbbyboryogggroyoowrbrr | $s_3$ | |
| ... | | |
| yyyyyyyyybbbbbbrrrrrrrrrgggggggggooooooooobbbwwwwwwwww | $s_{n-1}$ | |
| yyyyyyyyybbbbbbbbbrrrrrrrrrgggggggggooooooooowwwwwwwww | $s_n$ | Solving State |

Figure 17: Example trajectory of Rubik starting from initial state $s_0$ leading to the final solution $s_n$.

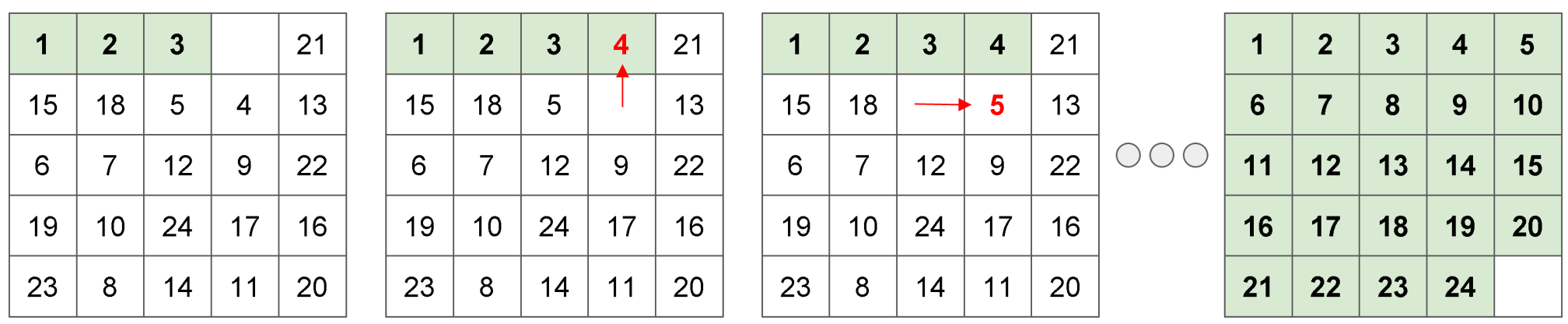

Figure 18: Example trajectory of n-puzzle starting from initial state $s_0$ leading to the final solution $s_n$. Red arrows indicate low-level actions.

entities - which makes action space very high-dimensional, enabling up to a million valid actions at a step. This large action space makes INT a desirable but challenging environment for expanding HRL paradigms to vast action spaces.

We used 25-step proofs for this paper, representing an uplift from 15 considered in (Czechowski et al., 2021; Zawalski et al., 2023) (the latter used longer proofs, but only for evaluating 15-trained models). Each step is an application of an axiom to an axiom-specific number of entities (entities are bracketed or bracketable parts of the theorem's goal).

**Example Theorems for INT environment**

**Theorem 1 Premises:** $((c + c) + d) \geq a$;
$(d + e) \geq 0$;
$((c + c) + f) \geq (0 + a)$;
$(b + g) \geq 0$;

**Goal:** $(((((((c + c) + (c + c)) \cdot 4c) + ((c + c) + d)) + (d + e)) + ((c + c) + f)) + (b + g))$
$\geq ((((0 + a) + 0) + (0 + a)) + 0)$

**Theorem 2 Goal:** $(((0 + b) + c) + a) \geq (0 + (0 + (b + (c + a))))$

**Theorem 3 Premises:** $(a + d) \geq 0$;
$(a + e) \geq (c \cdot c)$;
$(e + f) \geq 0$;
$(c + g) \geq 0$;
$(c + h) \geq (c + g)$;
$(c + i) \geq 0$;

**Goal:** $(((((((c \cdot c) \cdot (a + d)) + (a + e)) \cdot (e + f)) \cdot (c + g)) + (c + h)) \cdot (c + i))$
$\geq ((((((0 \cdot (a + d)) + (c \cdot c)) \cdot (e + f)) \cdot (c + g)) + (c + g)) \cdot (c + i))$

Figure 19: A comprehensive representation of theorems pertaining to goal achievement in mathematical expressions, showcasing the logical structure and underlying premises leading to the formulated goals.

# B Key Factors For Hierarchical Search

According to our experiments, the attributes pivotal for leveraging the advantages of high-level search include:

- learning from diverse data sources,
- hard-to-learn value function,
- complex action space,
- presence of dead ends

In Section 5, we show our main experiments that support our findings. In this appendix, we present an extended analysis of each property.

## B.1 Learning from diverse data sources

Achieving superhuman performance in complex tasks, as demonstrated by AlphaGo Silver et al. (2016), often involves large-scale datasets of demonstrations obtained from agents with varying skill levels and strategies. However, this diversity introduces challenges such as inconsistencies in demonstrations and variations in quality (Fu et al., 2020; Chen et al., 2021; Levine et al., 2020). Widely used datasets like D4RL (Fu et al., 2020), Open X-Embodiment (Collaboration et al., 2023), or Waymo Open Dataset (Sun et al., 2020) reflect this diversity, highlighting the need to address these challenges effectively. We want to answer the question whether such setting is handled better by high-level or low-level search algorithms.

**Experiment setup** For this analysis, we focus on the Rubik's cube environment. We collected a dataset of 500 000 trajectories, computed with four different solvers for the Rubik's cube:

- Beginner – the simplest human-oriented solving algorithm. It aims to order the cube layer by layer with a few primitive tactics. Because of that the solutions are structured, but also very long (typically between 150 and 200 moves).
- CFOP – an algorithm designed for speedcubers. It is based on the same principle as Beginner, but employs many advanced tactics that make the solutions faster (typically about 100 moves).
- Kociemba – a computational solver that finds near-optimal solutions (usually between 20 and 40 moves) in short time. It is heavily optimized based on the algebraic properties of the Rubik's cube.
- Random – solutions obtained by scrambling an ordered cube with random moves and reversing the trajectory.

Figure 30 shows example solutions generated with each solver. Clearly, the algorithmic solvers (Beginner and CFOP) generate much longer solutions that the other methods. They are also more structured, as they are based on building patterns. The computational solver Kociemba on the other hand go directly towards the solution because its moves are carefully optimized to ensure maximal advantage. Because of that, this dataset represent a truly diverse set of demonstrations.

**Results** As shown in Figure 2, the subgoal methods outperform the low-level methods by a wide margin. While $\rho$-BestFS is comparable on small budgets, it struggles with solving most of the instances. Also, it should be noted that the performance of the subgoal methods changes only slightly compared to training on a single Random solver (Figure 4) while the low-level searches are heavily affected.

**Learned values** To find the sources of that outcome, we checked the values learned by the heuristic function. Because of the diversity introduced by combining the experts, we should expect that the estimates are subject to high uncertainty and possibly high variance.

Figure 20 shows the distribution of the learned heuristic for random fully shuffled cubes. Although most instances can be solved optimally within 20-26 moves, the estimates range from 14 to 90 steps. Furthermore,

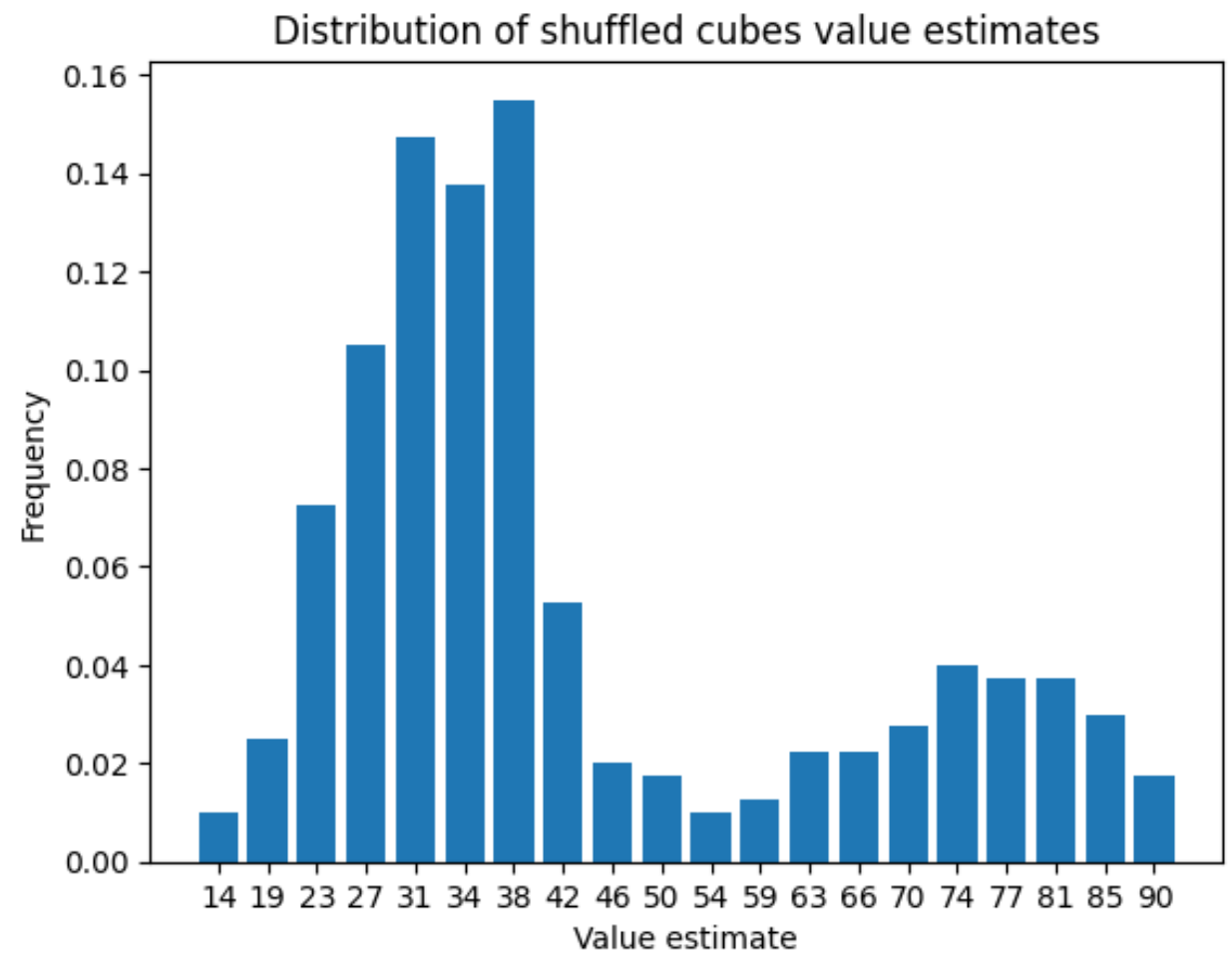

Figure 20: Value distribution for fully scrambled cubes, learned on data coming from diverse experts. The values are rescaled so that the x-axis represent the estimated number of steps to the solution. The values represent the mean of each interval.

the distribution is clearly bimodal – one mode correspond to a typical length of Kociemba solution, the other to CFOP.

Furthermore, Figure 25 shows the distribution of value estimates throughout the solutions for each solver. We observe that for the algorithmic solvers the initial distance is considerably underestimated. After about 20% moves the value network recognizes the pattern of layers built by the solvers and expect a long solution by assigning values close to 100. On the other hand, the values learned for the states visited by the computational solvers start as overestimated, but steadily decrease towards 0.

While it is a reasonable strategy for the value to fit to the provided dataset, it creates a challenge for the search. If a search algorithm aims to imitate Beginner or CFOP, it has to reach the layer pattern, characteristic of those solvers. However, the random states tend to have very low distance estimate, compared to the initial layer patterns. Because of that, for tens of steps the heuristic estimates would be actually increasing, making the reached states less and less probable to expand.

In practice, the low-level searches usually fail to cross this gap. On the other hand, the high-level methods are partially guided by the subgoal generators that ignore the values. The value gap that spans across about 30 steps can be crossed with as few as 5 subgoals of length 6. Because of that both kSubS and AdaSubS can successfully leverage the schematic algorithmic solutions.

To finally confirm that conclusion, we must answer the question whether the performance of low-level searches would increase if they could leverage the algorithmic solutions as well. For that purpose, we trained the components for each method using data only from the Beginner solver. This way we remove the challenge of noisy initial values. As shown in Figure 5, the low-level searches indeed perform much better. BestFS even matches the performance of AdaSubS. That confirms our observation that low-level searchas fail to utilize multimodal data because they rely too much on the value function and seek monotonic slopes.

At the same time we observe that since BestFS and AdaSubS show nearly identical performance on Beginner solutions, it is questionable that hierarchical methods handle long-horizon tasks better, which is a common belief (Nachum et al., 2018; Eysenbach et al., 2019; Chen et al., 2024).

### B.2 Value Approximation Errors

In many practical scenarios, value function estimates are based on either limited data samples or hand-crafted heuristics (Campbell et al., 2002; Mnih et al., 2015; Walke et al., 2023). This often leads to high

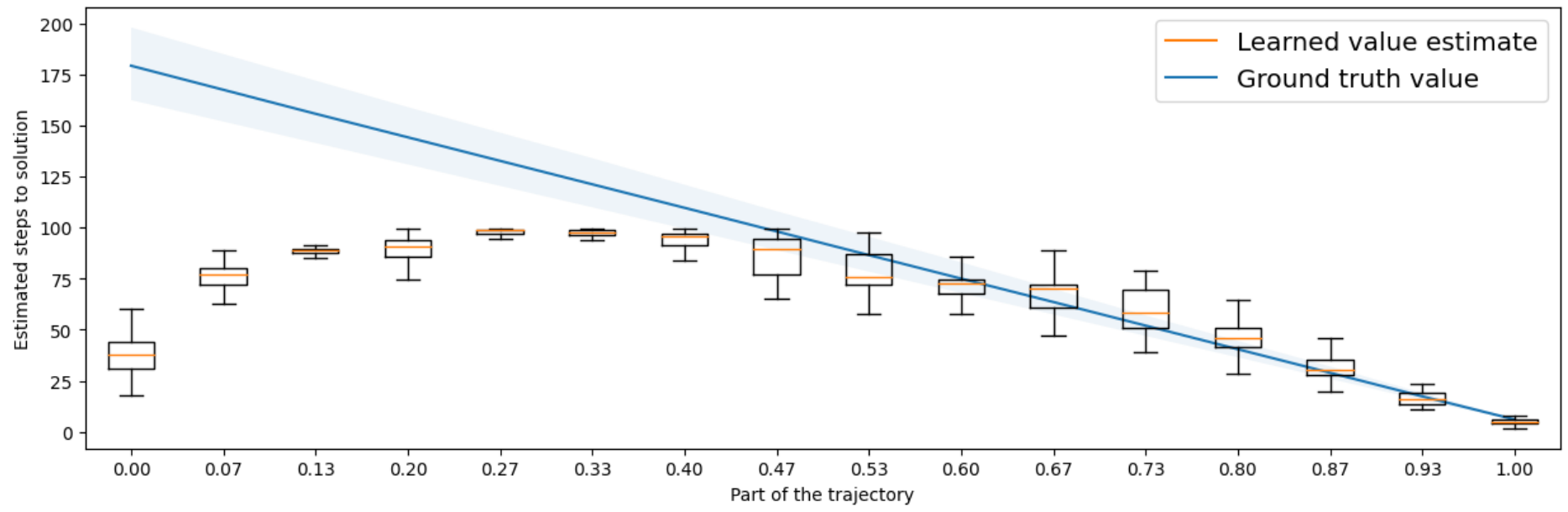

Figure 21: Beginner solver

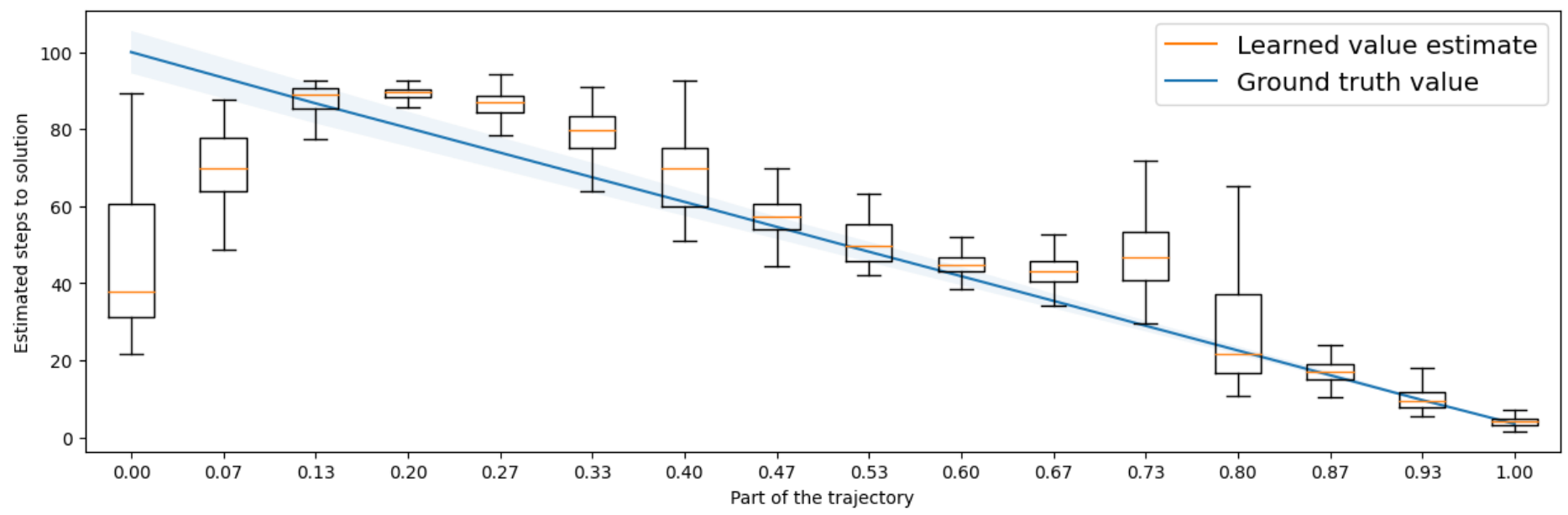

Figure 22: CFOP solver

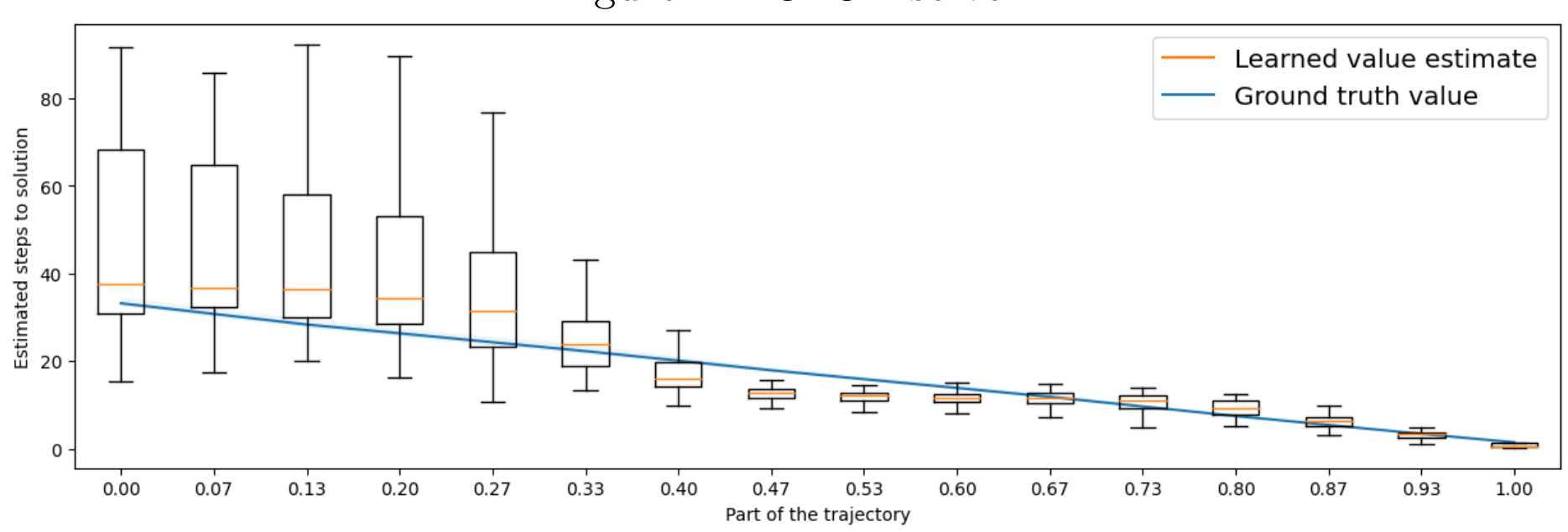

Figure 23: Kociemba solver

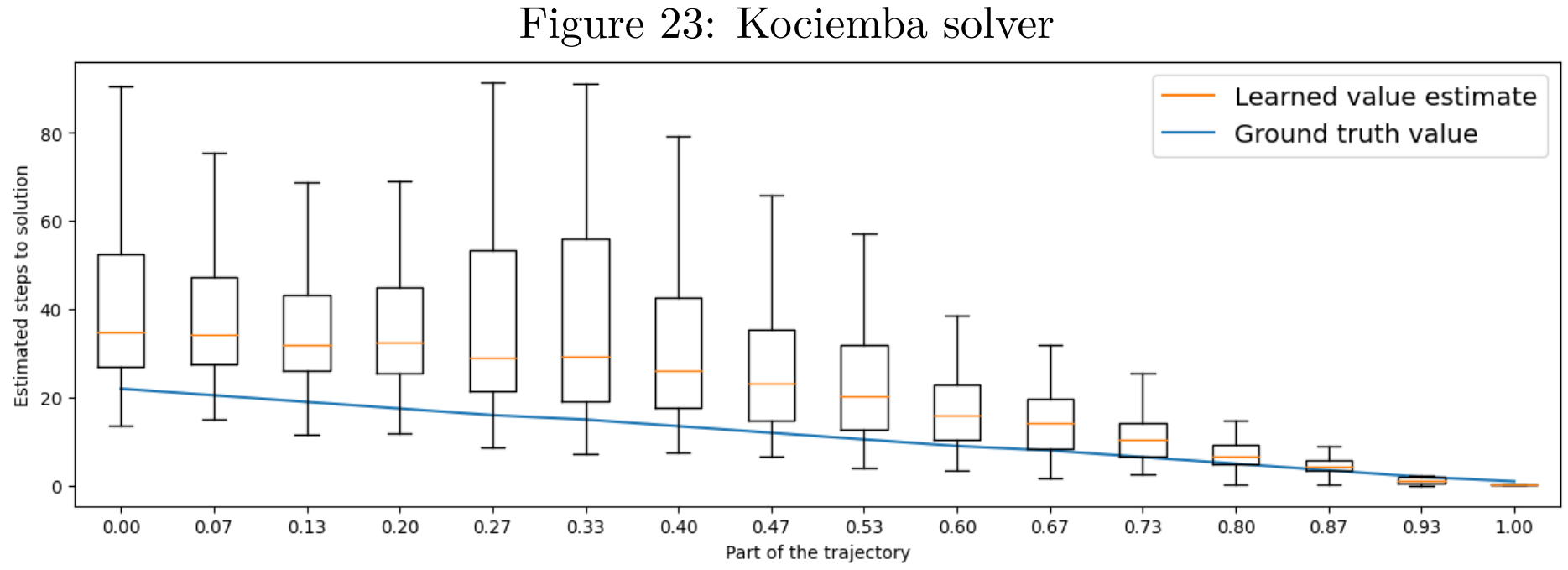

Figure 24: Reversed random 20-move trajectories

Figure 25: The learned value estimates distribution for various solvers. For each plot 100 episodes were solved using the respective solver. The boxes represent the distribution of value estimates for the consecutive points of the solution. The x-axis denotes the relative part of the trajectory (i.e., 0.5 denotes the middle point in each trajectory, regardless of its length). The blue line indicates the true number of steps to the solution.

approximation errors. If search algorithms rely too heavily on these imperfect estimates, they can make poor decisions, especially in large and complex environments where accurate value estimates are even harder to obtain (Collaboration et al., 2023; Vinyals et al., 2019).

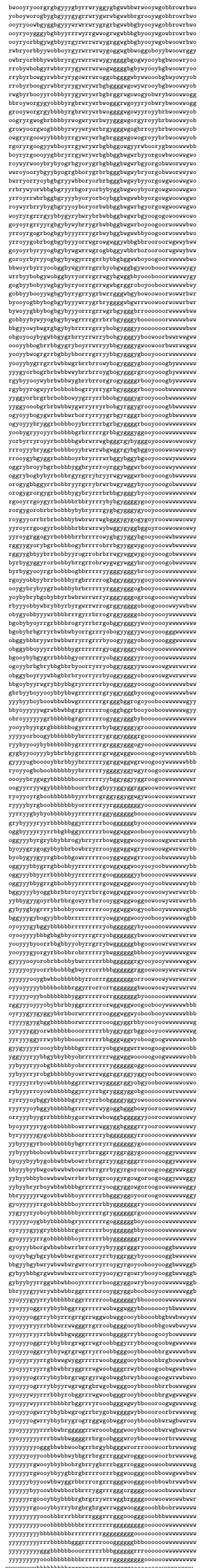

Figure 26: Beginner

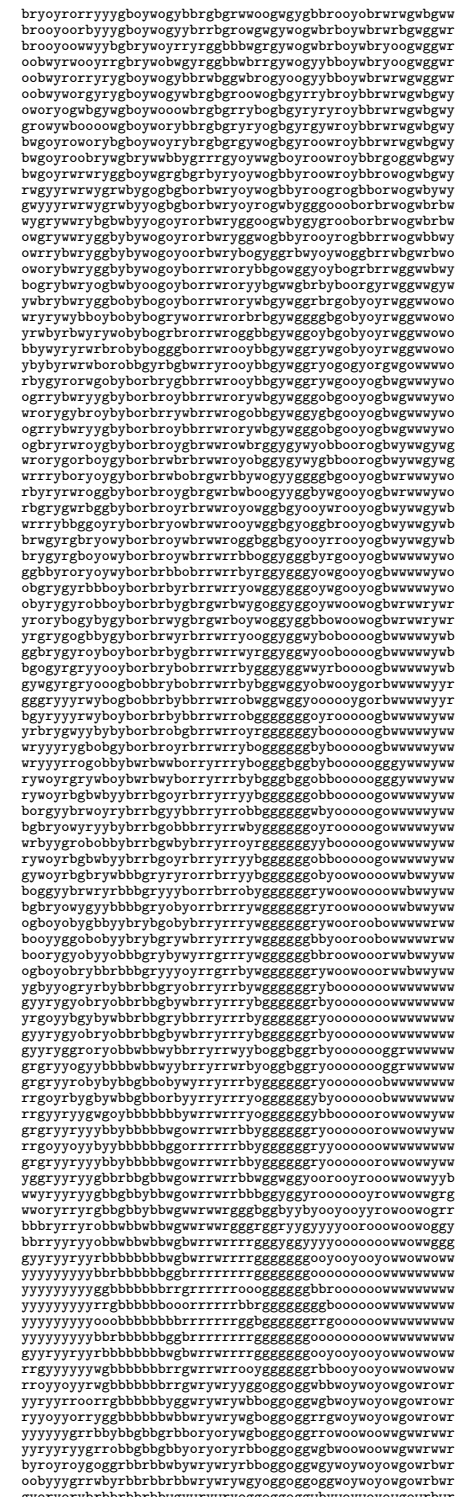

Figure 27: CFOP

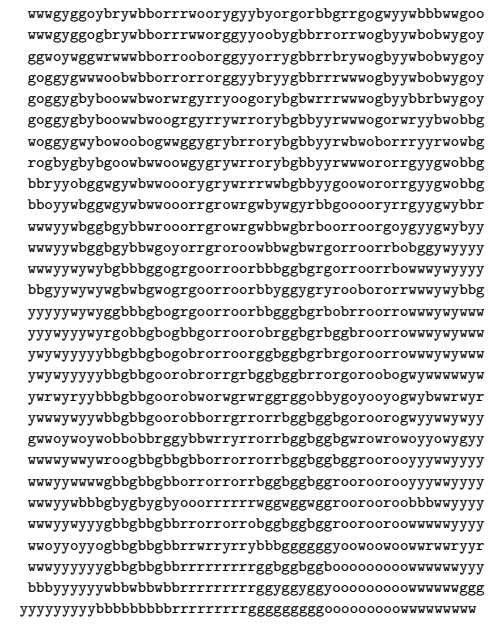

Figure 28: Kociemba

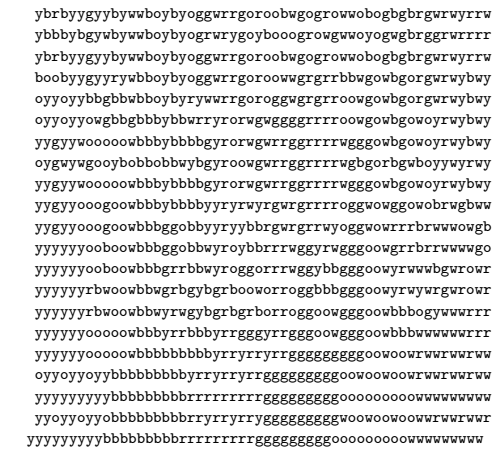

Figure 29: Random

Figure 30: Example solutions computed by each solver. Because the algorithmic solvers typically require over 100 steps, we use a tiny font to display it.

Section B.1 hints that when value estimates are subject to high uncertainty, subgoal methods should outperform low-level searches. To confirm that intuition, we run an experiment in a Rubik's cube, N-Puzzle, and Sokoban environments (Section 5.2). During inference, we add additional noise to the value estimates. That is, whenever a node is added to the search tree and its value estimate equals $\hat{v}$, we add it with the value of $\hat{v} + \mathcal{N}(0, \sigma)$ instead.

Figure 7 shows that as the amount of noise increases, each low-level method gets less and less efficient. On the extreme, when using fully random values ($\sigma = 100$), they struggle to solve any instance.

On the other hand, subgoal methods are much more resilient to noise in the value. Adaptive Subgoal Search is nearly not affected by the presence of noise. kSubS is able to retain as much as $40\% - 90\%$ success rate, even with completely random values.

Observe that the search performed by low-level methods is guided mainly by the value function. Hence, if the computed estimates are subject to high variance, low-level search struggles to make any progress. On the other hand, the subgoal search is guided both by the value function and the subgoal generator. Both the subgoal generator and the conditional policy that connects subgoals do not depend on the values. Hence, the value function is used only in the high-level nodes, which is only a fraction of the search tree.

An extreme case of that behavior is demonstrated by Adaptive Subgoal Search. Because in our configuration each generator outputs a single subgoal, the value is nearly not used at all for search. Only when the search is stuck, the secondary generators select the highest-ranked node to expand, which in this case is simply a random node of the tree. To summarize, given random value estimates, AdaSubS reduces to the following strategy:

1. Start from the root node,
2. Move from the current node to the subgoal until possible,
3. If the search is stuck, expand a random node in the search tree with a secondary generator and return to (2).

The experiments show that this simple strategy is surprisingly competitive to the greedy best-first approach, even without noise. Interestingly, it could be implemented in low-level search as well. We leave that promising experiment for future work.

### B.3 Complex Action Spaces

In environments with large action spaces, search methods often struggle due to the exponential increase in the number of choices at each decision point (Sutton & Barto, 1998). This complexity makes it difficult to efficiently identify optimal actions, slowing down decision-making and exploration (Dulac-Arnold et al., 2015; Silver et al., 2016).

The primary difference between low-level methods and subgoal methods is that the former predicts the next action, and the latter – the next state. In many environments, the action space is as simple as a few bits, allowing for iterating over all possible actions, and sampling them. At the same time, states may be considerably larger, up to the extreme of image observations. However, in some environments, the action space is comparable to the state space, or even more complex. A classic example is the AntMaze environment, in which actions are 8-dimensional, while the goal space is only 2-dimensional (Fu et al., 2020).

Among the combinatorial reasoning environments we consider, INT has the most complex action space. In INT, actions correspond to proof steps and are represented as the chosen axiom, specification of its input entities, and the required premises (Wu et al., 2021). Thus, the complexity of the action is at least comparable to the states. Moreover, solving the INT inequalities is based on constant simplification of the given expression, so the state is getting even smaller with each step.

Our experiments, shown in Figure 10, clearly confirm the advantage of using subgoal methods in the INT environment. To further verify the source of that advantage, we conducted another experiment, in a modified Rubik's cube environment. Recall that the experiment presented in Section 5.1 shows that subgoals offer no significant advantage in the *original* Rubik's cube (with a single data source). Now, we want to check whether the outcome would be different if the action space were more complex. For that purpose, we extended the action space 100 times. That is, the new action space consists of 1200 possible moves to choose from – 100 copies of each original action.

As shown in Figure 11, the subgoal methods are barely affected by the change, while the low-level searches are unable to exceed 20% success rate. That result confirms our proposition that when facing a complex action space, hierarchical methods offer considerably better performance.

According to our analysis, the primary issue with low-level searches in the augmented Rubik's cube is the lack of diversity of visited states. When for each state there are hundreds of actions that lead to a similar outcome, they are rated similarly by the policy. Hence, all the top actions essentially lead to the same outcome, which strongly limits the branching factor and trivializes the search trees. On the other hand, subgoal methods are not affected because subgoal generation does not depend on the action space. The conditional policy that connects the generated subgoals does not build a search tree, but always follows the single best action. Because of that, subgoal methods maintain their performance, even though the action space is much more complex.

It is also important to note that even though some state spaces may seem complex, the underlying manifold of possible configurations is in fact low-dimensional. For instance, we use 12x12 Sokoban boards, where each square is encoded as one-hot of 7 possible items, so technically the state space is 1008-dimensional, while there are only 4 actions. However, in practice the subgoal is defined by the positions of agent and boxes, which is at most 10-dimensional, hence rather simple to generate.

### B.4 Dead Ends

Dead-end states present a major challenge in decision-making and planning tasks. Once an agent encounters a dead end, reaching the goal becomes impossible, leading to wasted computational effort as the algorithm may continue exploring parts of the search space that do not contribute to solving the problem (Russell & Norvig, 2020). Failing to identify dead-ends may even lead to unsafe behavior (Fatemi et al., 2021; Sutton & Barto, 1998). At the same time, identifying dead-ends is NP-complete in many environments.

Specifically, a dead-end state $s$ is one from which there exists no feasible sequence of actions that leads to the goal state. Figure 31 shows an illustrative example of a dead-end state.

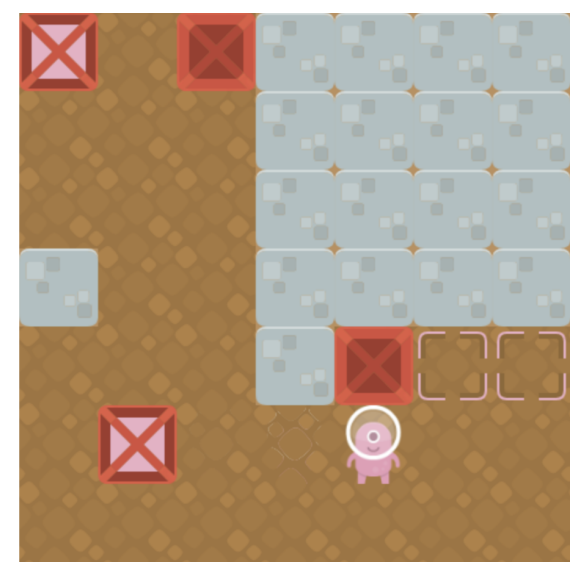

Figure 31: An example dead-end in Sokoban – a box that is pushed to the corner cannot be moved anymore, so the objective is not possible to achieve.

#### B.4.1 Examples Of Dead-Ends In kSubS vs. BestFS

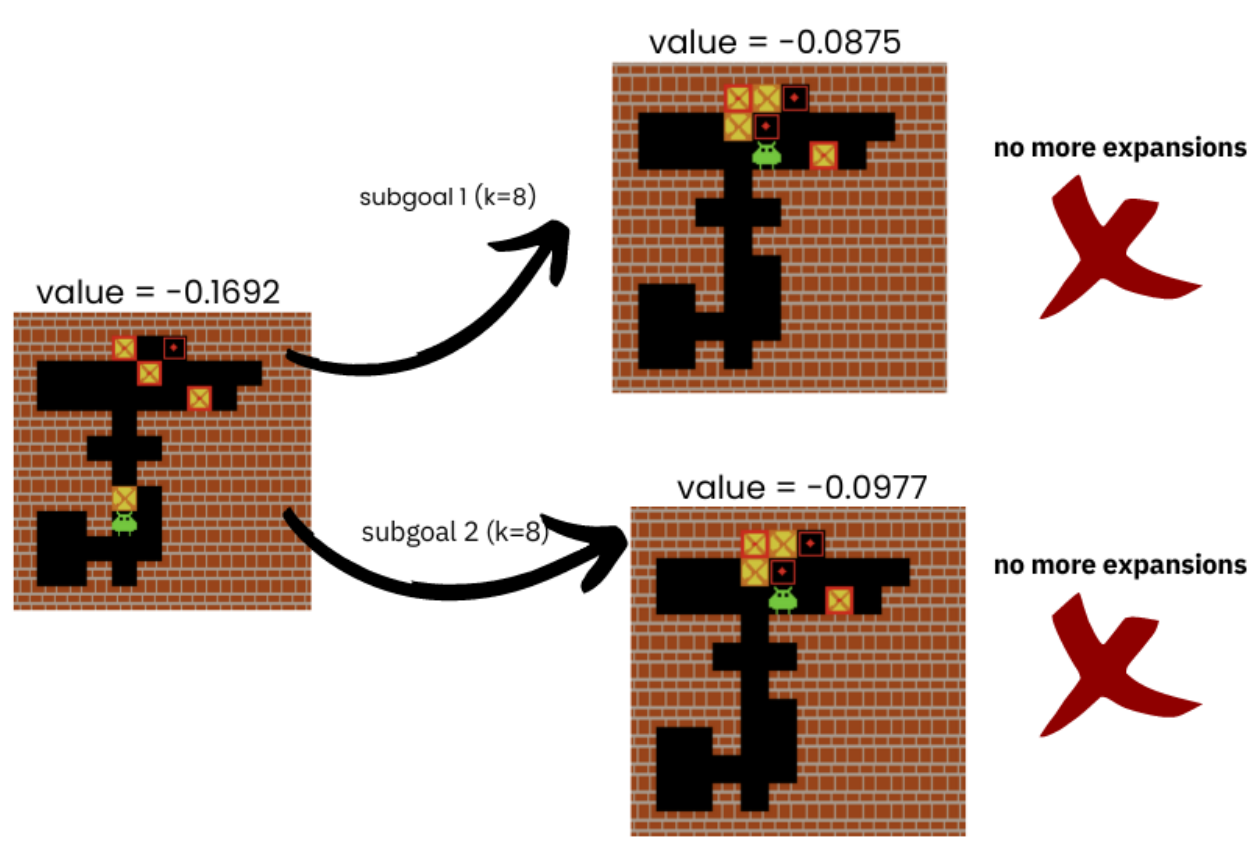

Figure 32: We illustrate a scenario where the kSubS algorithm encounters dead-ends, hindering the search process. The figure shows a case where the algorithm generates two subgoals at an expected distance (k=8), but both lead to dead-ends, wasting a portion of the search budget (18 nodes). As a result, the kSubS algorithm backtracks from this subtree and continues searching elsewhere within the tree.

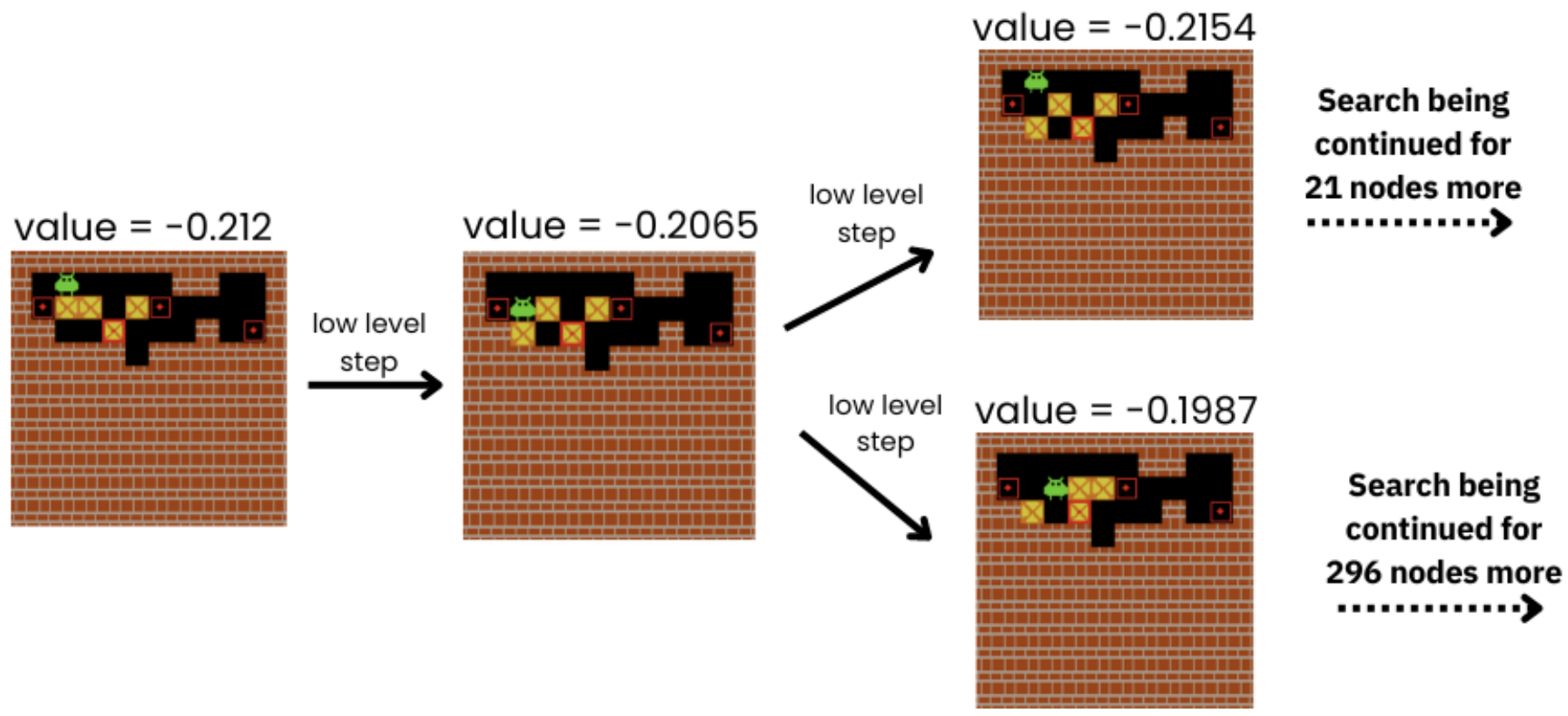

Figure 33: The figure shows BestFS expanding two nodes from a dead-end. This resulted in the exploration of over 300 additional nodes from that state, ultimately failing to find a solution within the given search budget.

In this subsection, we present examples of how each method handles dead-end situations during the search process.

For this presentation, we analyzed 128 search trees initiated from identical starting boards for both algorithms. The kSubS algorithm encountered dead-ends in 3 instances. To resolve these, it navigated through 13 high-level nodes and 105 low-level nodes within the corresponding subtrees. In contrast, the BestFS algorithm encountered dead-ends in 18 instances, requiring the traversal of 4431 nodes. Note that BestFS does not distinguish between high-level and low-level nodes in its search.

Examples of dead-end handling are shown in Figure 32 for kSubS and Figure 33 for BestFS. Observe that in the case showed in Figure 32 expanding the parent node resulted in adding two more dead-ends to the search tree. Because they have higher values, they were immediately expanded. However, the subgoal generator understood that the only way to reach solution is to make an invalid transition of releasing the blocked box. Such subgoals cannot be achieved by the conditional policy, hence no more subgoal was created in that branch. On the other hand, low-level search is unable to propose invalid transitions, so it stays in dead-end until the value estimates are higher than for other branches.

# C Network Architectures & Training Details

| Environment | Hyperparameter | Generator | CLLP | Value | Policy |
|---|---|---|---|---|---|
| INT | learning rate | 0.0001 | 0.0001 | 0.0003 | 0.0001 |
| | learning rate scheduling | linear | linear | linear | linear |
| | warmup steps | 4000 | 4000 | 2000 | 4000 |
| | batch size | 32 | 32 | 128 | 32 |
| | weight decay | 1e-05 | 1e-05 | 1e-05 | 1e-05 |
| | dropout | 0.1 | 0.1 | 0 | 0.1 |
| Rubik's Cube | learning rate | 0.0001 | 0.0005 | 3e-7 | 0.0001 |
| | learning rate scheduling | linear | linear | linear | linear |
| | warmup steps | 5000 | 50000 | 50000 | 1000 |
| | batch size | 512 | 5000 | 5000 | 2048 |
| | weight decay | 0.0001 | 0.001 | 0.00001 | 0.0001 |
| | dropout | 0.1 | 0 | 0 | 0 |
| Sokoban | learning rate | 0.00001 | 0.0001 | 0.0001 | 0.0001 |
| | learning rate scheduling | linear | linear | linear | linear |
| | warmup steps | 2500 | 1000 | 1000 | 1000 |
| | batch size | 512 | 2048 | 2048 | 2048 |
| | weight decay | 0.0001 | 0.0001 | 0.0001 | 0.000001 |
| | dropout | 0 | 0.1 | 0 | 0 |
| N-Puzzle | learning rate | 0.0001 | 0.0001 | 0.0001 | 0.0001 |
| | learning rate scheduling | linear | linear | linear | linear |
| | warmup steps | 5000 | 2000 | 2000 | 2000 |
| | batch size | 4096 | 4096 | 512 | 4096 |
| | weight decay | 0.00001 | 0.00001 | 0.00001 | 0.0001 |
| | dropout | 0.1 | 0 | 0 | 0 |

Table 1: Training-related hyperparameter values

We used BART (Lewis et al., 2020) and BERT (Devlin et al., 2019) architectures from HuggingFace Transformers for all components. Subgoal generators and INT's policies (CLLP and baseline policy) use BART. The remaining policies and value functions use BERT. Following the practice in (Zawalski et al., 2023), we've reduced model size parameters, as detailed in Table 2.

**INT** As states in INT are complex objects, we prefer to use their string representations and avoid mapping arbitrarily generated strings into complex states. Requisite modifications to the component definition are best illustrated analogously to D.1. A generator is redefined as follows:

$$\mathcal{G}_{\text{int}}: \underbrace{\mathcal{S}}_{\text{state to expand}} \rightarrow \underbrace{P(\mathcal{T})}_{\text{set of \textit{proposed} subgoals (in string format)}}$$

and conditional level policy:

$$\mathcal{P}_{\text{int}}: \underbrace{\mathcal{S}}_{\text{current state}} \times \underbrace{\mathcal{T}}_{\text{subgoal \textit{representation}}} \rightarrow \underbrace{\mathcal{A}}_{\text{action}}$$

**Sokoban** Unlike prior work (Zawalski et al., 2023; Czechowski et al., 2021), which used convolutional networks for all components, we work on tokenized representations of Sokoban boards and use BERT/BART architectures instead. This modification did not adversely impact our ability to replicate AdaSubS and kSubS results.

**Training pipeline** We trained our models from scratch using the HuggingFace Transformer pipeline. Detailed training parameters, which varied across environments, can be found in 1.

**Infrastructure** For training, we used a single NVIDIA A100 40GB GPU node, and each component's training took up to 48 hours. Because we used pre-trained trajectories, we did not need to use more than

one core during training. We ran an evaluation using 24-core CPU jobs on Xeon Platinum 8268 nodes with 192GB of memory.

| Environment | Hyperparameter | Generator | CLLP | Value | Policy |
|---|---|---|---|---|---|
| INT | d model | 512 | 512 | - | 512 |
| | decoder layers | 6 | 6 | - | 6 |
| | intermediate size | - | - | 256 | - |
| | encoder attention heads | 8 | 8 | - | 8 |
| | hidden size | - | - | 128 | - |
| | num hidden layers | - | - | 2 | - |
| | decoder ffn dim | 2048 | 2048 | - | 2048 |
| | encoder ffn dim | 2048 | 2048 | - | 2048 |
| | encoder layers | 6 | 6 | - | 6 |
| | decoder attention heads | 8 | 8 | - | 8 |
| Sokoban | d model | 256 | - | - | - |
| | decoder layers | 3 | - | - | - |
| | intermediate size | - | 512 | 128 | 512 |
| | encoder attention heads | 4 | - | - | - |
| | hidden size | - | 512 | 128 | 512 |
| | num hidden layers | - | 6 | 1 | 6 |
| | encoder ffn dim | 2048 | - | - | - |
| | decoder ffn dim | 1024 | - | - | - |
| | encoder layers | 3 | - | - | - |
| | decoder attention heads | 4 | - | - | - |
| N-Puzzle | d model | 64 | - | - | - |
| | decoder layers | 3 | - | - | - |
| | intermediate size | - | 128 | 128 | 256 |
| | encoder attention heads | 4 | - | - | - |
| | hidden size | - | 128 | 128 | 256 |
| | num hidden layers | - | 2 | 1 | 3 |
| | encoder ffn dim | 64 | - | - | - |
| | decoder ffn dim | 64 | - | - | - |
| | encoder layers | 3 | - | - | - |
| | decoder attention heads | 4 | - | - | - |
| Rubik's Cube | d model | 256 | - | - | - |
| | decoder layers | 3 | - | - | - |
| | intermediate size | - | 512 | 128 | 512 |
| | encoder attention heads | 4 | - | - | - |
| | hidden size | - | 512 | 128 | 512 |
| | num hidden layers | - | 2 | 1 | 6 |
| | encoder ffn dim | 2048 | - | - | - |
| | decoder ffn dim | 1024 | - | - | - |
| | encoder layers | 3 | - | - | - |
| | decoder attention heads | 4 | - | - | - |

Table 2: Model-related hyperparameter values

# D Offline Pretraining

Models are pretrained using an offline imitation learning approach. Specifically, given a set of solution trajectories $\{(s_0, s_1, \ldots, s_{n_i})\}_{i=1}^N$ produced by an expert $\mathcal{M}$, or multiple experts $\{\mathcal{M}_j\}_{j=1}^M$ in cases where offline trajectories are collected from multiple experts, the objective is to learn from these trajectories. It is important to note that these trajectories are not required to be optimal; they may include loops or numerous redundant actions. Description of all components can be found in section $D.1$ and supervised training objectives in section $D.2$.

## D.1 Components

During the pretraining phase, models undergo an offline imitation learning process. Specifically, they are trained on a set of solution trajectories $\{(s_0, s_1, \ldots, s_{n_i})\}_{i=1}^N$, which are collected to facilitate the learning of decision-making strategies.

**Generator** The generator component is responsible for generating subgoal propositions upon receiving a state. These propositions are designed to facilitate progress toward the solution by suggesting intermediate steps that direct the search process more efficiently.

$$\mathcal{G} : \underbrace{\mathcal{S}}_{\text{state to expand}} \rightarrow \underbrace{P(\mathcal{S})}_{\text{set of subgoal propositions}}$$

**Conditional Low-Level Policy** The Conditional Low-Level Policy (CLLP) plays a crucial role in node expansion by evaluating each subgoal proposition. For a given current state and a subgoal, the CLLP recommends actions that lead toward achieving the subgoal. A path from the current node to the subgoal is constructed through the iterative execution of these actions. Subgoals reached within a predefined number of steps, $k$, are incorporated into the graph, while those that are not are discarded.

$$\mathcal{P} : \underbrace{\mathcal{S}}_{\text{current state}} \times \underbrace{\mathcal{S}}_{\text{subgoal state}} \rightarrow \underbrace{\mathcal{A}}_{\text{action}}$$

**Value** The value function estimates the distance from a current state to the final solution. This estimation is used to guide the selection and expansion of nodes, influencing the overall search strategy.

$$\mathcal{V} : \underbrace{\mathcal{S}}_{\text{state to evaluate}} \rightarrow \underbrace{\mathbb{R}}_{\text{value of the state}}$$

**Behavioral Cloning Policy** The policy $\Pi_{\text{BC}}$ is a decision-making function that maps the current state to an action. It encapsulates the strategy derived from the learning process, guiding the agent's actions towards achieving the final goal.

$$\Pi_{\text{BC}} : \underbrace{\mathcal{S}}_{\text{current state}} \rightarrow \underbrace{\mathcal{A}}_{\text{action}}$$

## D.2 Supervised Objectives

Each expert trajectory is defined as a sequence of states and corresponding actions $(s_0, a_0), \ldots, (s_{n-1}, a_{n-1}), s_n$ that delineate a path to a solution. The training methodology leverages this data through several key self-supervised imitation mappings:

- A $k$-subgoal generator that maps a state $s_i$ to a future state $s_{i+k}$, simulating the achievement of intermediate goals.
- A value function that estimates the remaining steps to the solution by mapping state $s_i$ to a numerical value $(i - n)$, representing the estimated distance from the goal.
- A policy that maps each state-action pair $(s_i, s_{i+d})$, with $d \leq k$, to the corresponding action $a_i$, thereby guiding the decision-making process towards the solution.

# E Offline Pretraining: Trajectories

## E.1 Rubik's Cube

### E.1.1 Random

To construct a random successful trajectory, we performed 20 random permutations on an initially solved Rubik's Cube and took the reverse of this sequence, replacing each move with its reverse. Such solutions are usually sub-optimal since random moves are not guaranteed to increase the distance from the solution. They can even make loops in the trajectories. However, a cube scrambled with 20 moves is usually close to a random state, so such trajectories give a decent space coverage.

### E.1.2 Beginner, CFOP

*Beginner* and *CFOP* are algorithms commonly used by humans. They solve the cube by ordering the stickers layer by layer. Because of that, the solutions are highly structured and long – usually between 100 and 200 moves. Both algorithms are composed of several subroutines that help building the consecutive layers. Thus, the structure of such solutions highly resembles the subgoal search.

### E.1.3 Kociemba

The *Kociemba two-stage solver* leverages the algebraic structure of the Rubik's Cube. In the first stage, its goal is to enter a specific subgroup. Since that subgroup is much smaller than the whole space, completing the solution may be done efficiently. *Kociemba* finds reasonably short solutions (usually between 20 and 40 moves) and works reasonably fast.

### E.1.4 Size Of Datasets

For training the components on a dataset collected by a single solver, we generate 100 000 trajectories. For the experiment with diverse experts, each solver generates 25 000 trajectories for a total of 100 000.

## E.2 INT

Trajectories are constructed from sequences of axiom applications, similarly to (Zawalski et al., 2023), who followed (Wu et al., 2021). A set of up to 15 (out of 18) axioms is first selected, and then a random axiom order is set and validated. Finally, a proof is converted to a relevant trajectory. Approximately 500,000 trajectories were generated for model pre-training.

We capped the number of axioms at 15 because some pairs of axioms (eg. terminal axioms) cannot be in one trajectory.

## E.3 N-Puzzle

To collect data for N-puzzles, we utilized an algorithm that initially arranges block number 1, followed by block number 2, and so forth, as depicted in Figure 18. The training set comprises approximately $10,000$ trajectories.

## E.4 Sokoban

To collect trajectories for Sokoban, we used a trained MCTS agent that gathered approximately $100,000$ trajectories.

# F Algorithms

## F.1 Best-First Search

**Overview** Best-First Search greedily prioritizes node expansions with the highest heuristic estimates, aiming for paths that likely lead to the goal. While not ensuring optimality, BestFS provides a simple yet efficient strategy for navigating complex search spaces. The high-level pseudocode for BestFS is outlined in Algorithm 1, and the detailed pseudocode is presented in Algorithm 2.

**Algorithm 1** Pseudocode for Best-First Search

**while** has nodes to expand **do**
  Take node $N$ with the highest value
  Select children $n_i$ of $N$
  Compute values $v_i$ for the children
  Add $(n_i, v_i)$ to the search tree
**end while**

**Heuristic** In our implementation, we adhere to the Best-First Search principle by utilizing the learned value function, a common practice in the planning domain (Brunetto & Trunda, 2017; Czechowski et al., 2021; Zawalski et al., 2023; Kujanpää et al., 2023a). It should be noted that in each of our experiments, all the compared algorithms use the same value function network. This way we ensure that the differences come solely from the algorithmic part.

**Selecting children** When expanding a node during search, the standard BestFS algorithm adds all its children. However, in our implementation, we aimed to reduce the search tree size by selecting only the most promising children. We achieve this by sorting the children according to their probability distribution predicted by the policy network. For choosing the final subset of children, we employ two approaches. In the simpler variant, we always select the top $k$ actions. In the second variant, we add top children until their cumulative probability exceeds a fixed threshold $t_{conf}$.

This pruning does not adversely affect the standard algorithm, as nodes are still chosen based on their heuristic values, while the threshold sets a practical limit on the search space. Our results demonstrate that BestFS tends to perform much better with a confidence threshold (Figure 34). However, its performance is highly sensitive to this threshold as it balances exploration and exploitation, illustrating the impact of different confidence thresholds on success rates.

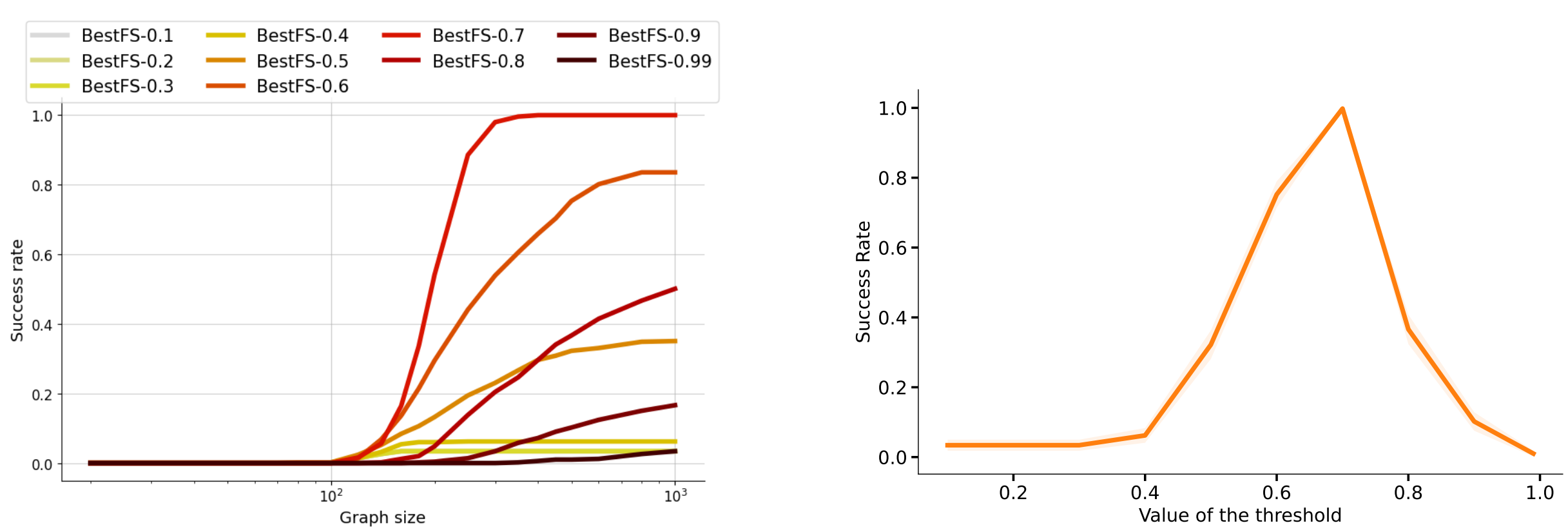

Figure 34: Comparison of success rates for the BestFS algorithm on the Rubik's Cube with various confidence threshold values. BestFS-X represents the BestFS algorithm with the confidence threshold set to X. *Left:* The plot displays the achieved success rate relative to the graph size. *Right:* The plot illustrates the success rate for a budget of 500 nodes.

**Completeness** In the Rubik's Cube environment with random trajectories, the subgoal methods solve more instances than BestFS given a low search budget, but with more resources, BestFS takes the lead (see Figure 4). Also, in other experiments, we may observe that BestFS typically requires higher computational budget to solve the simplest instances, but its performance increases considerably with more resources.

That behavior is related to the fact that the search trees built by hierarchical methods are much sparser because the branching occurs only in the high-level nodes. On the other hand, the low-level algorithms can cover a higher fraction of the space. On the extreme, if we used all the available actions for every expansion, the low-level search would be *guaranteed* to find a solution if one exists. Our mechanism of selecting the actions removes that guarantee. However, at the same time, it drastically improves performance (compare BestFS-0.7 with BestFS-0.99 which is complete), which makes it a much better choice for our study.

We note that the high-level algorithms could be made complete, as proposed in (Kujanpää et al., 2023b; Zawalski et al., 2023). However, to maximize the efficiency we choose to keep the tested algorithms in their original form. The ability to search with sparse trees not only lets the methods advance fast, but also withdraw quickly if the branch does not lead to the solution (is a dead end).

**Hyperparameters** To identify the most suitable solving parameters, we used grid search. Initially we grid over coarse values (namely 0.1, 0.2, 0.3, 0.4,0.5, 0.6, 0.7, 0.8, 0.9, and 0.99). Then we check finer values (with precision of 0.05) around the best-performing threshold. The best-performing thresholds range from 0.6 to 0.85, depending on the environment and the components that are used.

For determining the best number of top actions $k$ for the simpler variant, we simply check every possible number of actions. Usually selecting 2 actions is by far the best choice.

Details regarding hyperparameters of the networks are listed in Appendix D.1.

**Algorithm 2** Complete pseudocode for Best-First Search

```
Require:
  value function network V,
  policy ρ_BFS
  predicate of solution SOLVED

  function SEARCH(s_0)
  T ← ∅ {priority queue}
  T.PUSH((V(s_0), s_0))
  parents ← {}
  seen.ADD(s_0) {seen is a set}

  while 0 < LEN(T) and LEN(seen) < max_budget do
    _, s ← T.EXTRACTMAX() {select node with the highest value}
    actions ← ρ_BFS(s)

    for a in actions do
      s' ← ENVSTEP(s, a)

      if s' in seen then
        continue
      end if

      seen.ADD(s')
      parents[s'] ← s
      T.PUSH((V(s'), s'))

      if SOLVED(s') then
        {solution found}
        return EXTRACTLOWLEVELTRAJECTORY(s', parents)
      end if
    end for
  end while

  return False {solution not found}
```

### F.2 Monte Carlo Tree Search

**Overview** Our Monte Carlo Tree Search (MCTS) solver, designed for a single-player setting, is based on the AlphaZero framework (Silver et al., 2018). The high-level workflow of MCTS is illustrated in Figure 35, and detailed pseudocode is provided in Algorithm 3.

The algorithm's operation consists of four primary stages:

- **Selection**: The most promising node is selected using Polynomial Upper Confidence Trees (PUCT), augmented with an exploration weight to strike a balance between exploiting known strategies and investigating new pathways.
- **Expansion**: The selected node is expanded, generating new child nodes that correspond to prospective future actions. This expansion widens the search tree and enables the exploration of various outcomes.
- **Simulation**: Following the AlphaZero approach (Silver et al., 2018), policy and value networks replace traditional simulations. The policy network suggests favorable moves, while the value network predicts their probability of success, directing the algorithm towards beneficial trajectories.
- **Backpropagation**: The insights derived from the networks are used to update node values, improving future decision-making.

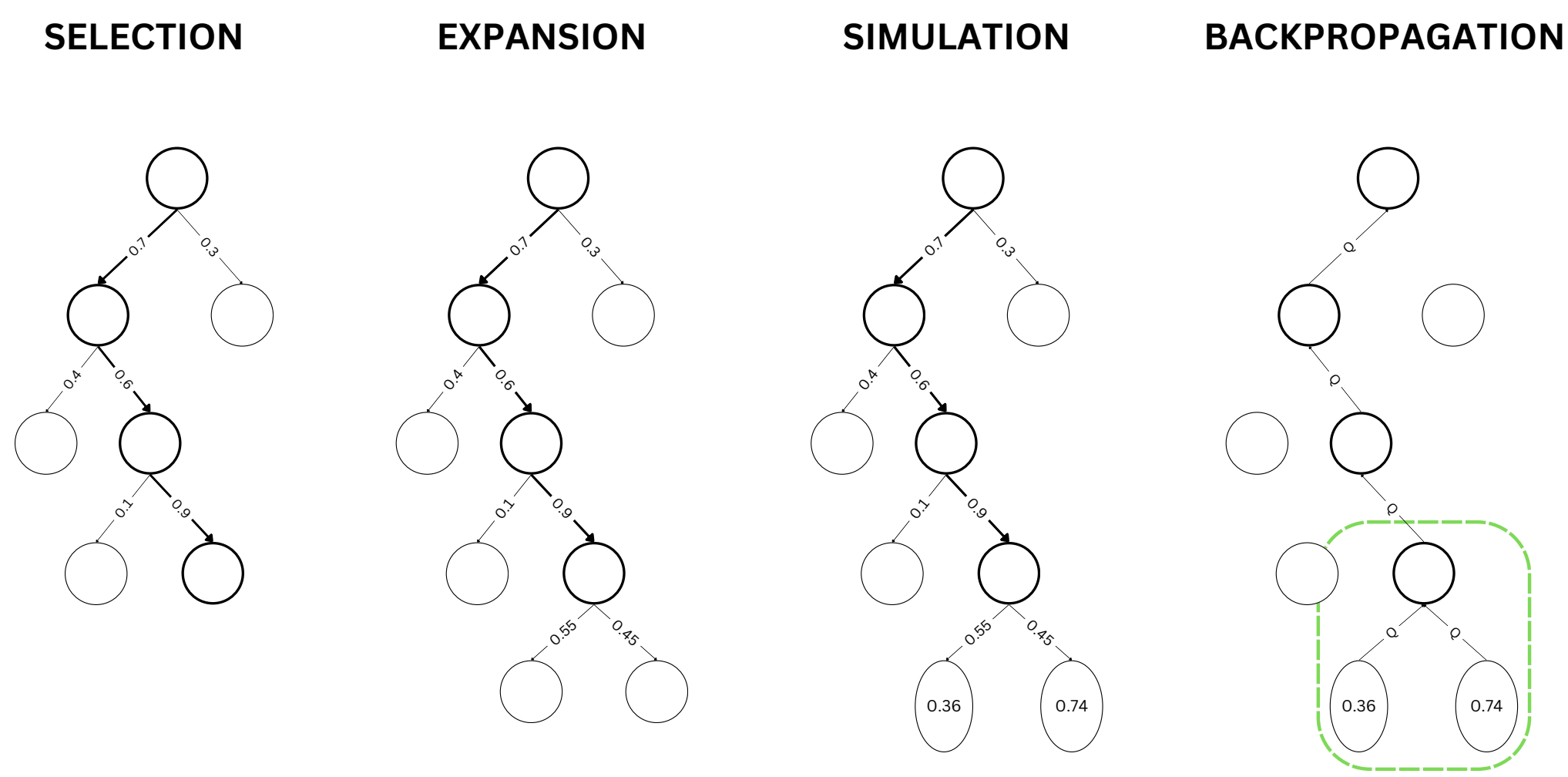

Figure 35: Schematic diagram of the MCTS algorithm in our implementation. Arrows show policy network probabilities and node values are valued network predictions. Q values, calculated via PUCT, integrate these with exploration-exploitation balance.

**Hyperparameters** In the MCTS algorithm, the parameters were set as follows: sampling temperatures were chosen from [0, 0.5, 1]. The number of steps varied between 200 and 1000, and the number of simulations ranged from 5 to 300. The discount factor and exploration weight were consistently set at 1.

**Algorithm 3** MCTS Solver

**Require:**
Number of simulations: $N_s$
Discount factor: $\gamma$
Exploration weight: $c_{\text{puct}}$
Sampling temperature: $\tau$
Value function: $V$
Environment model: $M$
Initial state: *initial_state* from env

**function** SEARCH((*initial_state*))
$root \leftarrow initial_state$
$iteration \leftarrow 0$
**while** $iteration < N_s$ **do**
$node \leftarrow root$
**while** *node* is not a leaf **do**
$node \leftarrow$ SELECTCHILD(*node*), according to PUCT formula
**end while**
$leaf \leftarrow node$
Expand the leaf using the environment model $M$, policy $\pi$, value function $V$, and discount factor $\gamma$
Backpropagate results through the path to update $N, W, Q$
$iteration \leftarrow iteration + 1$
**end while**
$best_child \leftarrow$ Sample child of the *root* according to $\tau$ and $N$
**return** action leading to *best_child*

### F.3 A* Search

**Overview** Like Best-First Search, A* prioritizes the exploration of promising nodes. However, A* strategically guides its search by incorporating both the actual cost to reach a node and a heuristic estimate of the remaining distance to the goal. This way it balances the greedy exploitation and conservative exploration. The high-level pseudocode for A* is outlined in Algorithm 4, and the detailed pseudocode is presented in Algorithm 5.

**Algorithm 4** Pseudocode for A*

```
while has nodes to expand do
    Take node N with the highest value
    Select children n_i of N
    Compute values v_i for the children
    Compute depth d_i for the children
    Add (n_i, λd_i + v_i) to the search tree
end while
```

**Heuristic** A* guidance is achieved through the following cost function:

$$f(node) = \lambda g(node) + h(node)$$

where:

- $g(node)$: The cost to reach *node* from the start state, in our case its depth in the search tree.
- $h(node)$: A heuristic estimate of the cost from *node* to the goal state.
- $\lambda$: A scaling factor balancing the influence of actual cost and heuristic estimate.

For heuristic $h$, we used a value network, like for BestFS (see Appendix F.1). If the heuristic used for A* is *admissible*, i.e. it never overestimates the cost of reaching the goal, A* is guaranteed to find an optimal solution. For instance, if we used $h(node) \equiv 0$, A* would reduce to the Dijkstra algorithm. The heuristic that we learn is not guaranteed to be admissible. Firstly, it estimates the distance according to the demonstrations, which is always an upper bound for the optimal distance. Secondly, the approximation errors introduce additional uncertainty. However, our main focus is on finding any solution, not necessarily an optimal one.

**Selecting children** During the search, A* maintains a priority queue of nodes to be explored. Similarly to BetsFS (Appendix F.1) for reducing the search tree size, we select the most promising children. At each iteration, the node with the lowest $f(node)$ value is selected for expansion. The algorithm proceeds until the goal state is reached or the computational budget is exceeded.

**Hyperparameters** The key parameter for A* is the cost weight $\lambda$. On the extreme, setting $\lambda = 0$ reduces A* to greedy BestFS, while setting $\lambda = \infty$ makes it equivalent to Breadth-First Search. By tuning that parameter, we control the trade-off between exploration and exploitation of the search.

To tune the depth parameter for our experiments, we grided over values $[0.1, 0.2, 0.5, 1, 2, 5, 10]$. However, usually the best choice was to keep the cost weight low (0.1 or 0.2, see Figure 36). While conservative search allows A* avoid more dead-ends than BestFS (see Figure 5.4), usually greedy steps lead to finding the solution much faster.

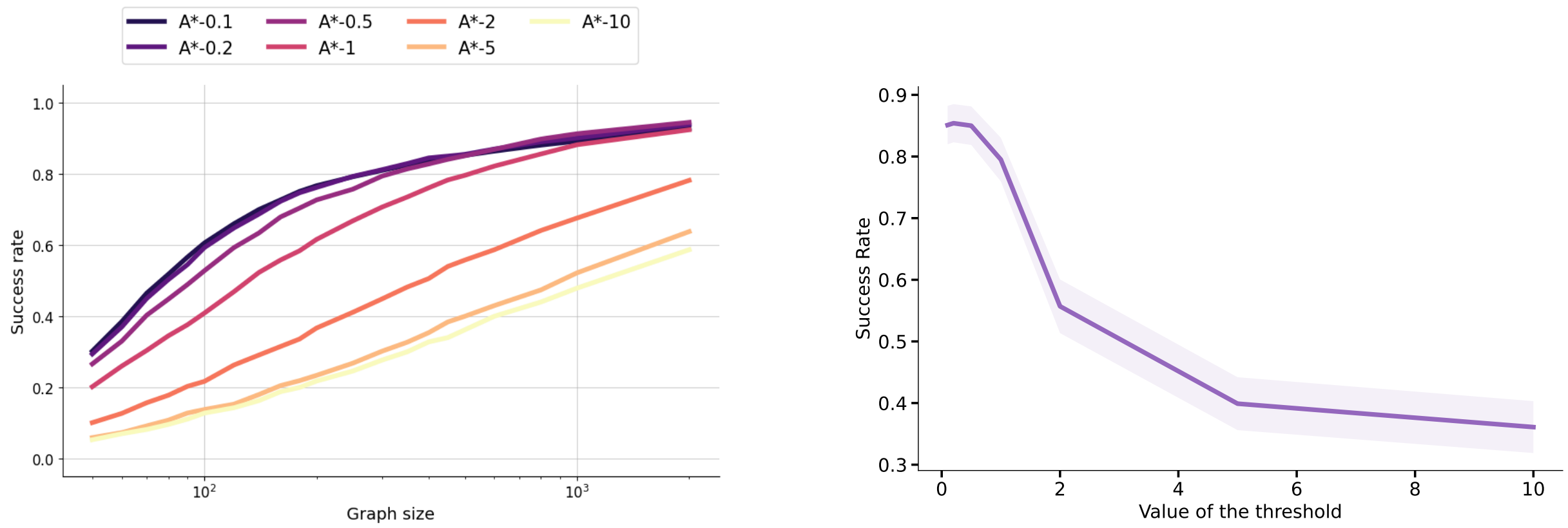

Figure 36: Figures presented above illustrate the impact of depth cost scaling on the overall success rate of the A* algorithm on Sokoban, employing a confidence threshold of 0.85. In most experiments, the smaller the depth scaling factor is, the better is the final success rate. The left figure shows the success rate curves for different choices of cost weight $\lambda$, while the right plot compares those variants for a fixed budget of 500 computation nodes.

**Algorithm 5** Complete pseudocode for $A^*$ Search

**Require:**
  value function network $V$
  policy $\rho_{BFS}$
  predicate of solution SOLVED
  depth scaling factor $\lambda$

```
function SEARCH(s_0)
T ← ∅ {priority queue}
T.PUSH((V(s_0), s_0))
parents ← {}
seen.ADD(s_0) {seen is a set}
while 0 < LEN(T) and LEN(seen) < max_budget do
  _, s ← T.EXTRACTMAX() {select node with the highest value}
  actions ← ρ_BFS(s)
  for a in actions do
    s' ← ENVSTEP(s, a)
    if s' in seen then
      continue
    end if
    seen.ADD(s')
    parents[s'] ← s
    T.PUSH((V(s') − λ · depth(s'), s'))
    if SOLVED(s') then
      {solution found}
      return EXTRACTLOWLEVELTRAJECTORY(s', parents)
    end if
  end for
end while
return False {solution not found}
```

### F.4 kSubS And AdaSubS

**Overview** AdaSubS is a hierarchical search algorithm designed to solve combinatorial problems by operating on high-level nodes, which represent multiple steps rather than single actions. It employs multiple generators $\mathcal{G}_{k_1}, \mathcal{G}_{k_2}, \dots, \mathcal{G}_{k_m}$ to generate subsequent subgoals, a value function $\mathcal{V}$ to estimate the distance from a given state to the solution, and a conditional low-level policy $\mathcal{P}$ to execute a series of actions leading from one subgoal to the next. kSubS is a special case of AdaSubS, where only a single generator is used. These methods are introduced and studied in (Czechowski et al., 2021; Zawalski et al., 2023).

**Stages** The method begins by adding $m$ initial nodes (one per each generator) to a priority queue, where each initial node $i$ is assigned a priority $(k_i, \mathcal{V}(s_0))$. Here, $k_i$ is the length of the generator used during the node's expansion, and $\mathcal{V}(s_0)$ estimates the distance (in low-level actions) between $s_0$ and the solution. The following steps are repeated until a solution is found or the budget is exhausted:

- **Selection for expansion**: The node $((k, \mathcal{V}(s), s)$ with the highest priority is extracted from the queue. This priority structure ensures that the algorithm prioritizes expanding the longest subgoals whenever possible.
- **Generating subgoals**: The current state $s$ is passed to the selected generator $\mathcal{G}_k$, which produces multiple subgoal propositions represented as states $s_1^*, s_2^*, \dots, s_p^*$.
- **Verifying reachability**: Since $\mathcal{G}_k$ can produce invalid or unreachable subgoals, each proposed subgoal must be verified. The conditional low-level policy $\mathcal{P}$ begins an iterative process, taking single steps from $s$ towards the proposed subgoal $s_j^*$. If $s_j^*$ is reached within $k$ steps, the subgoal is accepted, and new high-level nodes $\{\big((k_i, \mathcal{V}(s_j^*)), s_j^*\big)\}_{i \in \{1 \dots m\}}$ are added to the priority queue as potential future subgoals to expand.

For a graphical overview of how AdaSubS works, see Appendix H.

**Algorithm 6** Complete pseudocode for Adaptive Subgoal Search

```
Require:
  C_1 max number of nodes,
  V value function network,
  ρ_{k_0}, ..., ρ_{k_m} subgoal generators,
  SOLVED predicate of solution

  function SOLVE((s_0))
  T ← ∅ {priority queue with lexicographic order}
  parents ← {}
  for k in k_0, ..., k_m do
    T.push((k, V(s_0)), s_0)
  end for
  seen.add(s_0) {seen is a set}
  while 0 < len(T) and len(seen) < C_1 do
    (k, _), s ← T.extract_max()
    subgoals ← ρ_k(s)
    for s′ in subgoals do
      if s′ not in seen then
        if IS_VALID(s, s') then
          seen.add(s′)
          parents[s′] ← s
          for k in k_0, ..., k_m do
            T.push((k, V(s′)), s′)
          end for
          if SOLVED(s') then
            return EXTRACTLOWLEVELTRAJECTORY(s', parents)
          end if
        end if
      end if
    end for
  end while
  return False
```

### F.5 HIPS And HIPS-$\varepsilon$

Here we show a pseudocode for HIPS and HIPS-$\varepsilon$ methods. For details see Alg. 7

**Algorithm 7** Complete pseudocode for HIPS with BestFS-PHS* and VQ-VAE

**Require:**
  $C_1$ max number of nodes,
  $VAE$ Variational Autoencoder for subgoal generation,
  SOLVED predicate of solution,
  $\epsilon$ exploration parameter for balancing,
  $V$ value function for PHS* cost estimation

```
function EXTENDED_HIPS_SOLVE((s_0))
Initialize search data structures, including priority queues.
seen.add(s_0) {Track seen states}
while search conditions are met do
   Use PHS* search strategy to select a state s.
   Generate subgoals subgoals ← VAE(s).
   for each s′ in subgoals do
      if s′ not seen and is valid then
         Evaluate s′ using V for PHS* cost.
         Update priority queue based on PHS* cost.
         if SOLVED(s′) then
            return Construct solution path.
         end if
      end if
   end for
end while
return False {Solution not found}
```

## G Wall Times

In our experiments, we focus on measuring the search budget in terms of the number of visited states before finding the solution. However, it is also important to consider the total running time for completeness.

Subgoal methods introduce computational overhead. However, we note that each low-level method calls policy and value function once in every visited state, and similarly, subgoal methods also call policy and value once in every visited state. The only additional computation in subgoal methods comes from invoking the subgoal generator, which occurs in a fraction of the nodes explored. In each experiment, all methods share exactly the same heuristic function and use policies of equal size. As a result, the Complete Search Budget metric should be closely aligned with computational cost.

We opted to focus on a budget metric that is hardware-independent, reproducible, and widely applicable, ensuring that our results can serve as a reference point for future research. The Complete Search Budget answers the question "How many states must be explored before finding a solution?" rather than "How long does it take to find a solution?". These are slightly different questions, but both are relevant when assessing planner quality.

We acknowledge that we did not optimize the implementation for runtime efficiency, instead opting for the architectures used by (Czechowski et al., 2021) and (Zawalski et al., 2023) rather than optimizing computational complexity. Additionally, measuring wall-clock time introduces confounding factors, such as a bug in Hugging Face's beam search implementation that prevents decoding parallelization, introducing bias against subgoal methods.

For completeness, we report wall-clock times of each method in Table 3.

| | $\rho$-BestFS | $\rho$-A* | $\rho$-MCTS | kSubS | AdaSubS |
|---|---|---|---|---|---|
| Rubik | 26 | 26 | 153 | 214 | 96 |
| INT | 1997 | 1985 | - | 1444 | 1999 |
| Sokoban | 34 | 36 | 59 | 125 | 123 |
| NPuzzle | 27 | 32 | 29 | 40 | 39 |

Table 3: Comparison of evaluation time of search algorithms. The values express the total time of solving 500 instances, in minutes.

While all methods perform with similar runtime in the INT environment, subgoal methods generally require more time during evaluation in most other experiments. However, even the largest observed differences in evaluation time mildly affect the main conclusions. For example, the robustness of subgoal methods to value noise remains evident.

# H Hierarchical Search

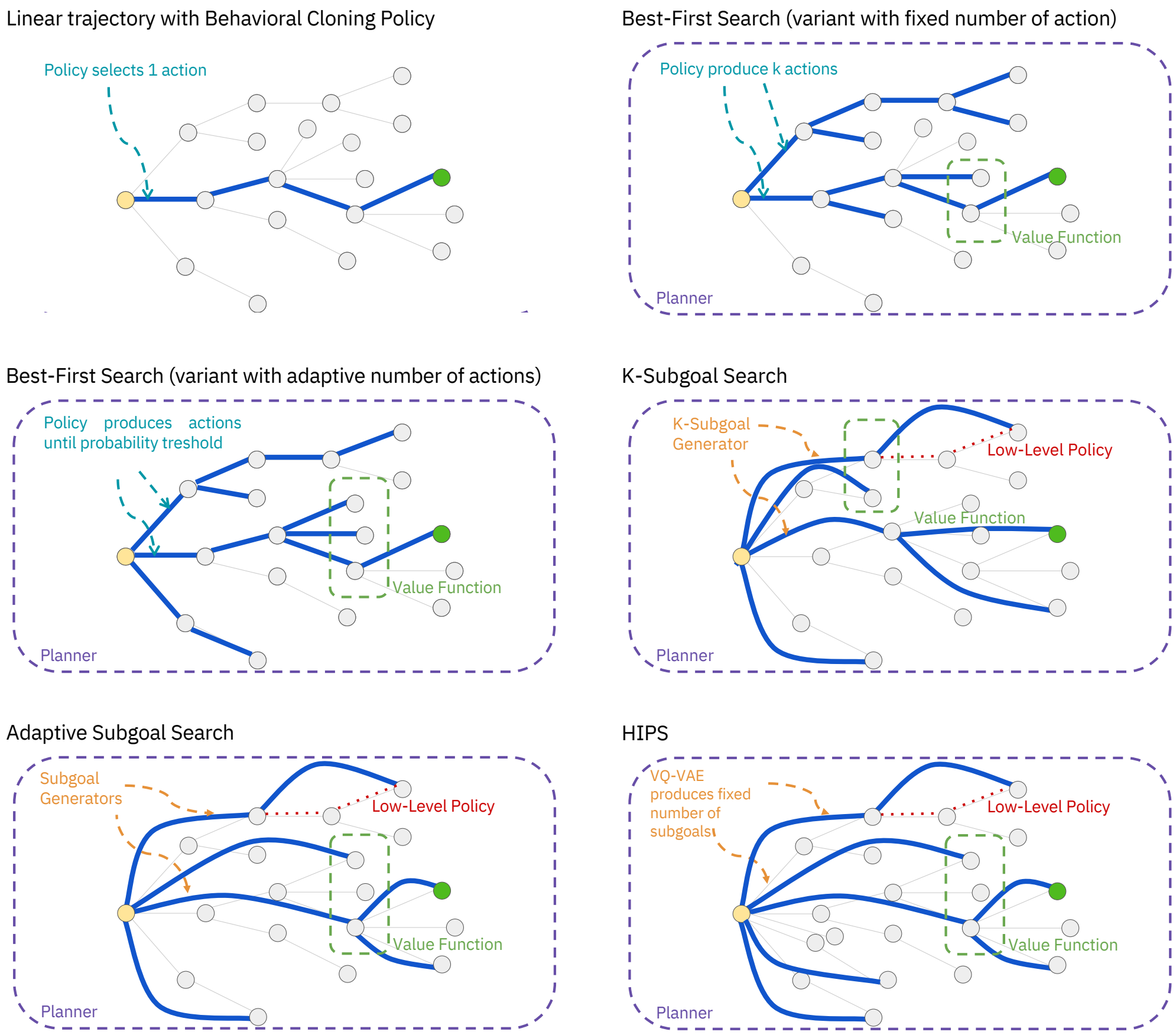

Figure 37: Overview of the search methods under consideration, accompanied by illustrative examples depicted in various plots for each method. Specifically, straight blue lines are utilized to represent low-level actions that occur within the search space. In contrast, long skip connections are used to symbolize subgoals within the search process.

# I Further Discussion On HIPS Results

HIPS and HIPS-$\varepsilon$ (Kujanpää et al., 2023a;b) are recent hierarchical search algorithms proposing to generate subgoals with variational autoencoders. We attempted to use HIPS and HIPS-$\varepsilon$ in greedy and prior-informed variations, and for all HIPS methods, the cost of inference was prohibitively high.

To compare these methods, we used A*-generated data from HIPS papers, in contrast to all other experiments (which use data generated by us).

Our evaluation, illustrated in 38, shows that HIPS uses 100x more low-level nodes in search than comparable subgoal search methods and baselines - despite relatively similar subgoal efficiency as calculated in relevant papers. These findings informed our decision not to evaluate HIPS in the rest of the paper.

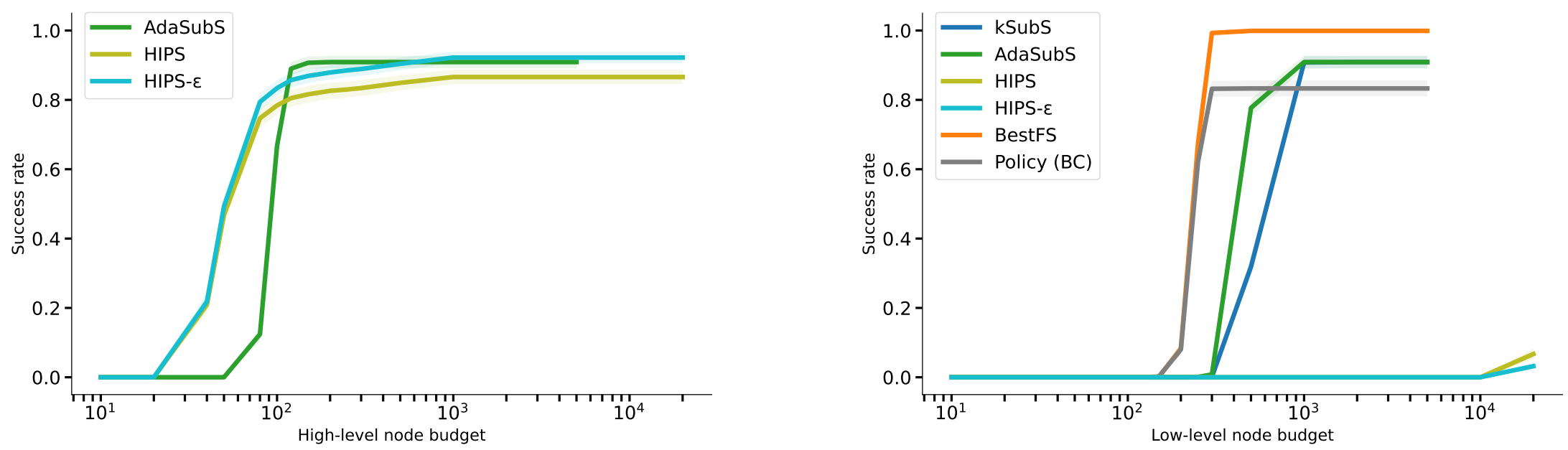

Figure 38: A comparison of high-level and low-level node budgets for considered methods: HIPS, subgoal search methods, and baselines on N-Puzzle. The low-level node budget represents the number of all states that have ever been visited during the search. The bimodal distribution indicates that HIPS methods use disproportionately (over 100x) more low-level nodes than comparable subgoal search methods and baselines. This directly translates to prohibitively slow solving time.

# J Common Pitfalls In Hierarchical Search evaluations

In this study, one of our primary goals is to identify common but often overlooked pitfalls in evaluating hierarchical search methods, which can lead to misleading conclusions. Based on our findings, we propose a set of guidelines that help ensure meaningful and consistent comparisons across different methods. We observed that the nature of hierarchical search makes it easy, whether intentionally or not, to present results in a way that favors certain methods, often without readers being aware. In this section, we present key insights on this issue, with an emphasis on the following evaluation guidelines:

- Report results using a *complete search budget*.
- Include $\rho$-BestFS with a confidence threshold as a baseline.
- Ensure careful tuning of the confidence threshold.
- Use up-to-date code for running experiments.

## J.1 Complete Search Budget

We define the performance metric in terms of *success rate*, which is the percentage of problem instances solved within a specified *complete search budget*. This budget refers to the total number of states visited during the search process. For hierarchical methods, this includes both the subgoals generated and the states visited by the low-level policies connecting those subgoals.

Reporting the *complete search budget* is crucial, as opposed to the *sparse search budget*, which counts only the high-level nodes in the search tree. As discussed in Appendix I, Kujanpää et al. (2023a) rely on the sparse search budget for their evaluations. This creates a misleading impression that HIPS outperforms low-level baselines, while in reality, it requires significantly more computational effort to solve the same problems.

To illustrate this issue, consider a simple environment where an agent must navigate a 100x100 empty room to reach a goal on the opposite side. In this case, a hierarchical method may require only a single subgoal – directly corresponding to the goal state – while a low-level method, even if following the optimal path, would require at least 100 steps. A sparse search budget would misleadingly indicate that the hierarchical method solves the task in one step, while the low-level approach requires 100 steps, implying a 100x higher cost. However, both methods traverse the same path, making this comparison inaccurate. Using the *complete search budget*, both methods would be assigned the same cost, providing a much more meaningful comparison.

This issue arises in practical settings as well. Figure 40 compares subgoal methods and low-level BestFS on the Sokoban environment. The dashed line represents the same runs but evaluated with the sparse search budget instead of the complete search budget. For BestFS, both budget measures are equivalent. The figure clearly demonstrates that while kSubS and $\rho$-BestFS visit a similar number of states to solve an instance, the sparse search budget falsely amplifies the difference between the two methods.

## J.2 Baselines

A common evaluation practice in hierarchical search studies is to compare hierarchical methods against the search algorithm used as the planner (Czechowski et al., 2021; Zawalski et al., 2023; Kujanpää et al., 2023a;b). While this is generally a good approach, it is critical to ensure that baseline methods are properly tuned to allow for fair comparisons.

Our study shows that the most effective low-level method is $\rho$-BestFS with a confidence threshold. This simple greedy search often performs significantly better than other low-level methods and, in some cases, is competitive with subgoal methods. However, if we were to follow prior works such as (Czechowski et al., 2021; Zawalski et al., 2023) and restrict our comparisons to variants of BestFS that select a fixed number of actions in each node expansion, without employing a confidence threshold (see Appendix F.1 for detailed definitions and analysis), we would artificially widen the gap between BestFS and subgoal methods. As noted in Appendix F.1, the performance of $\rho$-BestFS is highly sensitive to the confidence threshold, and

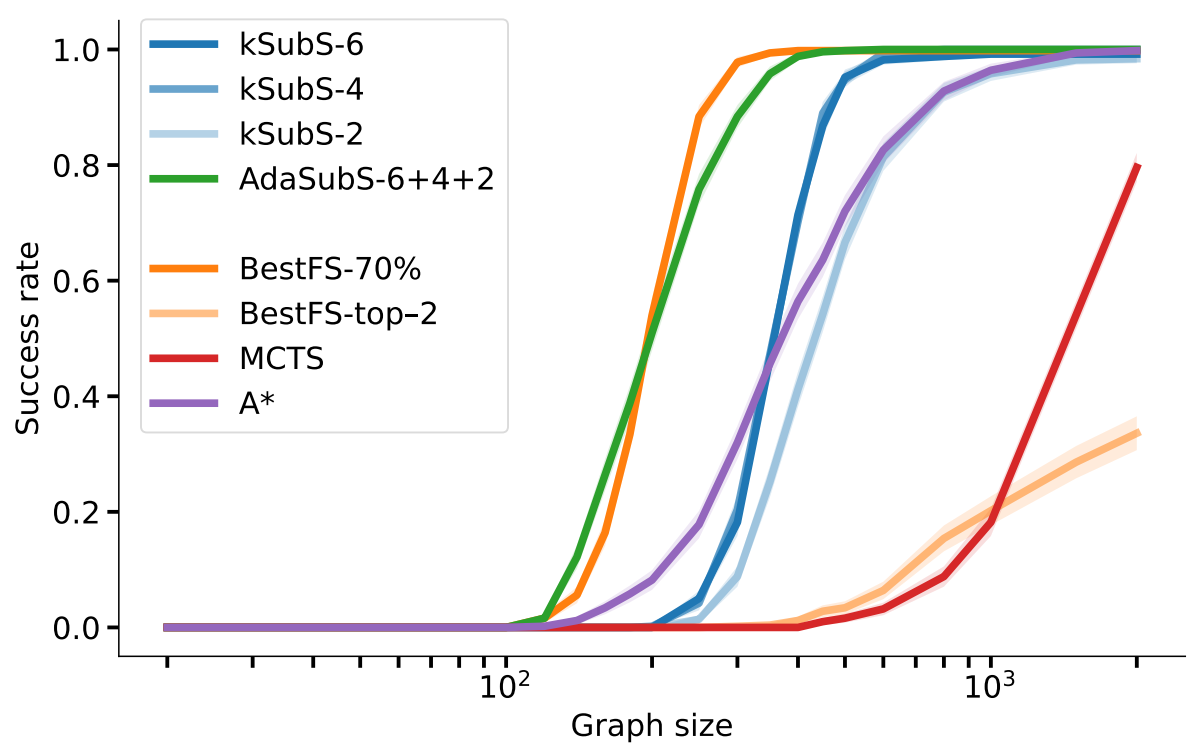

Figure 39: Solving the Rubik's Cube. The light orange line represents the best-performing variant of BestFS that selects a fixed number of actions for each expansion. The solid orange line represents BestFS with actions confidence threshold, which is much more efficient.

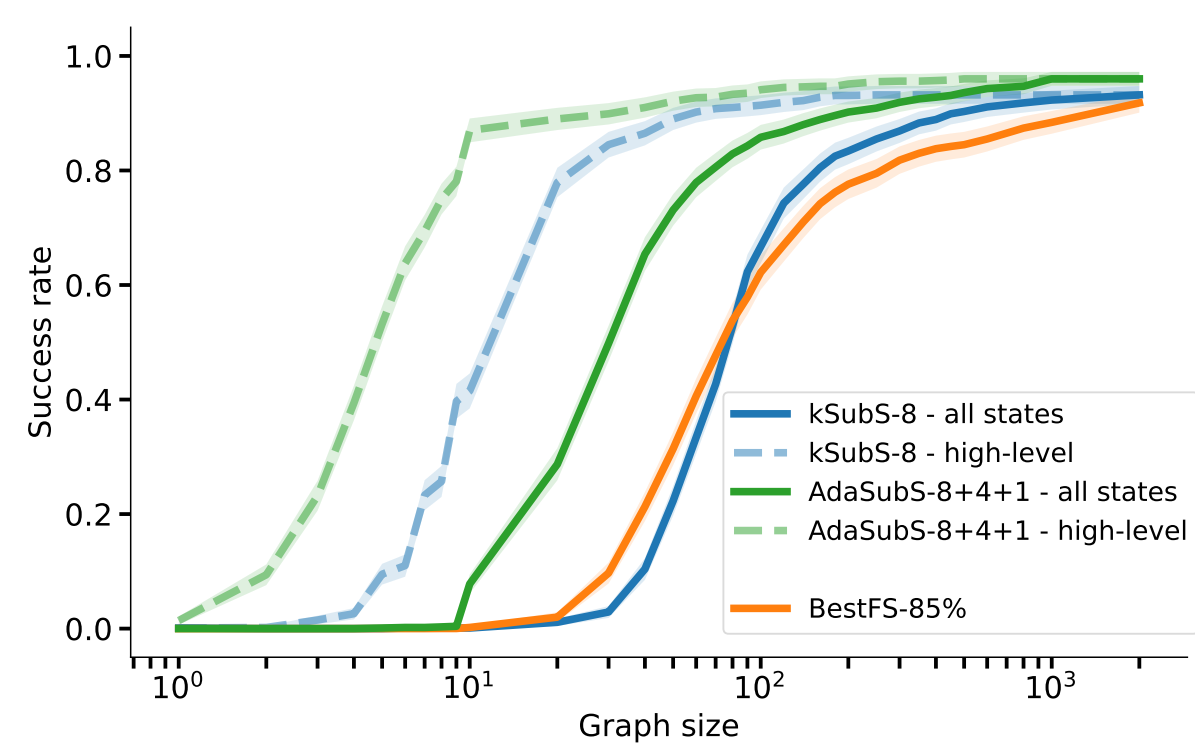

Figure 40: Solving Sokoban. Solid lines correspond to using *complete search budget* as the search tree size metric. Dashed lines correspond to the same runs, but using *sparse search budget* as the search tree size metric. For BestFS, both methods are equivalent.

proper tuning is essential. Nevertheless, we advocate for using $\rho$-BestFS with a confidence threshold as a standard baseline in evaluations of hierarchical methods.

### J.3 Code Quality

While our results generally align with the findings of (Czechowski et al., 2021; Zawalski et al., 2023), we observed some notable differences. Most strikingly, when components were trained on reverse random shuffles of the Rubik's Cube, our models demonstrated significantly better performance. In particular, (Zawalski et al., 2023) reports that both kSubS and AdaSubS substantially outperform $\rho$-BestFS. However, in our experiments, these methods perform similarly, with only minor differences between them (see Figure 41).

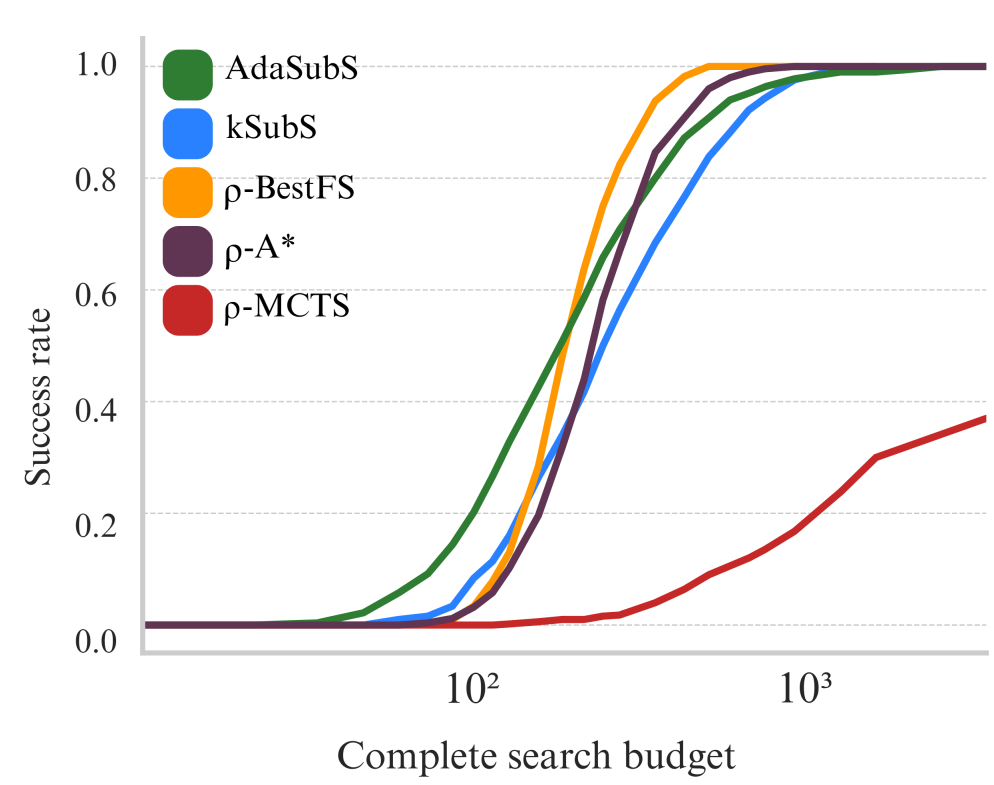

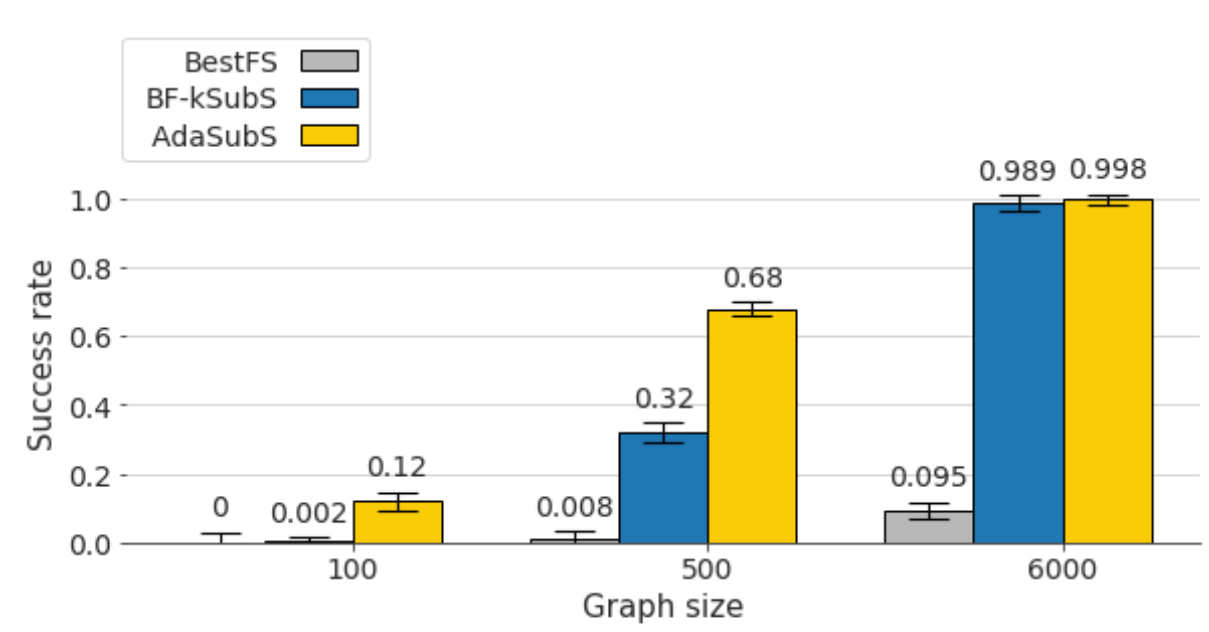

Figure 41: Solving the Rubik's Cube. Components are trained on reverse random shuffles. The left chart present our results, while the right presents results of the same experiment from (Zawalski et al., 2023).

For this study, we re-implemented all algorithms from scratch, using up-to-date libraries and carefully tuning hyperparameters. Our experiments revealed that low-level methods are highly sensitive to the quality of the value function, whereas subgoal-based methods are more resilient (Section 5.2). We hypothesize that the discrepancy in performance compared to (Czechowski et al., 2021; Zawalski et al., 2023) may stem from insufficient training of the value function in their implementation, leading to the observed performance gap.

Using the original implementations of kSubS and AdaSubS, which is a common practice, would replicate the same limitation. This shows the importance of re-implementing algorithms independently and carefully tuning their components, ensuring that evaluations are not biased by potential shortcomings in the original implementations.

## K Proof Of The Search Advancement formula

**Theorem 3** (Search advancement formula, complete statement)**.** *Let $g_k : S \to \mathcal{P}(S)$ be a stochastic $k$-subgoal generator that, given a state $s \in S$ samples a set of $b$ subgoals $\{s_i\}$ such that the distances $d(s_i, s)$ are independent, uniformly distributed in the interval $[-k; k]$. Let $V : S \to \mathbb{R}$ be a value function with approximation error uniformly distributed in the interval $[-\sigma; \sigma]$.*

*Then, after $n$ iterations of search, the expected total progress toward the goal is:*

$$\mathbb{E}_{Adv} = \frac{nb}{4\sigma k} \int_{-k}^{k} x \left( \int_{-\sigma}^{\sigma} \tilde{u}(x+h)^{b-1} \mathrm{d}h \right) \mathrm{d}x, \tag{3}$$

*where $\tilde{u}(x)$ is CDF of the sum of two uniform variables $U(-k,k)+U(-\sigma,\sigma)$. Additionally, if we approximate that sum as $U(-k-\sigma, k+\sigma)$, we get*

$$\mathbb{E}_{Adv} \approx \frac{n\left((k+\sigma)^b(bk^2 + bk\sigma - 2k\sigma - 2\sigma^2) + \sigma^b(2k\sigma + bk\sigma + 2\sigma^2) - k^b(bk^2)\right)}{(b+1)(b+2)k\sigma(k+\sigma)^{b-1}} \tag{4}$$

*Proof.* Let $A_1, \ldots, A_b$ be independent and identically distributed (i.i.d.) random variables sampled from $U(-k,k)$, and let $B_1, \ldots, B_b$ be i.i.d. random variables sampled from $U(-\sigma,\sigma)$. Denote the CDF of the sum $A_i + B_i$ as $\tilde{u}(x)$, and its corresponding probability density function (PDF) as $p(x) = \tilde{u}'(x)$. Let $I = \arg\max_i(A_i + B_i)$.

We now define the cumulative likelihood of selecting the largest sum among the subgoals:

$$CLS(x) = \mathbb{P}\left(\forall_{1 \le i \le b}\, A_i + B_i < x\right).$$

Since the $A_i$'s and $B_i$'s are independent, it follows that $CLS(x) = \tilde{u}(x)^b$, which represents the cumulative distribution of the largest sum $A_i + B_i$. Differentiating this expression gives the PDF of the largest sum:

$$PLS(x) = CLS'(x) = b \cdot \tilde{u}(x)^{b-1} \cdot p(x).$$

Now, consider the event that $A_I = x$, which is equivalent to the event that the maximum $\max_i(A_i + B_i) = x + h$ for some $h \in [-\sigma, \sigma]$ and $B_I = h$. Given that $\max_i(A_i + B_i) = x + h$, there are $p(x+h) \cdot 4\sigma k$ possible values of $B_I$, since $A_I \in [-k, k]$ and $B_I \in [-\sigma, \sigma]$. Therefore, the PDF of this variable is

$$q(x) = \int_{-\sigma}^{\sigma} \frac{PLS(x+h)}{p(x+h) \cdot 4\sigma k} \,\mathrm{d}h = \int_{-\sigma}^{\sigma} \frac{b \cdot \tilde{u}(x+h)^{b-1}}{4\sigma k} \,\mathrm{d}h.$$

Thus, the expected value of $A_I$, which represents the progress in each step, is given by

$$\mathbb{E}[A_I] = \int_{-k}^{k} x q(x) \,\mathrm{d}x = \frac{b}{4\sigma k} \int_{-k}^{k} x \left( \int_{-\sigma}^{\sigma} \tilde{u}(x+h)^{b-1} \,\mathrm{d}h \right) \mathrm{d}x.$$

If we model the search process as advancing to the best subgoal in each iteration, the total expected progress after $n$ iterations is

$$\mathbb{E}_{Adv} = n\mathbb{E}[A_I] = \frac{nb}{4\sigma k} \int_{-k}^{k} x \left( \int_{-\sigma}^{\sigma} \tilde{u}(x+h)^{b-1} \,\mathrm{d}h \right) \mathrm{d}x.$$

Finally, by approximating the PDF $p(x) \approx \frac{1}{2k+2\sigma} \mathbb{1}_{[-k-\sigma, k+\sigma]}$, and substituting this approximation into the previous expression, we arrive at the closed-form approximation:

$$\mathbb{E}_{Adv} \approx \frac{n\left((k+\sigma)^b(bk^2 + bk\sigma - 2k\sigma - 2\sigma^2) + \sigma^b(2k\sigma + bk\sigma + 2\sigma^2) - k^b(bk^2)\right)}{(b+1)(b+2)k\sigma(k+\sigma)^{b-1}}.$$

□

## L Proof Of The Densification Of The Action Space Theorem

In Section 5.3, we showed experimentally that both in the mathematical INT environment and Rubik's Cube with multiplied action space the advantage of subgoal methods is significant. We attributed those benefits to the ability of subgoal methods to use states as actions and the reduced diversity in low-level search. And indeed, we can prove in general that as the action space gets more complex, the diversity of top actions drops.

To give an illustrative example, in the Rubik's Cube experiment, to model the increasingly complex action space, for an arbitrary state we can view the training data as a ground-truth density function $f$ over an interval $[0, 1]$, that is split evenly between the actions (i.e. into 12 intervals of length 1/12). Then, we can define arbitrarily dense action spaces $A_n$ consisting of $n$ points distributed evenly in the domain. For instance, $A_{12}$ corresponds to the standard Rubik's Cube action space, while $A_{1200}$ corresponds to the variant multiplied 100 times. Our theorem confirms that the actions selected by the policy gets less diverse as the complexity of the action space increases, up to the extreme of converging to a single point as $n$ approaches infinity. In practice, it is even more general, since the data-driven action distribution $f$ may also model smooth interpolation between actions.

While this is rather intuitive when the learned distributions are perfect, it may seem that approximation errors, induced both by the limited training data and the policy network can actually improve diversity. We show that the result holds even in presence of arbitrarily large approximation errors, which is a bit counter-intuitive.

Formally, the theorem is as follows:

**Theorem 4** (Densification of the action space)**.** *Fix any state $s$ from the state space $S$. Let $f : A \to [0, 1]$ be the action distribution induced by the data-collecting policy for the state $s$. Assume that $f$ is continuous and has a unique maximum. For clarity, assume $A = [0, 1]$.*

*Consider a sequence of increasingly dense discrete action spaces $A_n := \{i/n\}_{i=0}^{n} \subset A$. Let $\rho_n : S \times A_n \to [0, 1]$ be a family of policies that learn the distribution $f|_{A_n}$ over actions, with uniform approximation error $U(-E, E)$, where $E \in \mathbb{R}_+$. Let $r_n$ be the range of the top $K$ actions according to the probabilities estimated by $\rho_n$. Then*

$$\lim_{n\to\infty} \mathbb{E}[r_n] = 0.$$

Intuitively, this theorem states that as the action space become more dense and complex, the actions sampled for search become increasingly less diverse, which strongly impedes successful planning. Note that this analysis is strictly more general than the experiment in Section 5.3 with the Rubik's Cube environment, where we simply copied the available actions. Here we model the complexity by adding dense intermediate actions, which leads to a similar conclusion.

While we assume a one-dimensional action domain for clarity, it is straightforward to generalize the proof to cover arbitrarily high-dimensional action spaces.

Firstly, we shall prove the following key lemma.

**Lemma 1.** *Let $f : [0, 1] \to \mathbb{R}$ be a continuous function with a unique maximum. Let $\{a_n\}$ be a partition of the interval $[0, 1]$ into $n$ uniformly spaced points, i.e., $a_{n,i} = \frac{i}{n}$ for $i = 0, 1, \ldots, n$. Define $e_{n,i}$ as i.i.d. samples from a uniform distribution $U(-E, E)$. For a fixed $n$, let $r_n \in \mathbb{R}$ denote the smallest interval length such that the points in $\{a_n\}$ corresponding to the top $K$ values of $f(a_{n,i}) + e_{n,i}$ are contained within this interval. Then*

$$\lim_{n\to\infty} \mathbb{E}[r_n] = 0.$$

*Proof.* Define $p_{n,i,k}$ as the probability that $f(a_{n,i}) + e_{n,i}$ is the $k$-th highest value among all points in $\{a_n\}$. Let $m$ be the unique point such that $f(m)$ is maximal. Without loss of generality, we may assume that $m = 0$.

Let $d_{n,k}$ denote the expected distance of the $k$-th highest point from 0, expressed as

$$d_{n,k} := \sum_{i=0}^{n} p_{n,i,k} a_{n,i}.$$

For sufficiently large $n$, it holds that $r_n \leq d_{n,1} + \ldots + d_{n,K} \leq K d_{n,K}$. Thus, it suffices to prove that $\lim_{n\to\infty} d_{n,K} = 0$.

Fix $\alpha \in (0,1)$ such that $f(a_{n,\alpha n}) \geq f(a_{n,\alpha' n})$ for each $\alpha' > \alpha$. Since $f$ is continuous and $m = 0$ is the unique maximum of $f$, there exist such $\alpha$ arbitrarily close to 0. Let $q_{n,\alpha}$ be the probability that $f(a_{n,\alpha n}) + e_{n,\alpha n}$ is among the top $K$ values. Since $m$ is a unique maximum, there exists $0 < \beta < \alpha$ such that $f(a_{n,\beta n}) > f(a_{n,\alpha n})$. Therefore, if at least $K$ points $a_{n,i}$ with $i/n < \beta$ satisfy $e_{n,i} > E - (f(a_{n,\beta n}) - f(a_{n,\alpha n}))$, then $f(a_{n,\alpha n}) + e_{n,\alpha n}$ cannot be among the top $K$. The probability of this event is a strict upper bound on $q_{n,\alpha}$.

The events $e_{n,i} > E - (f(a_{n,\beta n}) - f(a_{n,\alpha n}))$ are pairwise independent, each occurring with probability

$$c := \frac{f(a_{n,\beta n}) - f(a_{n,\alpha n})}{2E} > 0.$$

For sufficiently large $n$, the probability that at most $K$ of the $\beta n$ trials succeed is bounded by

$$1 - K \binom{\beta n}{K} (1-c)^{\beta n}.$$

Using the asymptotic behavior of binomial coefficients and exponential terms, it follows that

$$\lim_{n\to\infty} n^2 q_{n,\alpha} = 0, \tag{5}$$

with convergence that is exponential.

Using the definition of $d_{n,K}$, decompose it as

$$d_{n,K} = \sum_{i=0}^{n} p_{n,i,K} a_{n,i} = \sum_{i=0}^{\alpha n} p_{n,i,K} a_{n,i} + \sum_{i=\alpha n}^{n} p_{n,i,K} a_{n,i}.$$

For $i \geq \alpha n$, since we know that $f(a_{n,\alpha n}) \geq f(a_{n,\alpha' n})$ for each $\alpha' > \alpha$, we can bound $p_{n,i,K}$ by $p_{n,\alpha n,K}$ for sufficiently large $n$. Therefore

$$\sum_{i=\alpha n}^{n} p_{n,i,K} a_{n,i} \leq (1-\alpha) n p_{n,\alpha n,K}.$$

Since $p_{n,\alpha n,K} \leq q_{n,\alpha}$, it follows that

$$(1-\alpha) n^2 p_{n,\alpha n,K} \leq (1-\alpha) n^2 q_{n,\alpha}.$$

According to Equation 5, this term converges to 0.

For $i \leq \alpha n$, observe that $a_{n,i} < \alpha$ and the probabilities $p_{n,i,K}$ sum to at most 1. Thus

$$\sum_{i=0}^{\alpha n} p_{n,i,K} a_{n,i} \leq \alpha.$$

Combining these bounds, we have

$$\lim_{n\to\infty} d_{n,K} \leq \alpha.$$

Since $\alpha > 0$ was an arbitrarily small constant, it follows that $\lim_{n\to\infty} d_{n,K} = 0$.

By the relation $r_n \leq K d_{n,K}$ and the fact that $\lim_{n\to\infty} d_{n,K} = 0$, we conclude that

$$\lim_{n\to\infty} \mathbb{E}[r_n] = 0.$$

□

Now, Theorem 4 is a straightforward implication of Lemma 1, applied to the sequence of policies $\rho_n$ and increasingly dense action spaces $A_n$.

## M Comparison with DeepCubeA

In contrast to the general-purpose search methods and pre-defined heuristics examined in our main study, DeepCubeA (McAleer et al., 2019) takes a different approach: it learns a value function and heuristic directly through deep reinforcement learning. This allowed DeepCubeA to successfully solve the Rubik's Cube without relying on human-provided knowledge. To provide a more complete picture of the performance landscape, and to understand the relative strengths of learned versus pre-defined heuristics, we include a comparison with DeepCubeA.

DeepCubeA employs Iterative Deepening A* (IDA*) as its core search algorithm. IDA* is a variant of A* that performs a series of depth-first searches with increasing cost thresholds. In each iteration, it explores nodes in a depth-first manner, but only up to a maximum cost defined by *f(node) = g(node) + h(node)*, where *g(node)* is the path cost (depth) and *h(node)* is the heuristic estimate of the remaining cost. If a solution is not found within the current threshold, the threshold is increased, and the search restarts. This process continues until a solution is found or a resource limit is reached.

While IDA* guarantees finding an optimal solution (given an admissible heuristic), it can revisit the same nodes multiple times across iterations, leading to redundant computations. A*, as described in Section 5, maintains an open list of all explored nodes, avoiding this redundancy. Because A* explores all nodes up to a given cost before expanding nodes with higher costs, and given that we are primarily concerned with finding any solution rather than necessarily the optimal solution, A* provides a more efficient exploration strategy for our analysis, and effectively majorizes the behavior of IDA*.

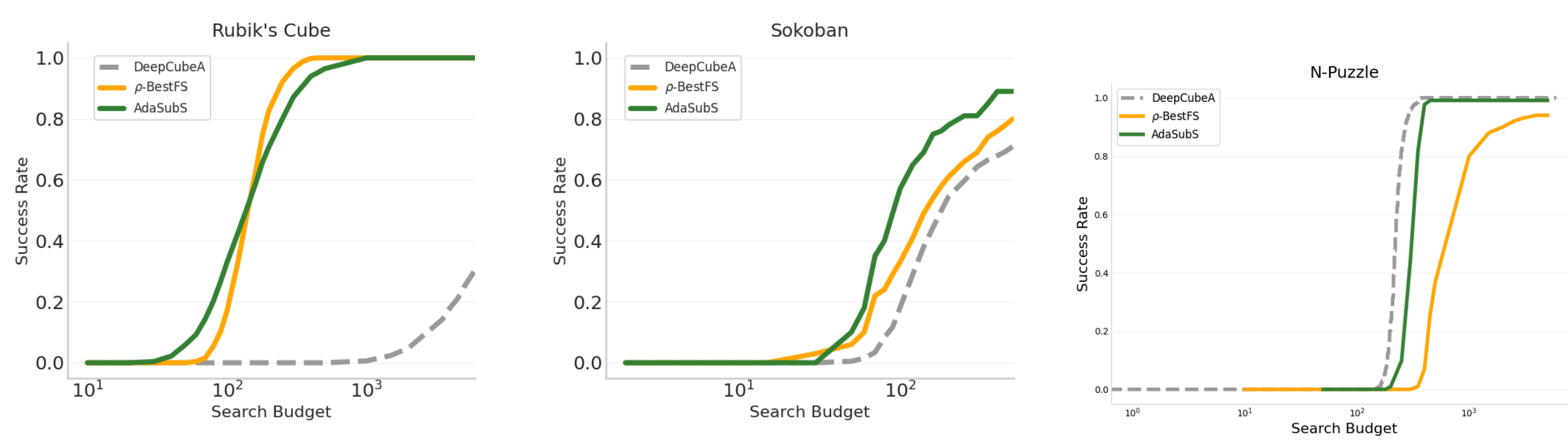

Figure 42: Comparison with DeepCubeA

Figure 42 presents a comparison of methods used in our study (hierarchical AdaSubS and low-level $\rho$-BestFS) with DeepCubeA – a well-established algorithm that solved the Rubik's Cube with deep learning and tree search, without human knowledge. The plots show evaluation in Rubik's Cube (left), Sokoban (middle), and N-Puzzle (right). The performance of DeepCubeA is weaker or on-par with the methods that we analyze in the paper.

The takeaway from this comparison is twofold. Firstly, performance of our implementations is competitive with well-established general-purpose solvers. Secondly, it is hard to understand the relation between search algorithms if they use different heuristics for solving. Hence, we stress that in each experiment presented in the main paper, all methods share the same value function to ensure a fair comparison.

## N Solution quality

| Environment | Algorithm | Tree size | Solution length | Solution length (subgoals) |
|---|---|---|---|---|
| N-Puzzle | BestFS | 354.43 | 354.08 | - |
| | A* | 354.09 | 353.56 | - |
| | MCTS | 742.04 | 347.43 | - |
| | kSubS-8 | 353.66 | 353.66 | 45.67 |
| Sokoban | BestFS | 185.24 | 48.98 | - |
| | A* | 85.04 | 45.68 | - |
| | MCTS | 255.0 | 45.1 | - |
| | kSubS-8 | 101.92 | 46.88 | 7.23 |
| Rubik's Cube | BestFS | 152.25 | 48.92 | - |
| | A* | 185.23 | 45.46 | - |
| | MCTS | 716.46 | 33.32 | - |
| | kSubS-4 | 303.52 | 73.58 | 26.65 |

Table 4: Average values of tree size, number of leaves, branching factor (average number of children), and solution length were calculated for 100 boards solved by all presented algorithms. Additionally, for the subgoal method, the average number of subgoals on the winning path was determined.

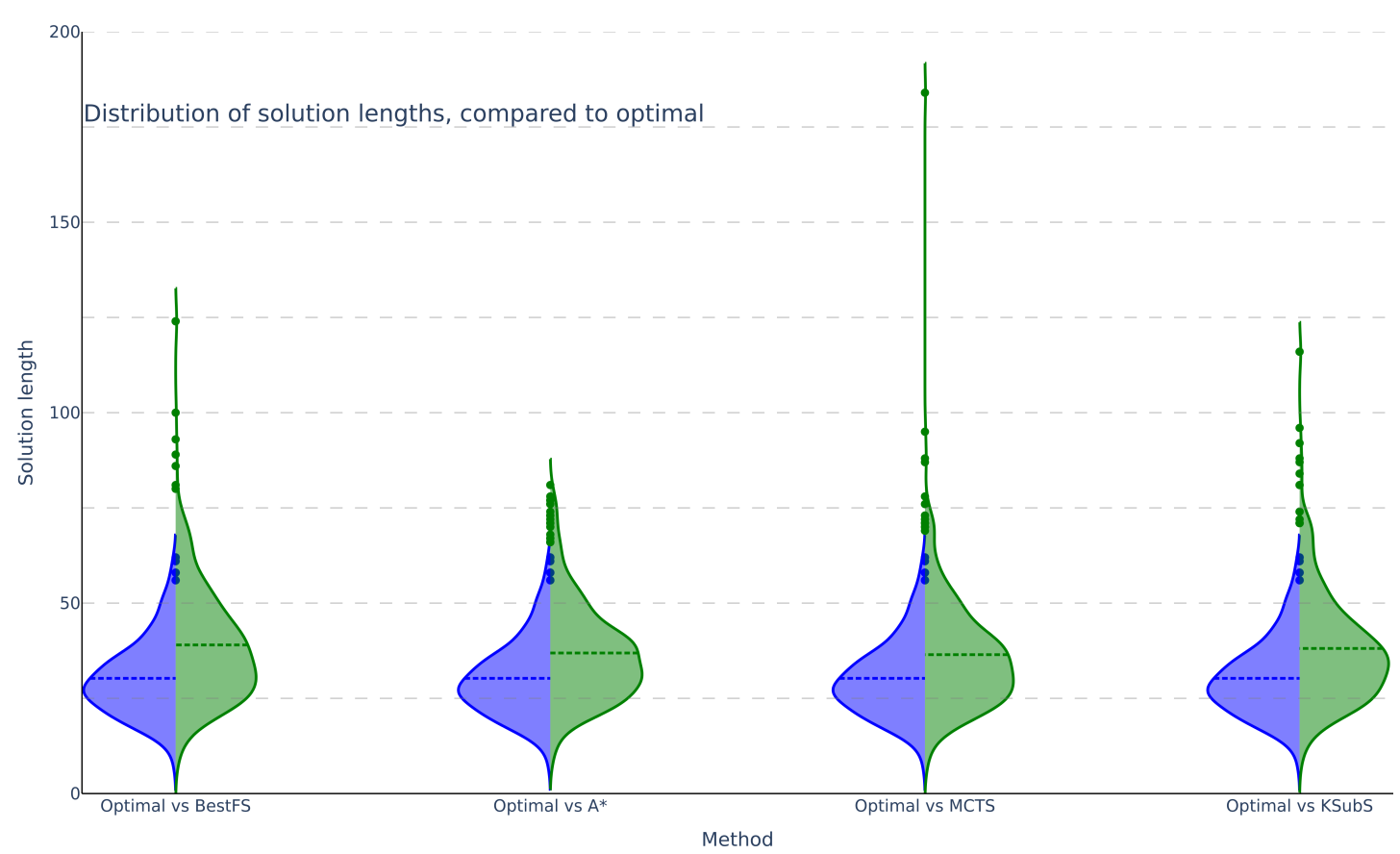

Figure 43: The distribution of solution length in Sokoban. The right part of each plot illustrates the distribution for the methods that we used. The left part corresponds to the optimal solutions for the tested instances obtained using Breadth-First Search. These algorithms were evaluated on 494 commonly solved instances.

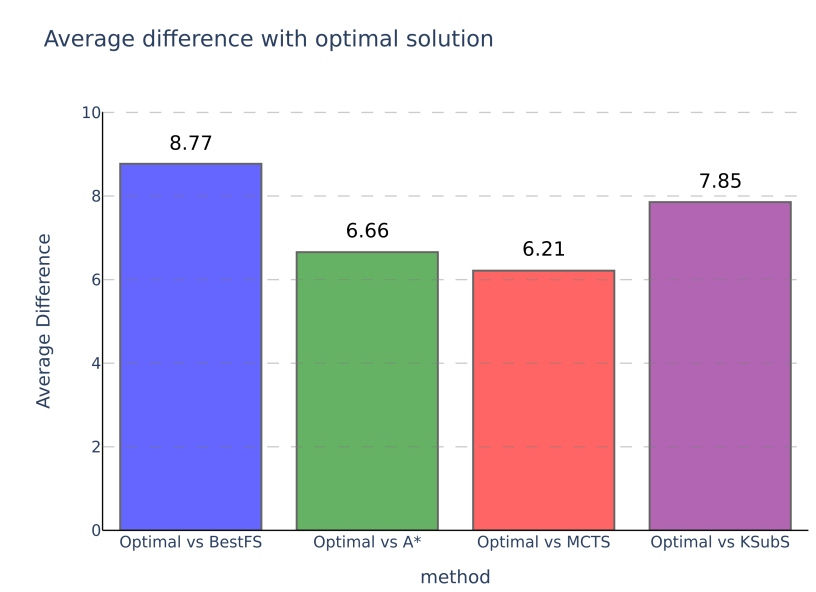

Figure 44: The average difference between the solutions found by each algorithm and the optimal solutions for the Sokoban environment. These algorithms were evaluated on 494 commonly solved instances.

Quite surprisingly, in each domain, the shortest solutions are found by p-MCTS. p-A* typically performs only slightly worse in that metric. While A* is theoretically capable of finding optimal solutions, our heuristic is not guaranteed to be admissible, and we additionally apply depth weighting for greater efficiency, which introduces suboptimality. Interestingly, Subgoal Search finds shorter solutions than its low-level counterpart p-BestFS in Sokoban and N-Puzzle, but longer ones in the Rubik's Cube.

We also compared the solutions found in Sokoban to optimal paths computed using BFS. On average, the computed solutions are about 6–9 steps longer.

We note that the quality of solutions found by hierarchical search is bounded by (1) how often the generator proposes subgoals on the optimal path, (2) the quality of low-level paths computed by the policy, and (3) the ability of the value function to recognize optimal states. In contrast, low-level methods are bounded only

by the value function (assuming they can expand actions exhaustively). However, we note that AdaSubS, one of the hierarchical methods we evaluate, can effectively leverage both long-horizon and short, 1-step subgoals, helping bridge this gap. We believe that further tuning that method could produce a hierarchical search that explicitly balances effectiveness and solution quality. However, while this is an exciting direction to explore, it lies beyond the scope of our current work and we leave it for future research.