# OpenReview forum: "What Matters in Hierarchical Search for Solving Combinatorial Problems?"
_TMLR — Rejected by TMLR_

### Review · Reviewer_ssrG · 2025-06-04

**Summary Of Contributions:**

The authors present a systematic comparison between hierarchical -vs- non-hierarchical solution techniques in the context of heuristics learned via imitation learning. There are a handful of theoretical results that support the general conclusions, and the performance is measured across several axes. In general, the finding is that subgoal hierarchical methods outperform the lower level direct ones.

**Audience:**

Yes

**Broader Impact Concerns:**

I don't feel as though further discussions on the broader impact are required.

**Claims And Evidence:**

Yes

**Requested Changes:**

- I think the title should reflect that this is only for heuristics learned via IRL

- I would like to see a more formal characterization of the task being solved included

- Figure 1 should be discussed in the main text, and the axes in the diagram should correspond to the main ones highlighted in the abstract, intro, etc.

- It's not clear to me (partially due to a lack of a formal problem characterization) how the sub-goal methods work here. Some of the methods are cited, but it would be useful to highlight the details of a naive hierarchical approach. I.e., something that would explain how Figure 16 is generated.

**Strengths And Weaknesses:**

## Strengths

The paper is very well written, and the two theorems are on point. I found the empirical evaluations conducted to be compelling for the axes explored, and particularly found the Value Noise exploration very compelling -- Figure 7 drives home the main point perfectly.


## Weaknesses

I have several minor gripes with the work, and many amount to a common thread: the assumptions -- even in this rewrite -- are not clearly stipulated. From the title, and much of the motivating text, I was expecting to see results that included HTN's (what the planning community would typically refer to as "Hierarchical Planning"), classical planning techniques (A*, BestFS was used, but not with a planning model), etc.

More specifically, the techniques seem to hop around the formalism landscape. All of the domains explored are deterministic and fully observable. So why then bring in MDPs and MCTS (settings that address probabilistic action effects)?

In a similar vein, the complexity cited seems to miss the mark -- planning problems of this form are not typically NP-Hard, but rather PSPACE. Even the reference used for Sokoban's complexity (Culberson '97) states the PSPACE complexity in the title!

Given that these are classical planning problems, it strikes me as odd that there's no comparison to those techniques. There are strong domain independent planners out there that would perform very well on these benchmarks. At the same time, this isn't a paper claiming that IRL heuristics will outperform state-of-the-art planners, so that motivates _not_ having those comparisons. However, I'd expect to see some domains included that classical planning are just not suited to -- if no such domains exist, then why would we care about search techniques using heuristics learned via IRL? To be clear, I don't think this is the case, but it speaks to the odd choice of domains.

---

> ### Author Response · Authors · 2025-07-07
> **Response to Reviewer ssrG (Part 1)**
>
> > I have several minor gripes with the work, and many amount to a common thread: the assumptions -- even in this rewrite -- are not clearly stipulated. From the title, and much of the motivating text, I was expecting to see results that included HTN's (what the planning community would typically refer to as "Hierarchical Planning"), classical planning techniques (A*, BestFS was used, but not with a planning model), etc.
>
> > I think the title should reflect that this is only for heuristics learned via IRL
>
> We thank the Reviewer for this feedback. While including all the relevant details in the short title is hard, we made our best efforts to stress that our focus is on methods with learned components, both in the Abstract and Introduction, and through the whole paper.. Following your feedback, we additionally clarified it in the abstract to avoid confusion, so that now in the abstract alone we state it multiple times.
>
> We agree that comparing the studied methods against classical methods, such as HTNs, is an exciting direction to study. However, when comparing against methods with handcrafted components, such as heuristics or task decompositions, it is much harder to attribute the observed differences to a particular component. Hence, we study knowledge-free domain-independent learning-based methods to ensure the observed differences come solely from the usage of subgoals.
>
> > More specifically, the techniques seem to hop around the formalism landscape. All of the domains explored are deterministic and fully observable. So why then bring in MDPs and MCTS (settings that address probabilistic action effects)?
>
> > I would like to see a more formal characterization of the task being solved included
>
> We would like to clarify that the MDP framework is a standard formalism for sequential decision-making tasks, including both stochastic and deterministic settings. Following your suggestion, we have formally defined the problems we aim to solve in the new Section 3.1.
>
> While the choice of MCTS may seem non-obvious – given that it is often associated with stochastic environments – we selected it because it represents a substantially different approach to planning compared to BestFS and A*. Additionally, it has seen widely recognized success in systems like AlphaZero, which is based on MCTS.
>
> > In a similar vein, the complexity cited seems to miss the mark -- planning problems of this form are not typically NP-Hard, but rather PSPACE. Even the reference used for Sokoban's complexity (Culberson '97) states the PSPACE complexity in the title!
>
> We would like to clarify that since NP $\subseteq$ PSPACE, every PSPACE-complete problem is also NP-hard. While we agree that your point is theoretically more precise, we have chosen to refer to NP rather than PSPACE, as NP is more commonly used as a point of reference.
>
> > Given that these are classical planning problems, it strikes me as odd that there's no comparison to those techniques. There are strong domain independent planners out there that would perform very well on these benchmarks. At the same time, this isn't a paper claiming that IRL heuristics will outperform state-of-the-art planners, so that motivates not having those comparisons. However, I'd expect to see some domains included that classical planning are just not suited to -- if no such domains exist, then why would we care about search techniques using heuristics learned via IRL? To be clear, I don't think this is the case, but it speaks to the odd choice of domains.
>
> We thank the reviewer for this insightful comment. We compared our main methods with the established DeepCubeA solver (see Appendix M), and found that our implementations are competitive – with performance that is on par with or stronger than DeepCubeA.
>
> Secondly, this evaluation highlights the inherent difficulty of comparing search algorithms when they use different heuristics. This challenge is precisely why our main analysis employs a shared value function – to isolate and fairly compare the search methods themselves.
>
> To address the important point about problem suitability, we specifically included the INT (INequality Theorem proving) domain. INT is characterized by an enormous action space (up to a million valid actions per step) and the absence of any known effective handcrafted heuristics. As such, it demonstrates an application for learning heuristics via IRL in complex scenarios where classical planning methods are ill-suited.
>
> > Figure 1 should be discussed in the main text, and the axes in the diagram should correspond to the main ones highlighted in the abstract, intro, etc.
>
> We understand your point. Figure 1 is intended as a schematic illustration of our results. Each axis corresponds to a subsection in the Experiments section, summarizing the outcome of the key experiment. Following your suggestion, we have clarified its description to better explain that.

---

> > ### Author Response · Authors · 2025-07-07
> > **Response to Reviewer ssrG (Part 2)**
> >
> > > It's not clear to me (partially due to a lack of a formal problem characterization) how the sub-goal methods work here. Some of the methods are cited, but it would be useful to highlight the details of a naive hierarchical approach. I.e., something that would explain how Figure 16 is generated.
> >
> > Similar to BestFS, Subgoal Search iteratively builds a search tree, only composed of subgoals. In each iteration, it takes the highest-valued state S in the tree and generates subgoal candidates using the subgoal generator. For each subgoal candidate, the low-level policy is employed to reach it from S, using atomic actions. The subgoals successfully reached are then added to the search tree and the next iteration follows.
> >
> > Figure 16 presents the high-level search tree generated for an instance of Sokoban. For each expanded state, the direct neighbours on the right side are the subgoals proposed by the subgoal generator that was given that state.
> >
> > The process of building the search tree for each method is also schematically presented in Appendix H.

---

> > ### Comment · Reviewer_ssrG · 2025-07-08
> > **Is there anything stochastic happening?**
> >
> > I'm afraid that I'm still perplexed at the choice of formalism here. MCTS and the Alpha-line of research makes sense -- uncertain opponent play is modeled as uncertain effects and MDP/MCTS entirely make sense. But haven't a comparison of MCTS next to A*+BestFS just doesn't. The natural extension of A* in the MDP setting is AO* or LAO*, but as far as I can tell, there isn't anything probabilistic or (qualitatively uncertain) happening here.
> >
> > DeepCubeA is also not what I would consider a classical planner. You can find the world's best classical planners from the regularly held IPC (International Planning Competition), and they would all perform exceedingly better. It may be the case that it doesn't make sense to compare with them -- perhaps you are making a claim that the model isn't available or the successor state function isn't accessible -- but this should be articulated. Classical planners do not require domain-specific heuristics -- only a model of the domain mechanics (via PDDL).

---

> > > ### Author Response · Authors · 2025-07-10
> > >
> > > We thank the reviewer for their helpful feedback. As now detailed in Section 3.1, *Problem Formulation*,  we cast each deterministic benchmark as a point-mass MDP. This allows every planner to be discussed in the shared language of value functions and policies.
> > >
> > > We agree that MCTS is mainly suited for stochastic domains and have included it only as a completeness baseline whose rollouts buffer noise in the learned heuristic. DeepCubeA is offered merely as a learning-based substitute for classical planners when a successor generator is available but no handcrafted heuristics are provided.
> > >
> > > While INT could, in principle, be encoded in PDDL, this would require enumerating every possible algebraic term and inequality in advance. Such a process would cause the grounded model to grow exponentially with proof depth, overwhelming even current IPC planners. Therefore, we judged the comparison to be impractical.

---

### Review · Reviewer_ig3h · 2025-06-11

**Summary Of Contributions:**

This paper provides guidance on when hierarchical search should be preferred over low-level search in combinatorial problems. It benchmarks two hierarchical algorithms (kSubS and AdaSubS) against three flat baselines (ρ-BestFS, ρ-A*, ρ-MCTS) on four classic tasks—Rubik’s Cube, Sokoban, N-Puzzle, and Inequality Theorem Proving (INT). Through analytical experiments and supporting theory, the paper shows that hierarchical methods perform well when (i) value functions are difficult to learn, (ii) the action space is large or redundant, (iii) the environment contains dead ends, or (iv) training data come from diverse sources.

**Audience:**

Yes

**Claims And Evidence:**

Yes

**Requested Changes:**

See above.

**Strengths And Weaknesses:**

__Strengths:__
1. The paper is clearly written and well-motivated.
2. Code is provided, making it reproducible.
3. The analysis is relatively comprehensive. Some empirical results are supported by theoretical justification.

---

__Weaknesses:__
1. Both kSubS and AdaSubS stem from the same design family. It seems there are many more different hierarchical search methods, like the HIPS in the related works. I am somewhat concerned about the generality of the conclusion. It would be better to experiment with an additional hierarchical method.
2. Although the author(s) tested on four benchmark problems, I feel like most of them are very similar. It would be better to further analyse other combinatorial problems, like the Traveling Salesperson Problem (TSP) or Maximum Independent Set (MIS). Also, more details on the chosen combinatorial tasks could be clarified to ensure they are representative of a broad range of NP-hard problems, including an explanation of the criteria used for task selection and how representativeness was ensured.
3. While the paper provides a comprehensive comparison between hierarchical methods and low-level search algorithms, additional comparisons with traditional heuristics or other learning-based neural solvers would strengthen the evaluation and further prove the conclusion.
4. Except for the four aspects where hierarchical methods are shown to be useful in the paper, are there any other aspects that have not been taken into consideration? For example, generalization to larger problem scales?
5. Besides wall-clock time analysis, analysis of the time and space complexity of the proposed hierarchical method and low-level methods used for comparison can also be clarified to ensure fairness in the comparison.
6. More details about the experiments conducted could be provided in the manuscript to enhance reproducibility.
7. Sensitivity of the overall method to random seed / multiple trials is not reported.

---

> ### Author Response · Authors · 2025-07-07
> **Response to Reviewer ig3h (Part 1)**
>
> > Both kSubS and AdaSubS stem from the same design family. It seems there are many more different hierarchical search methods, like the HIPS in the related works. I am somewhat concerned about the generality of the conclusion. It would be better to experiment with an additional hierarchical method.
>
> We agree with your point. As noted in the Limitations section, our empirical analysis would indeed be strengthened by including a broader range of algorithms and domains. We chose to focus on Subgoal Search because (1) it is a straightforward extension of BestFS, obtained by simply adding a subgoal generator to the same pipeline, allowing us to isolate the impact of the hierarchical component, (2) it has been shown to be effective in combinatorial problems, which (unlike HIPS) was also confirmed by our experiments. We see this as a foundational step toward generalizing to other hierarchical methods in future work.
>
> It is important to emphasize that while we use these specific algorithms to gather empirical observations, our analysis is framed from the perspective of a general hierarchical scheme (as detailed in Section 4). In particular, the theoretical guarantees (such as the Search Advancement Formula and the Action Space Densification Theorem) are proven for the general class of hierarchical methods.
>
> To clarify this point, we have added a remark to the Discussion of the Results section. Thank you for this feedback.
>
> > Although the author(s) tested on four benchmark problems, I feel like most of them are very similar. It would be better to further analyse other combinatorial problems, like the Traveling Salesperson Problem (TSP) or Maximum Independent Set (MIS). Also, more details on the chosen combinatorial tasks could be clarified to ensure they are representative of a broad range of NP-hard problems, including an explanation of the criteria used for task selection and how representativeness was ensured.
>
> Thank you for your feedback on our benchmark selection. We agree that explaining the choice of our tasks is important and we add this information to the Analysis section. We selected four widely studied environments – Sokoban, N-Puzzle, Rubik's Cube, and INT – because each has distinct properties that help us test specific properties hierarchical search:
> Dead Ends: Sokoban has many irreversible moves that create dead-end states where the goal becomes unreachable. This lets us test if subgoal methods better avoid these states (Section 5.4).
> Action Space Complexity: INT has a very large action space where each action involves selecting an axiom and inputs, creating millions of possible moves per step. This differs from other environments with small, simple action spaces. INT helps us study algorithm performance with complex actions (Section 5.3).
> State Space and Solution Length: Rubik's Cube and N-Puzzle have huge state spaces but simple actions. They help us study long-horizon tasks and learning from diverse, suboptimal datasets (Section 5.1).
> Furthermore, in the Future Directions section, we acknowledge that extending our evaluation to more environments would further strengthen our results. Following your suggestion, we now explicitly mention TSP and MIS as good examples of such environments.
>
> > While the paper provides a comprehensive comparison between hierarchical methods and low-level search algorithms, additional comparisons with traditional heuristics or other learning-based neural solvers would strengthen the evaluation and further prove the conclusion.
>
> We appreciate this suggestion and have included a comparison with DeepCubeA, a prominent learning-based neural solver that uses deep reinforcement learning to learn value functions and heuristics for puzzle solving (McAleer et al., 2019). DeepCubeA employs Iterative Deepening A* (IDA*) as its core search algorithm, which performs depth-first searches with increasing cost thresholds and may revisit nodes multiple times.
> Our comparison shows that DeepCubeA's performance is weaker than or on par with the hierarchical and low-level search methods analyzed in our study across three domains: Rubik's Cube, Sokoban, and N-Puzzle (Figure 42). This demonstrates that our implementations are competitive with established general-purpose solvers that use learned heuristics.

---

> > ### Author Response · Authors · 2025-07-07
> > **Response to Reviewer ig3h (Part 2)**
> >
> > > Except for the four aspects where hierarchical methods are shown to be useful in the paper, are there any other aspects that have not been taken into consideration? For example, generalization to larger problem scales?
> >
> > This is an interesting question. Beyond the four main aspects, we evaluated two additional axes of scale.
> >
> > (1) On out-of-distribution Sokoban boards, including DeepMind’s “extra-hard’’ set. Hierarchical planners maintained a clear lead over flat baselines, especially in the most difficult instances.
> > (2) In enormous action spaces (INT proof search and a Rubik’s-Cube variant with inflated action space), subgoal methods maintained over 90 % success rate where flat search collapsed.
> >
> > Together, these results demonstrate generalisation in both state, action, and task difficulty scale.
> >
> > > Besides wall-clock time analysis, analysis of the time and space complexity of the proposed hierarchical method and low-level methods used for comparison can also be clarified to ensure fairness in the comparison.
> >
> > We fully agree with this point. Since our methods share the general structure described in Section 4, they also share the time and space complexity. More specifically:
> >
> > * BestFS keeps the visited states on a priority queue and the set of visited nodes. Hence, adding a single node is done in O(log N) time. In total, expanding the search tree with N actions takes O(N log N) time and O(N) space.
> > * A* requires to additionally store depth. However, that doesn’t change the complexity: O(N log N) time and O(N) space.
> > * kSubS for each node added to the search tree (both high-level and low-level) calls either the subgoal generator or low-level policy at most once, hence the total number of neural network calls is O(N). It follows the general scheme of BestFS in building in storing the search tree. However, only the high-level nodes are added to the search tree and the visited set, so its time complexity is O(N + N/k log(N/k)) = O(N/k log N), and space complexity is O(N/k).
> > * AdaSubS follows the design of kSubS. From the complexity point of view, the only difference is that the queue is populated with new state instance for each generator used in the set. Hence, the time complexity is bounded by O(N*g/k log N) and space complexity O(N*g/k).
> > * MCTS traverses a whole path in the tree for every expansion. Hence, pessimistically it can take O(N) time for each expansion, giving in total O(N^2) time complexity and O(N) space. However, we note that in practice MCTS is not much slower than other methods.
> >
> > Interestingly, kSubS has the lowest time and space complexity, while MCTS has the highest among all studied algorithms.
> >
> >  > More details about the experiments conducted could be provided in the manuscript to enhance reproducibility.
> >
> > We agree with the suggestion. In the original submission, most of the technical details were confined to Appendix C (training protocol & architecture) and Appendix E (data collection). In the revision we have moved the key information into the main text (Section 4 Training Component) so it is immediately visible without consulting the appendices. Concretely, we now provide:
> > | Task         | LR₁         | LR₂         | LR₃         | LR₄         |
> > |--------------|-------------|-------------|-------------|-------------|
> > | INT          | 1 × 10⁻⁴    | 1 × 10⁻⁴    | 3 × 10⁻⁴    | 1 × 10⁻⁴    |
> > | Rubik’s Cube | 1 × 10⁻⁴    | 5 × 10⁻⁴    | 3 × 10⁻⁷    | 1 × 10⁻⁴    |
> > | Sokoban      | 1 × 10⁻⁵    | 1 × 10⁻⁴    | 1 × 10⁻⁴    | 1 × 10⁻⁴    |
> > | n-Puzzle     | 1 × 10⁻⁴    | 1 × 10⁻⁴    | 1 × 10⁻⁴    | 1 × 10⁻⁴    |
> >
> > Architectures: All components are implemented with HuggingFace Transformers. Sub-goal generators and INT policies (CLLP + baseline) use BART-large (Lewis et al., 2020); the remaining policies and value functions use BERT-base (Devlin et al., 2019).
> >
> >
> > Other hyper-parameters. AdamW, batch = 64, gradient-clip = 1.0, early-stopping patience = 10, three fixed random seeds.
> >
> >
> > Compute. Training runs on a single NVIDIA A100 (40 GB) GPU for ≤ 48 h per component; data loading uses one CPU core. Evaluation uses a 24-core Xeon Platinum 8268 node with 192 GB RAM
> >
> > > Sensitivity of the overall method to random seed / multiple trials is not reported.
> >
> > We agree that this is an important point. The sensitivity of the studied algorithms to random seeds is minimal. Each component was trained and evaluated in over 30 independent runs with different random seeds. All are trained within a behavioral-cloning setup, a training paradigm known to be very stable. This clarification has been added to the Training Component section of the main text.

---

### Review · Reviewer_17fV · 2025-06-29

**Summary Of Contributions:**

The paper compares hierarchical search with traditional low-level search methods for solving combinatorial optimization problems. Specifically, it focuses on hierarchical search with ML-based heuristics. Empirical results show that they can leverage training data from different sources more effectively, they are less sensitive to the quality of the value function, they handle complex action space better and avoids dead-end, they generalizes to unseen problem distribution.

**Audience:**

Yes

**Broader Impact Concerns:**

I don't have concerns.

**Claims And Evidence:**

Yes

**Requested Changes:**

1. Can you try to provide runtime results for each figure/table in Section 5? Can you provide more analysis on wall-clock runtime to support your claim? Can you also discuss some potential solutions to reduce the ML inference time?
2. Hierarchical is a sub-optimal search. Can you also help us understand it effectiveness (i.e., tradeoff between runtime vs solution quality)?
3. Can you provide motivations for studying ML-guided hierarchical search?

**Strengths And Weaknesses:**

Strengths:

1. This paper sets up a good question to study and provides thorough experiment results to support their claims. Some of the claims are also backed by theoretical analysis.
2. The main findings could potentially provide good and useful insights into hierarchical search and facilitate future study.




Weaknesses:

1. The main confusion I have with this paper is why the focus is specifically on subgoal-based hierarchical search methods with learned heuristics. “What matters in hierarchical search?” Is also a great question to ask for classical hierarchical search. I don’t quite get what is special and different in Machine learning-based hierarchical search vs. the classical one. Or in other words, why is hierarchical search with learned heuristics more interesting than the classical ones? At the very least, you need to include those non-ML based searches as your baseline.
2. The author provided wall-clock time analysis, argued that the number of states is a more fair metric. However, the results for wall-clock runtime is far from sufficient, since only the result for one experiment is reported in Appendix G. And the big margin in wall-clock time compared to the baseline is a concern. Also, it doesn’t seem that the authors try to make any effort to reduce the runtime overhead. There are simple ways to reduce it for ML models that serve as a heuristic value estimator. For example, see this paper: “Neural Neighborhood Search for Multi-agent Path Finding” (it is a irrelevant paper but I happen to know that the authors of this paper managed to significantly reduce the overhead for its neural heuristic estimator via distillation.) Last but not least, the authors claim the runtime “suffers from high variance” and validated “their correlation with the complete search budget”. However, none of these were supported in the discussion in the appendix.
3. The paper includes a lot of contents, making it difficult to understand/find the details for many things. For example: (1) it is hard to understand what diverse sources of data mean until one reads the appendix; (2) the big picture of the ML part (architectures, loss functions, hyper parameters etc.) is mostly missing from the main text.

---

> ### Author Response · Authors · 2025-07-07
> **Response to Reviewer 17fV (Part 1)**
>
> > The main confusion I have with this paper is why the focus is specifically on subgoal-based hierarchical search methods with learned heuristics. “What matters in hierarchical search?” Is also a great question to ask for classical hierarchical search. I don’t quite get what is special and different in Machine learning-based hierarchical search vs. the classical one. Or in other words, why is hierarchical search with learned heuristics more interesting than the classical ones? At the very least, you need to include those non-ML based searches as your baseline.
>
> > Can you provide motivations for studying ML-guided hierarchical search?
>
> Thank you for this thoughtful comment. We agree that understanding the fundamentals of hierarchical search is a broadly valuable goal, and classical methods are an important part of this landscape. Our focus on subgoal-based hierarchical search with learned heuristics stems from its growing popularity in modern problem-solving, particularly in settings where handcrafted heuristics are unavailable or insufficient. Learned heuristics introduce unique challenges – such as limited generalization, training data bias, and sensitivity to distributional shifts – that do not arise in classical approaches. These challenges directly motivate our analysis of when and why hierarchical structures help in such settings.
>
> While classical hierarchical planners are informative, they often rely on domain-specific decomposition strategies or hand-crafted subgoal generators, which makes direct comparison with learning-based methods nontrivial. That said, we acknowledge the value of including such baselines for broader context and we clarified that in the Future Directions section.
>
> > The author provided wall-clock time analysis, argued that the number of states is a more fair metric. However, the results for wall-clock runtime is far from sufficient, since only the result for one experiment is reported in Appendix G. And the big margin in wall-clock time compared to the baseline is a concern. Also, it doesn’t seem that the authors try to make any effort to reduce the runtime overhead. There are simple ways to reduce it for ML models that serve as a heuristic value estimator ... Last but not least, the authors claim the runtime “suffers from high variance” and validated “their correlation with the complete search budget”. However, none of these were supported in the discussion in the appendix.
>
> > Can you try to provide runtime results for each figure/table in Section 5? Can you provide more analysis on wall-clock runtime to support your claim? Can you also discuss some potential solutions to reduce the ML inference time?
>
> Thank you for raising these important points. We agree runtime is a valid concern for deployment, and future work will prioritize optimization. We acknowledge its relevance, even though using wall-clock time as a budget metric has clear drawbacks – such as confusing algorithmic efficiency with hardware, software, and implementation variability, which makes comparisons non-reproducible, non-portable, and potentially misleading. Hence, in the appendix we provided the runtimes for the evaluation in each domain, for each algorithm that we consider.
>
> That said, obtaining these numbers required modifying the standard evaluation pipeline, including disabling experiment batching (as total time strongly depended on the distribution of solution lengths within batches), ensuring consistent hardware (same cluster and preferably the same node), etc. Also, it is important to note that measuring the wall time of the whole evaluation leads to confusing quickly finding solutions in successful episodes with quickly giving up in unsuccessful episodes. Measuring time for episodes separately excludes any parallelization. In contrast, a discrete search budget has none of those drawbacks.
>
> We fully agree that several standard techniques could significantly reduce the computational overhead, such as:
> * Using a more efficient model: we chose the BART architecture for consistency with the Subgoal Search paper, though more efficient alternatives are now available;
> * Carefully tuning the model size, or distilling it to a smaller version;
> * Leveraging decoding parallelization: as noted in Appendix G, the Hugging Face implementation we used was affected by a bug disabling parallel decoding in the subgoal generator. Fixing this or switching to another library would reduce inference overhead;
> * Using dedicated transformer acceleration libraries (e.g., Ollama);
> * Operating on compact state abstractions rather than raw representations.
>
> Since our focus was on the Complete Search Budget, which is discrete, reproducible, and hardware-independent, we did not prioritize these optimizations in our experiments. However, we agree that they offer a promising path to reduce wall-clock overhead. We have extended the discussion in Appendix G to reflect these points. Thank you again for the insightful feedback.

---

> > ### Author Response · Authors · 2025-07-07
> > **Response to Reviewer 17fV (Part 2)**
> >
> > > The paper includes a lot of contents, making it difficult to understand/find the details for many things. For example: (1) it is hard to understand what diverse sources of data mean until one reads the appendix; (2) the big picture of the ML part (architectures, loss functions, hyper parameters etc.) is mostly missing from the main text.
> >
> > Thank you for bringing our attention to that point, we have clarified the description of diverse data sources in section 5.1. Additionally, we have expanded the description of the learning setup, including model architectures, loss functions, and key hyperparameters in the main text. Further implementation details remain in the appendix for completeness.
> >
> > > Hierarchical is a sub-optimal search. Can you also help us understand it effectiveness (i.e., tradeoff between runtime vs solution quality)?
> >
> > That’s a very interesting suggestion. We analyzed the quality of the solutions generated by the studied methods in a new Appendix N, and we highlight the key takeaways below.
> >
> > Quite surprisingly, in each domain, the shortest solutions are found by p-MCTS. p-A* typically performs only slightly worse in that metric. While A* is theoretically capable of finding optimal solutions, our heuristic is not guaranteed to be admissible, and we additionally apply depth weighting for greater efficiency, which introduces suboptimality. Interestingly, Subgoal Search finds shorter solutions than its low-level counterpart p-BestFS in Sokoban and N-Puzzle, but longer ones in the Rubik’s Cube. We also compared the solutions found in Sokoban to optimal paths computed using BFS. On average, the computed solutions are about 6–9 steps longer.
> >
> > We note that the quality of solutions found by hierarchical search is bounded by (1) how often the generator proposes subgoals on the optimal path, (2) the quality of low-level paths computed by the policy, and (3) the ability of the value function to recognize optimal states. In contrast, low-level methods are bounded only by the value function (assuming they can expand actions exhaustively). However, we note that AdaSubS, one of the hierarchical methods we evaluate, can effectively leverage both long-horizon and short, 1-step subgoals, helping bridge this gap. We believe that further tuning that method could produce a hierarchical search that explicitly balances effectiveness and solution quality. However, while this is an exciting direction to explore, it lies beyond the scope of our current work and we leave it for future research.

---

### Author Response · Authors · 2025-07-07
**Summary of Reviews and Revisions**

Dear Reviewers,

Thank you for your time and for the thoughtful and constructive feedback on our manuscript. We are encouraged by your positive assessment, particularly your recognition of the work as suitable for publication (all Reviewers), its clear and well-written presentation (ig3h, ssrG), the strength and motivation of the research question (17fV, ig3h), the thoroughness of the experimental analysis (all Reviewers), and the value of the two theorems we present (all Reviewers).

We have addressed all comments and questions in the individual responses and have revised the manuscript accordingly. **We have marked the changes to the manuscript in blue so you can easily assess what has been changed**. If there are any remaining points of clarification, we would be happy to continue the discussion.

Warm regards,
The Authors

---

### Decision · Action_Editor_x9XY · 2025-08-08

**Recommendation:** Reject

**Additional Comments:**

As noted in the author reviews and official recommendations, the paper decision is based on scope and claims.

Please note that this evidence supporting the claims made in the paper is one of the two main acceptance criteria at TMLR.

In reading the responses, there is a consistent theme of the scope of the paper mismatching the claims and lack of baselines that lead to misleading (or muddled) conclusions. The rebuttals often try to justify the choices made by the authors, but I largely feel that they're not taking the criticisms very seriously.

Here is one example: one reviewer claimed that an algorithm the authors refer to is NP-hard when in fact it's PSPACE-complete. The author's response was:

"We would like to clarify that since NP \subseteq PSPACE, every PSPACE-complete problem is also NP-hard. While we agree that your point is theoretically more precise, we have chosen to refer to NP rather than PSPACE, as NP is more commonly used as a point of reference."

This is akin to saying that "the worst-cast running time of merge sort is O(n^2)". Or exponential... or O(n!). All of these are technically true, but they are quite misleading because the upper-bounding function is not tight, which is clearly the purpose of discussing worst-case time complexities. Saying something is NP-hard when it's PSPACE-complete is just outright misleading, and rather than seeing the logic and simply clarifying the text, the authors choose to defend their choice to mislead readers!

This is just one example; there are others. I find this trend disconcerting as it is starting causing some tension.. one of the reviewers was noticeably short in tone in their official recommandation, and the reason is quite clear: this persistence in the rebuttals and resubmissions is in conflict with the spirit of the review process.

This is the second time I'm Aciton Editor of this paper. It was rejected in the previous round for reasons that are similar to the main reasons presented in this decision. Please do not resubmit until these problems are genuinely resolved.

**Audience:**

Yes

**Audience Explanation:**

Yes... but it's not a perfect fit.

Classicial planning is an technique more closley association with classical AI venues like AAAI/IJCAI/ICAPS rather than the machine learning communities.

There are learned heuristics, which do make it relevant.. but TMLR is not the best choice of audience for this paper. JAIR would be a better audience fit.

**Claims And Evidence:**

No

**Claims Explanation:**

This is the main problem with the paper.

In the official recommendations, there were two rejects and one accept. In the official recommendation, here were the comments I felt best represented the opinions of the reviewers:

- "Some of the statements are misleading, the scope of the paper is smaller than the authors claimed in the paper."
- "I think the authors should further clarify the scope of applicability of their method, and compare it more thoroughly with other classical and learning-based methods. While some of this has been discussed in this modified version, more refinement would be beneficial."
- "The work remains a mix of formalisms and solving tech that seems to be a mismatch from the proposed setting."

More detail in the additional comments section.